# Ribonucleotide incorporation into mitochondrial DNA drives inflammation

Amir Bahat[1,8], Dusanka Milenkovic[1,8], Eileen Cors[1], Mabel Barnett[1], Sadig Niftullayev[1], Athanasios Katsalifis[1], Marc Schwill[1], Petra Kirschner[1], Thomas MacVicar[1,7], Patrick Giavalisco[1], Louise Jenninger[2], Anders R. Clausen[2], Vincent Paupe[3], Julien Prudent[3], Nils-Göran Larsson[4], Manuel Rogg[5], Christoph Schell[5], Isabella Muylaert[2], Erik Larsson[2], Hendrik Nolte[1], Maria Falkenberg[2] & Thomas Langer[1,6 ✉]

Metabolic dysregulation can lead to inflammatory responses[1,2]. Imbalanced nucleotide synthesis triggers the release of mitochondrial DNA (mtDNA) to the cytosol and an innate immune response through cGAS–STING signalling[3]. However, how nucleotide deficiency drives mtDNA-dependent inflammation has not been elucidated. Here we show that nucleotide imbalance leads to an increased misincorporation of ribonucleotides into mtDNA during age-dependent renal inflammation in a mouse model lacking the mitochondrial exonuclease MGME1[4], in various tissues of aged mice and in cells lacking the mitochondrial i-AAA protease YME1L. Similarly, reduced deoxyribonucleotide synthesis increases the ribonucleotide content of mtDNA in cell-cycle-arrested senescent cells. This leads to mtDNA release into the cytosol, cGAS–STING activation and the mtDNA-dependent senescence-associated secretory phenotype (SASP), which can be suppressed by exogenously added deoxyribonucleosides. Our results highlight the sensitivity of mtDNA to aberrant ribonucleotide incorporation and show that imbalanced nucleotide metabolism leads to age- and mtDNA-dependent inflammatory responses and SASP in senescence.

Metabolic and signalling functions of mitochondria ensure cell survival but, when disrupted, are associated with inflammation, cell death and disease[5,6]. Various mitochondrial stressors, such as pathogen infection[7], defective mtDNA packaging[7–9], and pyrimidine[3] or fumarate hydratase deficiency[10], trigger the release of mtDNA into the cytosol and an inflammatory response. mtDNA binding to cGAS induces cGAS–STING–TBK1 signalling and the expression of type I interferons and interferon-stimulated genes (ISGs)[11–13]. This response provides protection against pathogens[7,14], but can also promote autoimmune and inflammatory diseases and contribute to senescence and ageing[15,16].

Studies of the mitochondrial protease YME1L revealed the critical role of metabolic cues for mtDNA release and inflammation[3]. YME1L-mediated proteolysis rewires the mitochondrial proteome in response to nutrient stress to support anaplerotic reactions and pyrimidine metabolism—a prerequisite for the growth of certain solid tumours and the maintenance of adult neuronal stem cells[17,18]. Nucleotide imbalance in cells lacking YME1L or after pharmacological inhibition of cytosolic pyrimidine biosynthesis by chemotherapeutic agents induces mtDNA release and inflammation[3]. However, the physiological relevance of nucleotide imbalance as a driver of mtDNA-dependent inflammation in ageing and disease and the mechanisms by which nucleotide imbalance or other mitochondrial stressors affect mtDNA, culminating in its release from mitochondria, remained unclear.

## mtDNA drives inflammation in *Mgme1⁻/⁻* mice

Mice lacking MGME1, a mitochondrial exonuclease regulating mtDNA maintenance, develop renal inflammation and start dying prematurely from renal failure at around 1 year of age[19–21]. To investigate whether mitochondria-dependent innate immune signalling contributes to this phenotype, we monitored the mRNA levels of a panel of ISGs in the kidneys of wild-type (WT) and *Mgme1⁻/⁻* mice at different ages. ISG expression was induced with age in *Mgme1⁻/⁻* mice and was highest in mice aged 70 weeks (Fig. 1a and Extended Data Fig. 1a,b). It occurred in a tissue-specific manner and was not observed in spleen, heart, liver or brain tissues of 55-week-old *Mgme1⁻/⁻* mice (Fig. 1b and Extended Data Fig. 1c). As mtDNA released into the cytosol may induce the innate immune response in vivo, we isolated mitochondrial and cytosolic fractions from kidneys of 55-week-old mice and analysed them using digital PCR (dPCR) using a set of mtDNA-specific probes spanning the control region, major and minor arcs of mtDNA (Fig. 1c). Although cytosolic fractions isolated from WT and *Mgme1⁻/⁻* kidneys largely lack mitochondrial proteins (Extended Data Fig. 1d), mtDNA was detected in the *Mgme1⁻/⁻* cytosol (Fig. 1c). We observed about a fivefold and threefold increase in mtDNA fragments containing the displacement loop (D-loop) region and cytochrome B (*Cytb*), respectively, in the cytosolic fractions of *Mgme1⁻/⁻* kidneys (Fig. 1c). By contrast, mtDNA regions more distant to the origin of heavy-strand replication ($O_H$) were not enriched (Fig. 1c). We also observed an

[1]Max Planck Institute for Biology of Ageing, Cologne, Germany. [2]Institute for Biomedicine, University of Gothenburg, Gothenburg, Sweden. [3]Medical Research Council Mitochondrial Biology Unit, University of Cambridge, Cambridge, UK. [4]Department of Medical Biochemistry and Biophysics, Karolinska Institutet, Stockholm, Sweden. [5]Institute of Surgical Pathology, Faculty of Medicine, Medical Center—University of Freiburg, Freiburg, Germany. [6]Cologne Excellence Cluster on Cellular Stress Responses in Aging-Associated Diseases (CECAD), University of Cologne, Cologne, Germany. [7]Present address: The CRUK Scotland Institute, Glasgow, UK. [8]These authors contributed equally: Amir Bahat, Dusanka Milenkovic. ✉e-mail: tlanger@age.mpg.de

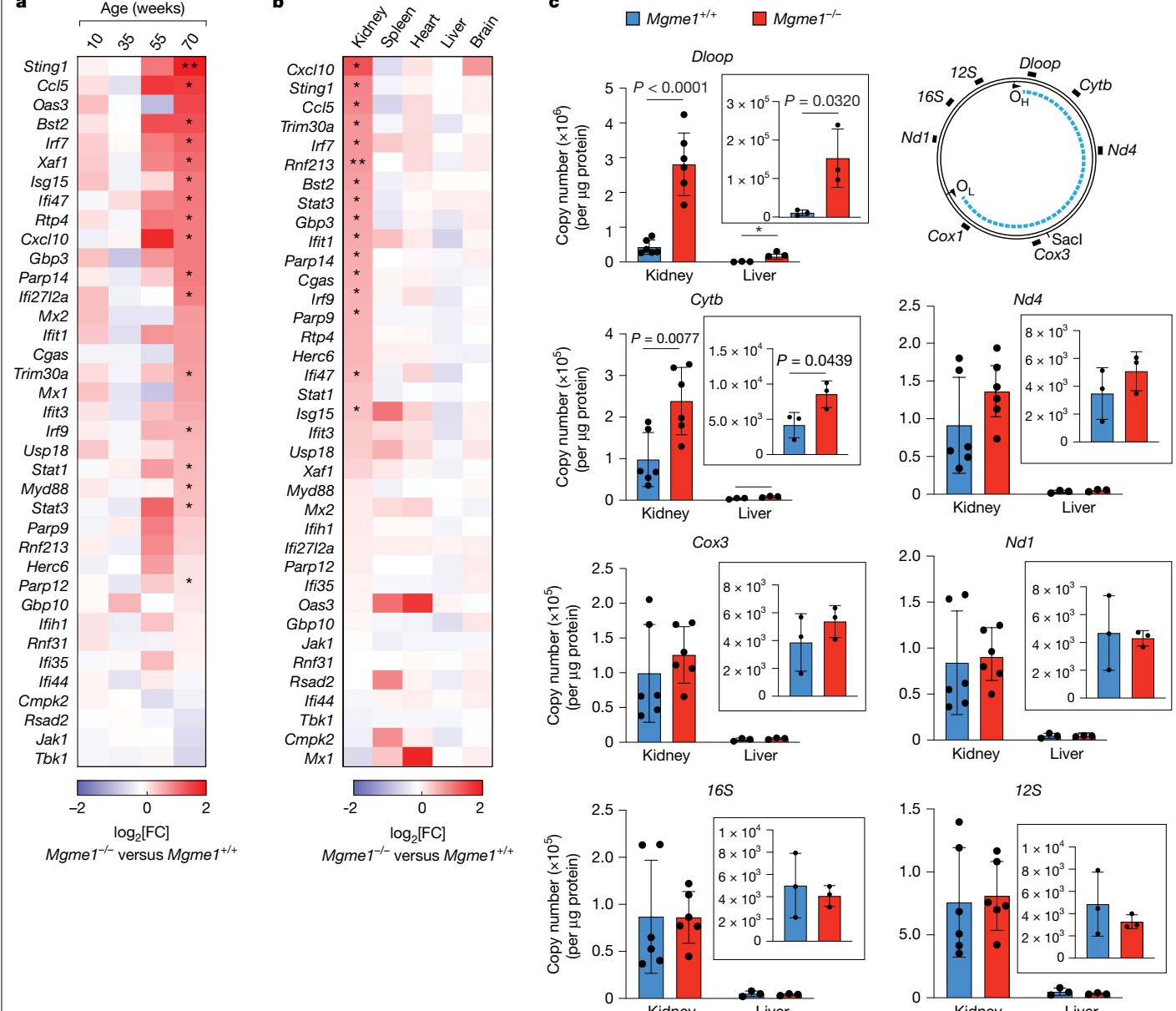

**Fig. 1 | Late-onset kidney inflammation in *Mgme1*⁻/⁻ mice. a**, Row-clustered heat map (log₂-transformed fold change (FC)) of mRNA ISG profiles obtained by NanoString profiling in *Mgme1*⁺/⁺ and *Mgme1*⁻/⁻ mouse kidneys at different ages. The asterisks mark ISGs of which the expression was significantly altered between *Mgme1*⁻/⁻ and *Mgme1*⁺/⁺ mouse kidneys at 70 weeks of age. *n* = 2 mice per genotype at 10, 35 and 55 weeks and *n* = 3 mice per genotype at 70 weeks. **b**, Heat map (log₂-transformed fold change, decreasing order) of mRNA ISG profiles obtained by NanoString profiling of various tissues of 55-week-old WT (*Mgme1*⁺/⁺) and *Mgme1*⁻/⁻ mice. The asterisks mark ISGs of which the expression

was significantly altered between *Mgme1*⁻/⁻ and *Mgme1*⁺/⁺ mice in the corresponding tissue. *n* = 3 mice per genotype per tissue. **c**, dPCR analysis of mtDNA levels in cytosolic fractions of kidney (*n* = 6 mice per genotype) and liver (*n* = 3 mice per genotype) tissue from 55-week-old control (*Mgme1*⁺/⁺) and *Mgme1*⁻/⁻ mice. A schematic of the mtDNA and of the probes used in the assay is shown at the top right. The blue dashed line represents the major arc, and the remaining part of mtDNA is minor arc. *P* values were calculated using unpaired two-tailed Student *t*-tests (**c**). Data are mean ± s.d. *q < 0.05, **q < 0.01 (**a**,**b**).

enrichment of cytosolic mtDNA fragments in the cytosolic fractions of *Mgme1*⁻/⁻ mouse liver, although at considerably lower levels (Fig. 1c (insets)). These data coincide with the lack of inflammation in the liver of *Mgme1*⁻/⁻ mice (Fig. 1b). It is conceivable that mechanisms for cytosolic DNA removal, such as the 3′ repair exonuclease 1 (TREX1)[22], are sufficient in preventing innate immune response induction under low levels of mtDNA release[3]. We conclude that the loss of MGME1 in the kidneys leads to an age-related innate immune response, which is associated with the release of mtDNA fragments predominantly derived from mtDNA regions proximal to the $O_H$. In accordance with the release of mtDNA fragments, TFAM—the main

protein nucleoid constituent—was not enriched in the cytosolic kidney fraction (Extended Data Fig. 1e).

## STING-dependent renal inflammation

To investigate whether cytosolic mtDNA fragments in the kidney of aged *Mgme1*⁻/⁻ mice trigger an inflammatory response through cGAS–STING–TBK1 signalling, we crossed *Mgme1*⁻/⁻ mice with mice carrying a loss-of-function mutation in *Sting1* (encoding STING, also known as *Tmem173*) to generate *Mgme1*⁻/⁻ *Sting1*ᵐᵘᵗ/ᵐᵘᵗ (hereafter, double-knockout (DKO)) mice. The strong ISG response in the kidneys of 55-week-old

*Mgme1⁻/⁻* mice was blunted after loss of STING in age-matched DKO mice, demonstrating STING involvement in the inflammatory response in vivo (Fig. 2a and Extended Data Fig. 2a).

In agreement with our in vivo results, *Mgme1⁻/⁻* primary mouse embryonic fibroblasts (MEFs) exhibited a pronounced ISG response, which was strongly suppressed after deletion of *Sting1* (Extended Data Fig. 2b,c). Depletion of cGAS or STING, or inhibition of TBK1 by BX795, inhibited ISG expression in immortalized *Mgme1⁻/⁻* cells (Extended Data Fig. 3a–c), whereas loss of RNA sensors showed no effect (Extended Data Fig. 3d,e). Moreover, cGAS, which resides mainly in the nucleus of mammalian cells under basal conditions[23], accumulated in the cytosol of *Mgme1⁻/⁻* cells (Extended Data Fig. 4a,b). Finally, we detected increased levels of cytosolic mtDNA in *Mgme1⁻/⁻* MEFs by digital droplet PCR (ddPCR) in cell fractionation experiments (Fig. 2b and Extended Data Fig. 4c). Immunofluorescence microscopy revealed an increased number of *Mgme1⁻/⁻* cells with cytosolic DNA foci and an increased number of cytosolic DNA foci per cell (Fig. 2c,d). Depletion of mtDNA using the chain-terminating nucleotide analogue 2′,3′-dideoxycytidine (ddC) inhibited innate immune signalling after MGME1 depletion or in *Mgme1*-deficient cells (Extended Data Fig. 5a–c). Together, these experiments show that the loss of MGME1 causes the release of mtDNA to the cytosol, activation of cGAS–STING–TBK1 signalling and expression of ISGs.

As the loss of STING suppressed renal inflammation in *Mgme1⁻/⁻* mice (Fig. 2a), we examined the nephropathy in DKO mice (Fig. 2e). We performed histological analysis of 55-week-old mice and found that STING ablation ameliorated renal phenotypes of *Mgme1⁻/⁻* mice, including glomerular sclerosis, accompanied by the formation of proteinaceous casts and periglomerular infiltrates of B220⁺ and CD3⁺ T cells (Fig. 2e). These results suggest that the disruption of the glomerular filtration barrier is the causative factor for the observed renal phenotype and establish a critical role for STING signalling in the disease (Fig. 2e).

## mtDNA-replication-dependent inflammation

The function of MGME1 as a mitochondrial replicative exonuclease raised the possibility that impaired mtDNA replication triggers an mtDNA-dependent immune response. However, in contrast to MGME1-deficient cells, we did not detect an ISG response after depletion of enzymes of the mtDNA replisome, including *PolgA*, *Twnk*, *Ssbp1*, *Polrmt* and *Rnaseh1* (Fig. 3a and Extended Data Fig. 6a). On the contrary, depletion of the helicase *Twnk*, *Ssbp1* or *Polg2* suppressed the ISG response in *Mgme1⁻/⁻* cells (Fig. 3b and Extended Data Fig. 6b,c), suggesting that ongoing mtDNA synthesis is essential to trigger mtDNA release.

Although the loss of essential mtDNA replisome components completely inhibits mtDNA replication, accumulation of a long linear subgenomic mtDNA species and unproductive mtDNA replication have been observed in *Mgme1⁻/⁻* mice[4]. Next-generation sequencing of mtDNA isolated from the kidney of *Mgme1⁻/⁻* mice revealed a decrease in sequence coverage in regions with increasing distance from $O_H$ (Fig. 3c). These results are consistent with the previously described replication-stalling phenotype in brain and heart tissues[4] and the increased detection of DNA fragments proximal to $O_H$ in the cytosol (Fig. 1c). Notably, the deep sequencing analysis of *Mgme1⁻/⁻* kidney suggests more mtDNA replication initiation events from $O_H$ when compared to WT (Fig. 3c). This has not been observed in other tissues[4] and may explain the observed tissue specificity of mtDNA-dependent inflammation.

Increased replication initiation and therefore increased nucleotide use in *Mgme1⁻/⁻* cells may deplete available deoxynucleotide pools, leading to cellular nucleotide imbalance and mtDNA-dependent innate immune signalling. We therefore determined nucleotide levels in *Mgme1⁻/⁻* cells using liquid chromatography–mass spectrometry (LC–MS). We observed decreased levels of deoxyribonucleotide

triphosphates (dNTPs) with a predominant increase in the levels of ribonucleotide monophosphates (rNMPs) (Fig. 3d and Extended Data Fig. 6d). As cellular dNTP pools are increased after depletion of the SAM domain and HD-domain containing deoxynucleoside triphosphate triphosphohydrolase 1 (SAMHD1)[24], we depleted SAMHD1 from *Mgme1⁻/⁻* cells. SAMHD1 depletion restored dNTP levels (Extended Data Fig. 6e) and suppressed the ISG response (Fig. 3e), demonstrating the critical role of reduced dNTP pools for the immune response. Moreover, we observed reduced ISG expression after depletion of the mitochondrial pyrimidine nucleotide carrier SLC25A33, which mediates pyrimidine nucleotide uptake into mitochondria to support mtDNA replication[25], from *Mgme1⁻/⁻* cells (Fig. 3f). These results show that the disturbed nucleotide metabolism combined with altered mtDNA replication triggers mtDNA release and cGAS–STING–TBK1 signalling in *Mgme1⁻/⁻* cells.

## rNTP incorporation into mtDNA

Nucleotide metabolic profiling of *Mgme1⁻/⁻* cells revealed an increase in the ratio of ribonucleotide triphosphates (rNTP) to dNTP (Fig. 4a). Similarly, the loss of YME1L and inhibition of cytosolic thymidylate synthase with 5-fluorouracil (5-FU), which are both known to trigger mtDNA-dependent innate immune signalling[3], increase the rNTP/dNTP ratio (Fig. 4b and Extended Data Fig. 7a). rNTPs are present in excess relative to dNTPs in eukaryotic cells[26]. Although replicative polymerases are highly selective for dNTP, erroneous incorporation of rNTP poses a challenge to DNA replication and can lead to DNA strand breaks, genomic instability[27,28] and autoimmune disease[29,30]. Nuclear ribonucleotide excision repair involving RNase H2 limits the rNMP content of nuclear DNA[26]. However, mechanisms to remove ribonucleotides from mtDNA have not been identified, leaving mtDNA vulnerable to rNTP incorporation. Notably, increased dNTP levels in *Samhd1⁻/⁻* mice have been associated with decreased rNTP incorporation into mtDNA in various tissues[31]. Thus, we hypothesized that decreased mitochondrial dNTP, and the therefore increased rNTP/dNTP ratios in *Mgme1⁻/⁻* and *Yme1l⁻/⁻* cells, would lead to increased ribonucleotide insertion into mtDNA and impaired genomic integrity.

As MGME1 directly regulates mtDNA replication and maintenance, we used *Yme1l⁻/⁻* cells to monitor the metabolic effects of increased rNTP/dNTP ratios on mtDNA replication. Pulse labelling experiments with ³²P-dATP in isolated WT and *Yme1l⁻/⁻* mitochondria revealed a marked decrease in mtDNA synthesis in *Yme1l⁻/⁻* mitochondria in the absence of exogenously added dNTP (Extended Data Fig. 7b). Pulse–chase experiments using ³²P-dCTP in isolated WT and *Yme1l⁻/⁻* mitochondria showed no difference in mtDNA stability (Extended Data Fig. 7c). Furthermore, labelling of WT and *Yme1l⁻/⁻* cells with 5-ethynyl-2′-deoxyuridine (EdU), a thymidine analogue that is incorporated in both genomic and mtDNA, revealed reduced levels of newly synthesized mtDNA in *Yme1l⁻/⁻* cells (Extended Data Fig. 8a). Finally, *Yme1l⁻/⁻* cells showed a slower rate of mtDNA repopulation after mtDNA depletion using ddC (Extended Data Fig. 8b). We conclude from these experiments that the loss of YME1L reduces the rate of mtDNA synthesis. In vitro replication assays with model circular DNA molecules and recombinant proteins of the mitochondrial replisome showed the dependence of the mtDNA synthesis length and rate on the rNTP/dNTP ratio (Extended Data Fig. 8c,d). These results suggest that the increased rNTP/dNTP ratio decreases de novo synthesis of mtDNA in *Yme1l⁻/⁻* cells, which is consistent with an increased rNTP incorporation into mtDNA[32].

To directly quantify the rNMP content of mtDNA in WT and *Yme1l⁻/⁻* cells, we performed hydrolytic-end sequencing (HydEn-seq)[33]. We detected a relative increase in guanine ribonucleotides in the heavy strand of the mtDNA in *Yme1l⁻/⁻* cells compared with in WT cells (Fig. 4c). To substantiate these findings, we monitored rNTP incorporation into mtDNA using alkaline gel electrophoresis[34]. Alkaline conditions facilitate the hydrolysis of phosphodiester bonds in DNA strands at

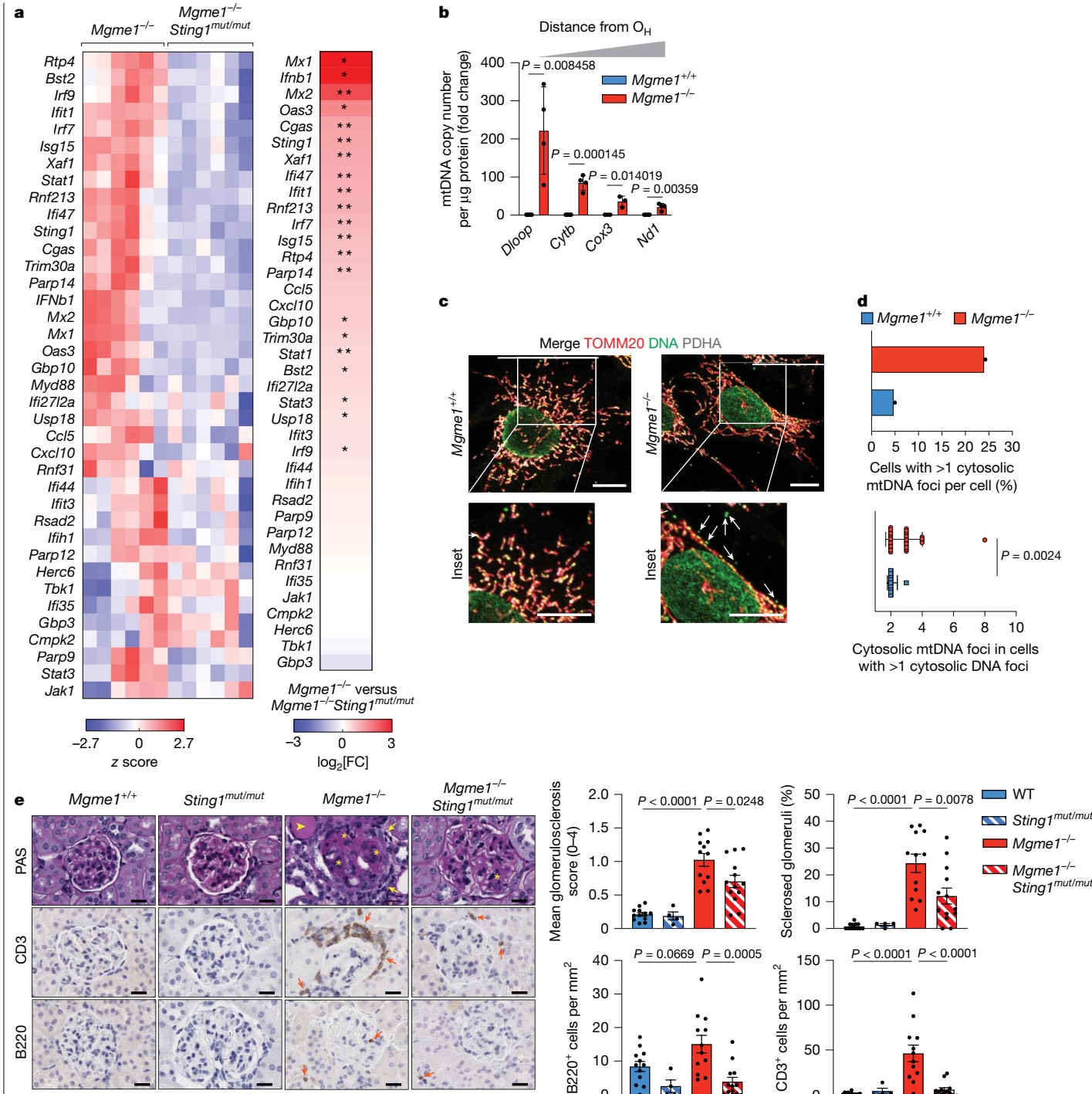

**Fig. 2 | STING-dependent inflammation after MGME1 loss. a**, Agglomerative heat map showing the *z*-score-normalized mRNA ISG intensities in kidneys of 55-week-old *Mgme1⁻/⁻* and *Mgme1⁻/⁻Sting1^{mut/mut}* mice obtained by NanoString profiling. Each column represents a different mouse (left). Heat map (log₂-transformed fold change, decreasing order) of mRNA ISG profiles obtained by NanoString profiling in kidneys of 55-week-old *Mgme1⁻/⁻* and *Mgme1⁻/⁻Sting1^{mut/mut}* mice (right). The asterisks mark ISGs of which the expression was significantly altered. *n* = 6 mice per genotype. **b**, dPCR data of mtDNA levels in the cytosolic fractions of immortalized *Mgme1⁺/⁺* and *Mgme1⁻/⁻* MEFs. The cytosolic mtDNA copy number was normalized to protein mass. *n* = 4 independent cultures. **c**, Localization of mtDNA foci determined by immunofluorescence analysis of immortalized MEFs. *Mgme1⁺/⁺* and *Mgme1⁻/⁻* cells were stained with antibodies against mitochondrial TOMM20 (green), pyruvate dehydrogenase (PDH) (grey) and double-stranded DNA (red). The arrows indicate non-mitochondrial mtDNA foci. Scale bars, 10 µm. **d**, Quantification of the immunofluorescence analysis shown in **c**. *n* = 150 cells. **e**, Representative images of the renal cortex sections stained with periodic acid–Schiff (PAS), and immunohistochemistry analysis (CD3 and B220 (CD45R)) of kidney sections from control (*Mgme1⁺/⁺*), *Mgme1⁻/⁻*, *Sting1^{mut/mut}* and *Mgme1⁻/⁻Sting1^{mut/mut}* mice. The yellow asterisks indicate regions of sclerosis within the glomerulus, the yellow arrows indicate immune cell infiltrates and the yellow arrowhead indicates proteinaceous casts. The orange arrows indicate CD3⁺ T cells and B220⁺ B cells (middle and bottom). Scale bars, 20 µm. Assessment of glomerulosclerosis, the percentages of sclerosed glomeruli and quantification of immune cell infiltration are shown on the right. *n* = 12 (*Mgme1⁺/⁺*), *n* = 4 (*Sting1^{mut/mut}*), *n* = 12 (*Mgme1⁻/⁻*) and *n* = 13 (*Mgme1⁻/⁻Sting1^{mut/mut}*) mice. *P* values were calculated using unpaired two-tailed Student *t*-tests (**b**) with Welch's correction (**d**) or using one-way analysis of variance (ANOVA) with Tukey's multiple-comparison test (**e**). Data are mean ± s.d. (**b** and **d**) and mean ± s.e.m. (**e**). **q* < 0.05, ***q* < 0.01 (**a**).

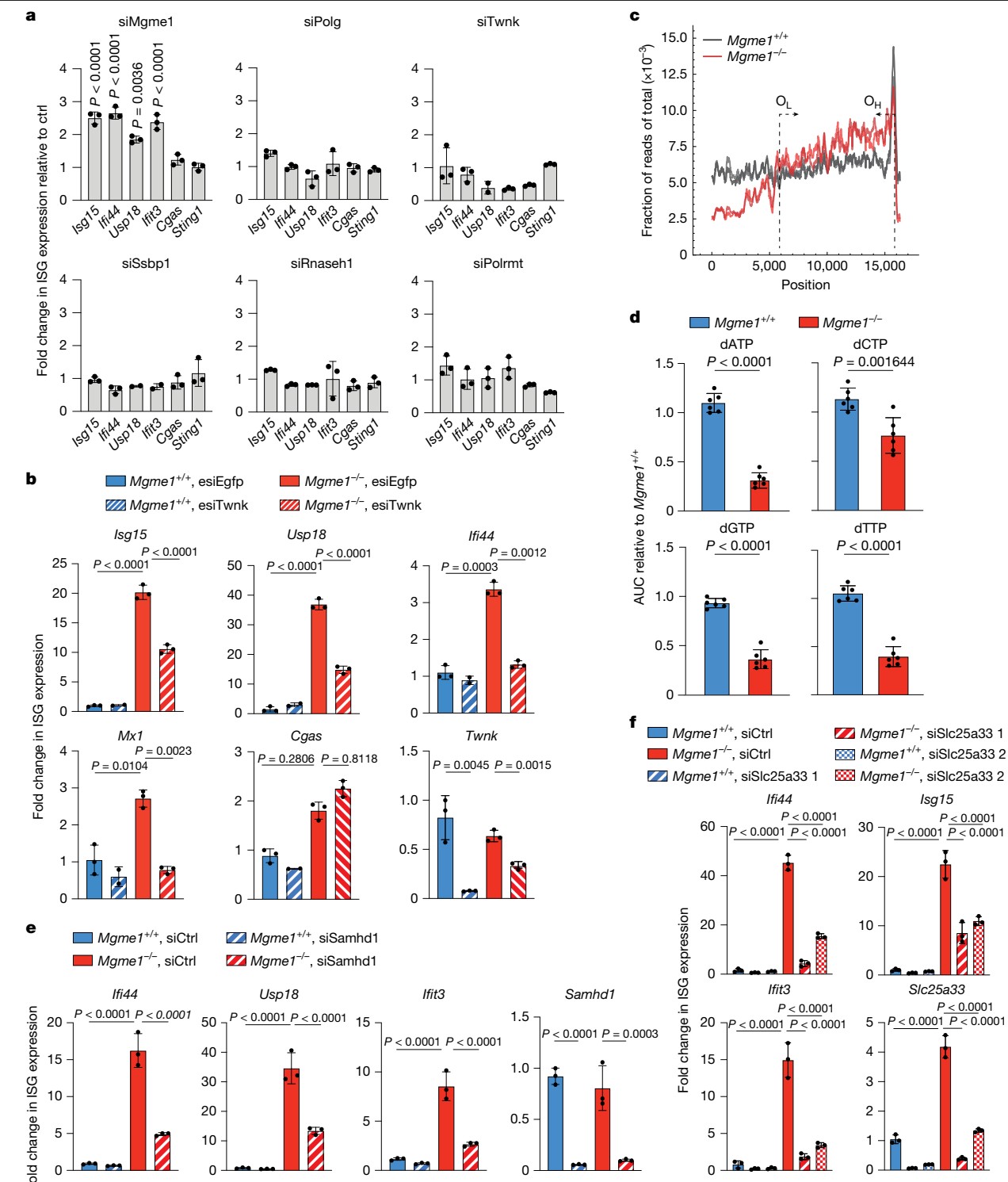

**Fig. 3 | Loss of MGME1 leads to depletion of dNTPs and mtDNA replication-dependent innate immune signalling. a**, ISG expression in immortalized MEFs after small interfering RNA (siRNA)-mediated downregulation of mtDNA-replication-related enzymes for 72 h. Fold changes in ISG expression are shown (relative to cells transfected with scrambled siRNA (ctrl)). $n = 3$ biologically independent experiments. **b**, ISG expression in *Mgme1*$^{+/+}$ and *Mgme1*$^{-/-}$ immortalized MEFs (relative to *Mgme1*$^{+/+}$) treated with the indicated endoribonuclease-prepared siRNA (esiRNA) for 72 h. $n = 3$ biologically independent experiments. **c**, Sequence coverage after next-generation sequencing of kidney mtDNA from *Mgme1*$^{-/-}$ and control (*Mgme1*$^{+/+}$) mice. Mitochondrial genome position ($x$ axis) versus sequence coverage divided by the total coverage for each sample. For each genotype, two samples derived

from different mice were analysed. The approximate locations of the origins of light-strand ($O_L$) and heavy-strand ($O_H$) replication are indicated by dotted lines with arrows. **d**, dNTP levels in cell lysates of *Mgme1*$^{+/+}$ and *Mgme1*$^{-/-}$ immortalized MEFs as measured by LC–MS. $n = 6$ independent cultures. Data are representative of three biologically independent experiments. Normalization was performed to input protein values. **e**,**f**, ISG expression in *Mgme1*$^{+/+}$ and *Mgme1*$^{-/-}$ immortalized MEFs treated with siRNA against *Samhd1* (**e**) and *Slc25a33* (**f**) as indicated for 72 h. $n = 3$ biologically independent experiments. Knockdown controls for **a** are shown in Extended Data Fig. 6a. The complete nucleotide landscape relevant to **d** is shown in Extended Data Fig. 6d. $P$ values were calculated using two-way ANOVA with the Tukey's multiple-comparison test (**a**,**b**,**e**,**f**) or two-tailed unpaired Student $t$-tests (**d**). Data are mean ± s.d.

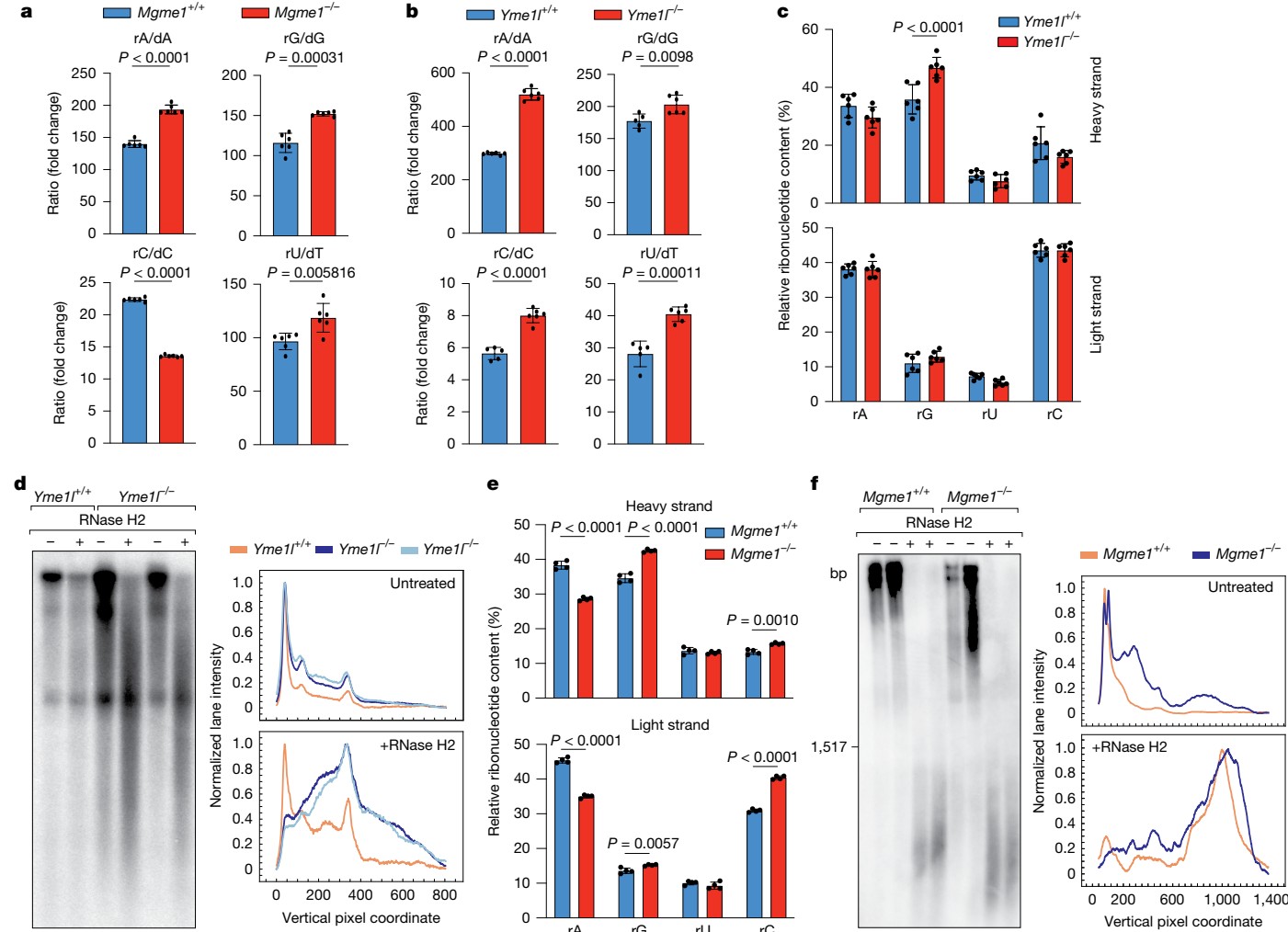

**Fig. 4 | Increased ribonucleotide incorporation into mtDNA in *Yme1l*⁻/⁻ MEFs and *Mgme1*⁻/⁻ kidneys. a,b,** LC–MS analysis of the ratio of rNTPs (rA, rG, rC, rU) and dNTPs (dA, dG, dC, dT) from cell extracts of *Mgme1*+/+ and *Mgme1*⁻/⁻ (**a**) or *Yme1l*+/+ and *Yme1l*⁻/⁻ (**b**) immortalized MEFs. *n* = 6 independent cultures. **c,** HydEn-seq analysis of mtDNA isolated from *Yme1l*+/+ and *Yme1l*⁻/⁻ immortalized MEFs. Each ribonucleotide is shown as the fraction of the total ribonucleotide content in the heavy strand (top) and light strand (bottom). *n* = 6 independent cultures. **d,** Alkaline agarose gel analysis of mtDNA isolated from *Yme1l*+/+ and *Yme1l*⁻/⁻ immortalized MEFs, treated with RNase H2 as indicated and analysed using Southern blotting. mtDNA was visualized using a *Cox1* probe (left). Southern blots were quantified and visualized using MultiGauge and Instant Clue software, respectively. The signal intensities of individual pixels were determined to generate plots for untreated (top) and RNase-H2-treated samples (bottom). The intensity data were scaled between 0 and 1. A representative plot of *n* = 3 biologically independent experiments is shown. **e,** HydEn-seq analysis of mtDNA isolated from *Mgme1*+/+ and *Mgme1*⁻/⁻ immortalized MEFs as described in **c**. *n* = 4 independent cultures. **f,** Untreated and RNase-H2-treated mtDNA isolated from control and *Mgme1*⁻/⁻ kidney tissue was resolved on alkaline agarose gels and analysed by Southern blotting using a *Cox1* probe (left). Densitometry plots of the Southern blots were normalized as described in **d** (right) and aggregated using the average normalized intensity of an individual replicate. A representative plot containing two biological replicates of *n* = 3 technical replicates is shown. *P* values were calculated using unpaired two-tailed Student *t*-tests (**a**,**b**) and two-way ANOVA with Šidák's multiple-comparison test (**c**,**e**). Data are mean ± s.d.

incorporated ribonucleotides, enabling DNA strand breaks by exogenously added RNase H2[34]. A different degradation pattern and the formation of shorter DNA fragments showed an increased alkaline sensitivity of mtDNA isolated from *Yme1l*⁻/⁻ cells, indicating increased rNTP incorporation (Fig. 4d). Similar to *Yme1l*⁻/⁻ cells, HydEn-seq analysis of *Mgme1*⁻/⁻ cells revealed significantly higher relative levels of guanine ribonucleotide in the heavy strand of mtDNA, as well as increased cytidine ribonucleotide insertion in the light strand and decreased adenosine ribonucleotide incorporation in both strands (Fig. 4e). We also observed an increased sensitivity of mtDNA isolated from kidneys of 55-week-old *Mgme1*⁻/⁻ mice to alkaline and RNase H2 treatment, indicating increased mtDNA rNMP content in vivo (Fig. 4f). Taken together, our results show that the increased rNTP/dNTP ratio in models of YME1L or MGME1 deficiency leads to higher mtDNA rNMP

content, which is associated with mtDNA release and innate immune signalling.

## mtDNA with rNTP in senescence and ageing

Similar to our findings in *Yme1l*⁻/⁻ and *Mgme1*⁻/⁻ cells, we also observed an increased rNMP content of mtDNA after treatment of WT cells with hydroxyurea (HU)—a potent inhibitor of cytosolic ribonucleotide reductase (RNR) and inducer of mtDNA-dependent innate immune signalling[3] (Extended Data Fig. 8e). RNR converts rNTP to dNTP and consists of two non-homologous subunits, termed RRM1 and RRM2[35,36]. RRM2 expression is cell cycle dependent and is halted in cell-cycle-arrested senescent cells, resulting in reduced dNTP levels[37]. As mtDNA release contributes to the pro-inflammatory SASP of senescent cells[15] and as

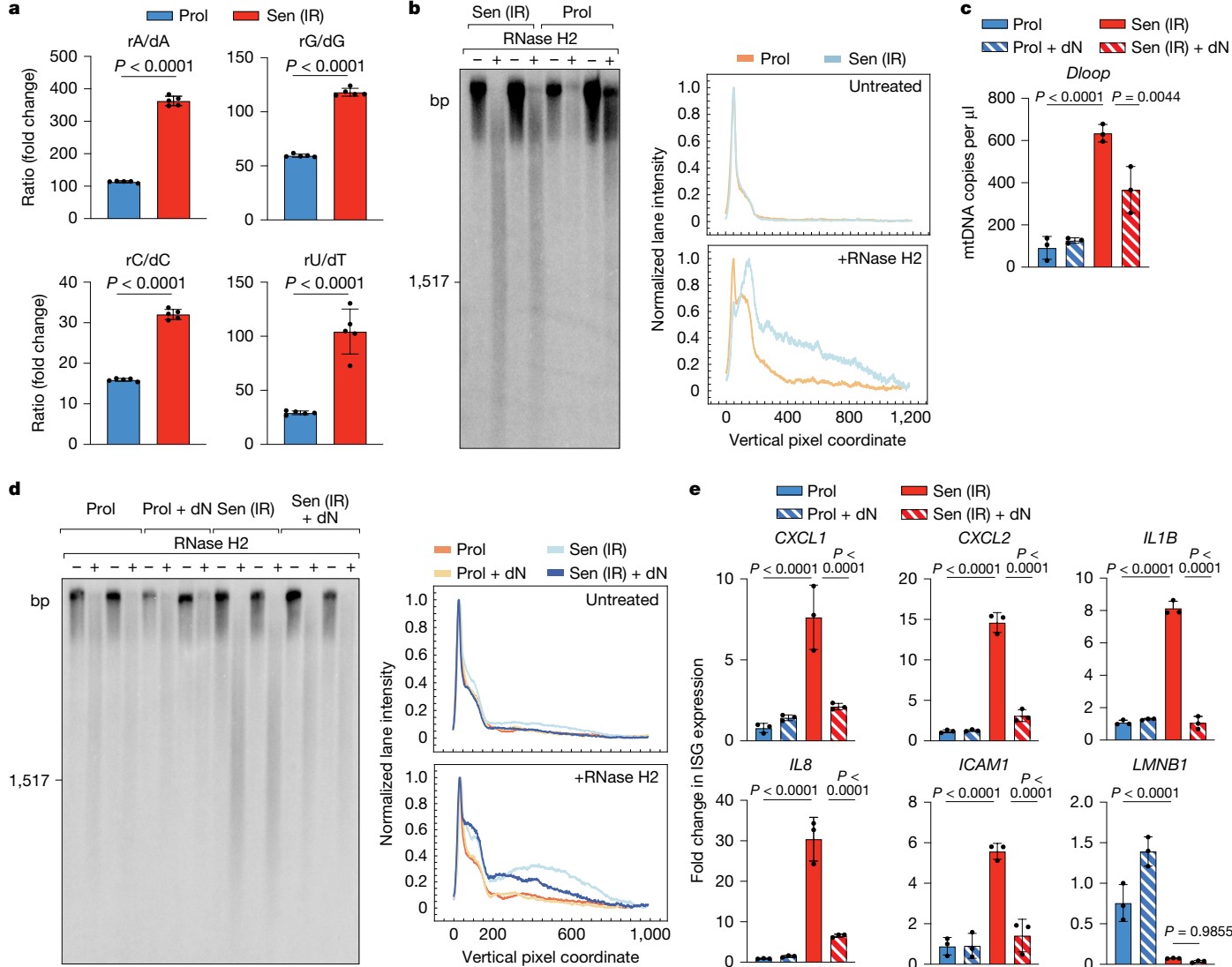

**Fig. 5 | Increased ribonucleotide incorporation into mtDNA from senescent cells. a**, LC–MS analysis of the ratios of ribonucleotide and deoxyribonucleotide triphosphates from cell extracts of proliferating (Prol) and irradiation-induced senescent (Sen (IR)) IMR90 cells. $n = 5$ independent cultures. **b**, Untreated and RNase-H2-treated DNA isolated from proliferating and irradiation-induced senescent IMR90 cells was resolved on alkaline agarose gels and further analysed by Southern blotting using a *Cox1* probe (left). Southern blots were analysed as described in Fig. 4d. A representative plot of $n = 3$ biologically independent experiments is shown. **c**, dPCR analysis of mtDNA levels in the cytosolic fractions of proliferating and irradiation-induced senescent IMR90 cells with and without deoxyribonucleoside supplementation. A probe/primer set for the D-loop region was used. $n = 3$ biologically independent experiments.

**d**, Untreated and RNase-H2-treated mtDNA isolated from proliferating and irradiation-induced senescent IMR90 cells, which were supplemented with all four deoxyribonucleosides where indicated, was resolved on alkaline agarose gels and further analysed by Southern blotting and quantified as described in Fig. 4d. **e**, SASP gene expression was determined using quantitative PCR with reverse transcription (RT–qPCR) in proliferating and irradiation-induced senescent IMR90 cells, which were supplemented with deoxyribonucleosides where indicated. $n = 3$ biologically independent experiments. $P$ values were calculated using unpaired two-tailed Student $t$-tests (**a**), one-way ANOVA with Tukey's multiple-comparison test (**c**) or two-way ANOVA with Tukey's multiple-comparison test (**e**). Data are mean ± s.d.

mtDNA replication is cell cycle independent[38] (Extended Data Fig. 9a), we reasoned that altered rNTP/dNTP ratios and increased rNTP incorporation into mtDNA could trigger mtDNA release and SASP in senescent cells.

To induce cellular senescence, we either exposed human lung IMR90 fibroblasts to ionizing radiation (IR) or treated these cells with the nucleoside analogue decitabine or the topoisomerase II inhibitor doxorubicin for therapy-induced senescence (TIS)[15,39]. We confirmed the establishment of the senescent state by monitoring senescence-associated β-galactosidase (SA-β-gal) activity and the changes in senescence-associated protein markers (Extended Data Fig. 9b,c). mtDNA levels were moderately increased in senescent cells after IR (Extended Data Fig. 9d). Consistent with the loss of dNTP during

cellular senescence, metabolomic analysis revealed increased rNTP/dNTP ratios after IR or TIS (Fig. 5a and Extended Data Fig. 9e). Analysis of mtDNA in senescent cells after IR treatment by alkaline gel electrophoresis showed an increased sensitivity to alkaline and RNase H2 treatment (Fig. 5b), indicating a higher rNMP content of the mtDNA of these cells. We made similar observations in senescent IMR90 fibroblasts after decitabine or doxorubicin treatment and in irradiated kidney proximal tubule epithelial cells (Extended Data Fig. 9f,g). Thus, decreased dNTP levels are associated with an increased rNTP incorporation into mtDNA in senescent cells.

To test whether the increased rNMP content of mtDNA affects mtDNA release and SASP, we treated senescent cells with deoxyribonucleosides. Deoxyribonucleoside supplementation did not affect

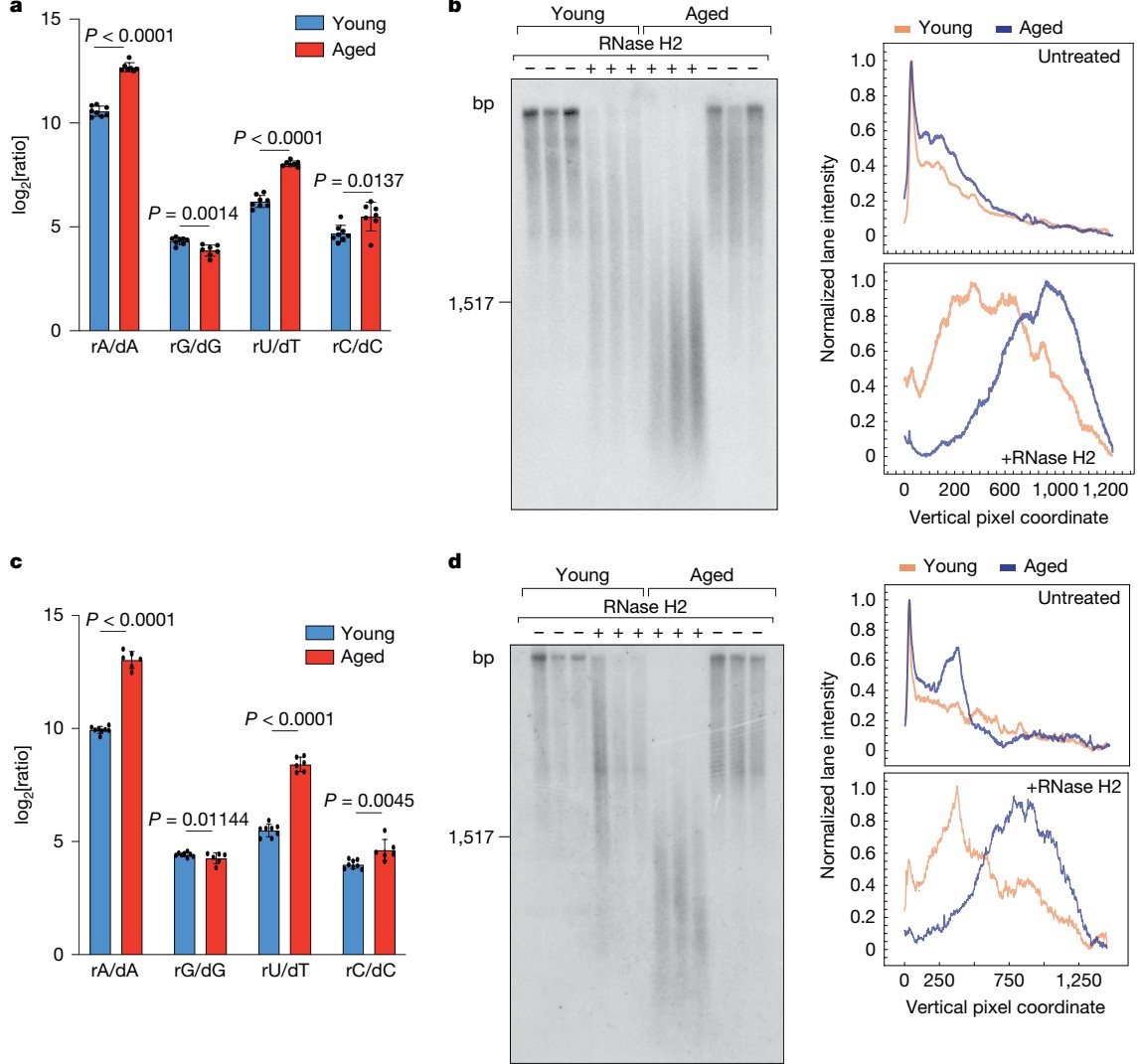

**Fig. 6 | Increased ribonucleotide content of mtDNA in aged kidney and liver tissues. a**, LC–MS analysis of the ratios (log₂-transformed fold change) of rNTP and dNTP levels from kidney lysates of young (aged 1 week; $n$ = 8 mice) and old (aged 80–96 weeks; $n$ = 7 mice) mice. **b**, Untreated and RNase-H2-treated DNA isolated from kidney of young (aged 1 week) and old (aged 87 weeks) mice was analysed by alkaline agarose gels and Southern blotting. Quantification of Southern blots was performed as described in Fig. 4d. **c**, LC–MS analysis of the ratios (log₂-transformed fold change) of rNTP and dNTP levels from liver lysates of young (aged 1 week; $n$ = 8 animals) and old (aged 80–96 weeks; $n$ = 6 animals) mice. **d**, The same as **b**, except that liver tissue was analysed. $P$ values were calculated using unpaired two-tailed Student $t$-tests (**a** and **c**). Data are mean ± s.d.

the senescent state per se as assessed by monitoring SA-β-gal activity (Extended Data Fig. 9h). However, it significantly reduced cytosolic mtDNA (Fig. 5c and Extended Data Fig. 9i) and the mtDNA sensitivity for alkali and RNase H2 treatment, indicating a lower rNMP content (Fig. 5c,d). Reduced rNTP incorporation after dNTP complementation of senescent cells suppressed the expression of inflammatory SASP genes (Fig. 5e and Extended Data Fig. 9j). We therefore conclude that SASP gene expression depends on the increased rNTP incorporation into the mtDNA of senescent cells.

As senescent cells accumulate with age and higher rNMP content of mtDNA has been observed during skeletal muscle and heart development[31], we further investigated whether rNTP incorporation into mtDNA occurs during natural ageing. Metabolomic analysis of kidney, liver, heart and spleen from aged mice revealed an increased rNTP/dNTP ratio in all tissues, with diverse effects on different nucleotides (Fig. 6a,c and Extended Data Fig. 10a,c). mtDNA isolated from these tissues showed an increased sensitivity to alkaline and RNase H2 treatment compared with mtDNA isolated from tissues from young mice (Fig. 6b,d and Extended Data Fig. 10b,d), demonstrating an increased rNTP incorporation in aged tissues.

## Discussion

We demonstrate that increased incorporation of rNTPs into mtDNA during replication leads to the release of mtDNA fragments from mitochondria and proinflammatory signalling. Our results therefore highlight the challenge that the high molar excess of rNTPs relative to dNTPs poses to cells. Although RNase H2 removes incorporated rNMPs from nuclear DNA as part of the ribonucleotide excision repair pathway, this repair mechanism is not present in mitochondria, which are therefore prone to accumulating rNMPs in their genome. Similar to the effect of rNMPs on nuclear DNA replication[27,40], due to the inherent reactivity of the 2′-OH group of the ribose ring or collisions with the replication fork, misincorporated rNMPs may cause DNA strand breaks during replication, priming the release of mtDNA fragments from mitochondria.

The deleterious effect of rNMPs incorporated into nuclear DNA is illustrated by cGAS–STING-driven inflammatory pathologies caused by mutations of RNase H2 subunits, leading to Aicardi–Goutières syndrome and systemic lupus erythematosus[28,30,41]. Accordingly, our results suggest that an increased rNMP content in mtDNA and mtDNA

damage contribute to immune activation in mitochondrial disorders. Mice lacking MGME1 exhibit age-dependent renal inflammation, which is explained by perturbations in cellular nucleotide metabolism, leading to misincorporation of rNTPs into mtDNA, mtDNA release into the cytosol and cGAS–STING signalling. Suppression of immune signalling improves renal pathology, demonstrating the contribution of inflammation to disease progression. Similarly, disturbances in the dNTP pools and increased rNTP incorporation into mtDNA may explain inflammatory responses and contribute to renal failure in a mtDNA-depletion syndrome caused by mutations in *RRM2B*[42] or other mitochondrial disorders affecting the nucleotide metabolism[43,44].

The increased rNMP content of mtDNA in aged tissues of WT mice provides further support that mtDNA contributes to inflammatory responses during natural ageing[15,16]. Indeed, the vulnerability of mtDNA to ribonucleotide incorporation becomes apparent in the senescent state, which is characterized by a cell cycle arrest, reduced RNR activity and decreased dNTP levels. Whereas nuclear DNA replication is halted, mtDNA replication continues in senescent cells. We show that the increased rNTP/dNTP ratio in senescent cells leads to increased ribonucleotide incorporation into mtDNA, mtDNA-driven cGAS–STING signalling and SASP, highlighting the potentially far-reaching effects of this mitochondria-dependent inflammatory mechanism in ageing, neurodegenerative diseases and cancer.

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

# Methods

## Cell culture, transfection and RNA interference

MEFs and HeLa cells were maintained in DMEM-GlutaMAX (Thermo Fisher Scientific) containing $4.5 \, \text{g} \, \text{l}^{-1}$ of glucose supplemented with 1 mM sodium pyruvate (Gibco), 100 µM non-essential amino acids (Gibco) and 10% FBS (Biochrom). IMR90 cells were maintained in MEM-GlutaMAX (Thermo Fisher Scientific) and 10% FBS. MEF and HeLa cell lines were maintained at 37 °C under 5% $CO_2$ and ambient oxygen. IMR90 cells were maintained at 37 °C under 5% $CO_2$ and 3% $O_2$. Cell lines were routinely tested for *Mycoplasma* infection. Cell numbers were monitored by Trypan Blue exclusion and cell counting using the Countess automated cell counter (Thermo Fisher Scientific). Cells were seeded at equal densities and grown to confluency over a period of 72 h without medium changes unless stated otherwise. Opti-MEM + GlutaMAX (Gibco) and lipofectamine RNAiMax (Invitrogen) were used for reverse transfection of endoribonuclease-prepared short interfering RNA (esiRNA) and short interfering RNA (siRNA) for 72 h. A list of the esiRNAs and siRNAs used in this study is provided in Supplementary Table 1. Where indicated, the following compounds were added to the medium: ddC (Sigma-Aldrich), BX795 (Sigma-Aldrich), VACV-70 70 bp oligonucleotides (InvivoGen) and tetracycline (Sigma-Aldrich). For deoxyribonucleoside supplementation during senescence, deoxyadenosine, deoxyguanosine, deoxycytosine and thymidine (Sigma-Aldrich) were added at 100 µM each to cultured cells and refreshed every 72 h.

## Generation of KO and stable cell lines

Immortalized WT and *Yme1l*[−/−] MEFs were described previously[17]. In brief, primary MEFs were isolated from *Yme1l*[loxP/loxP] mice and immortalized using SV40 large T antigen-encoding plasmids. After transduction with Cre recombinase, individual clones were expanded and genotyped to identify WT (*Yme1l*[loxP/loxP]) and *Yme1l*[−/−] MEFs. WT, *Mgme1*[−/−], *Sting1*[mut/mut] and *Mgme1*[−/−]*Sting1*[mut/mut] (DKO) primary MEFs were isolated from intercrossing of *Mgme1*[+/−]*Sting1*[mut/mut] mice and intercrossing of *Mgme1*[+/−]*Sting1*[+/+] mice.

## Establishment of cellular senescence

IMR90 fibroblasts were seeded on a diverse size of culture vessels with a density of 2,100 cells per cm² for DMSO (0.01%, v/v) and treated with decitabine (1 µM), or 6,500 cells per cm² for doxorubicin (300 nM) for TIS. The medium was replaced every other day. DMSO and decitabine were present in the medium at all times, whereas doxorubicin was washed out after the first medium change. The experiments were performed 10 days after the irradiation. Stress-induced senescence was achieved by exposing cells to X-ray irradiation at 20 Gy. Senescence was confirmed by the presence of p16 and p21, SA-β-Gal positivity (Cell Signaling, 9680, according to the manufacturer's instructions) and the absence of genomic EdU incorporation (Thermo Fisher Scientific, C10337, according to the manufacturer's instructions).

## RT–qPCR

Cells were plated $1 \times 10^5$ cells per ml in six-well plates and incubated for 72 h. Cells were scraped in cold PBS and pelleted. Total RNA from cells was isolated upon cell lysis using the NucleoSpin RNA isolation kit. cDNA was synthesized using the GoScript Reverse Transcription Mix (Promega) and qPCR was performed using PowerSYBR Green PCR Master Mix (Applied Biosystems). For each independent sample, qPCR was performed in technical duplicates. A list of the primer sequences used in this study is provided in Supplementary Table 2.

## Animals and housing

This study was performed in accordance with the recommendations and guidelines of the Federation of European Laboratory Animal Science Associations (FELASA). The protocol was approved by the 'Landesamt für Natur, Umwelt und Verbraucherschutz Nordrhein-Westfalen'

(reference numbers 81-02.04.2019.A378, 81-02.04.2020.A082, 81-02.04.2022.A453 and 2024-657). The mice were housed in standard individually ventilated cages (45 × 29 × 12 cm) under a 12 h–12 h light–dark schedule in controlled environmental conditions of $22 \pm 2$ °C and 50 + 10% relative humidity and fed a normal chow diet and water ad libitum. Generation of *Mgme1*[−/−] mice was described previously[4]. *Mgme1*[−/−]*Sting1*[mut/mut] mice were generated by intercrossing those mice with *Sting1*[mut/mut] (*C57BL/6J-Sting1*[gt]/J, Jackson laboratory).

## mtDNA extraction and Southern blot analysis

Total DNA and mtDNA was isolated from pulverized tissue/cellular pellets or purified mitochondria using Gentra Puregene Tissue Kit (Qiagen) or DNeasy Blood & Tissue Kit (Qiagen) according to the manufacturer's instructions. DNA quantification was performed using the Qubit 1.0 fluorometer (Thermo Fisher Scientific). Then, 300–1,000 ng of DNA was digested with SacI (mouse) or BamH1 (human) restriction nucleases and DNA fragments were separated by neutral agarose gel electrophoresis, transferred to nitrocellulose membranes (Hybond-N+ membranes, GE Healthcare) and hybridized with $\alpha$-P[32]-dCTP-labelled probes). For estimation of rNMP content, DNA was treated with RNase H2 and separated by electrophoresis on 0.8% agarose gels under denaturing conditions. Southern blot signals were quantified using Multi-Gauge or ImageJ software and visualization and analysis was performed using InstantClue software[45]. Primer and probe sequences used in this study can be found in Supplementary Table 3.

## In organello replication

Freshly isolated mitochondria (300–500 µg) from MEFs were resuspended in 0.5 ml of incubation buffer (25 mM sucrose, 75 mM sorbitol, 100 mM KCl, 10 mM $K_2HPO_4$, 0.05 mM EDTA, 5 mM $MgCl_2$, 1 mM ADP, 10 mM glutamate, 2.5 mM malate, 10 mM Tris–HCl, pH 7.4, 1 mg ml$^{-1}$ fatty-acid-free BSA, 20 µCi $\alpha$-$^{32}$P-dATP (3,000 Ci mmol$^{-1}$). Then, 50 µM each of dTTP, dCTP and dGTP was added in the incubation buffer where indicated. Incubation was carried out at 37 °C for 2 h on a rotating wheel. For the chase, reisolated mitochondria were incubated in 0.5 ml of incubation buffer supplemented with all four non-radiolabelled dNTPs (50 µM) for the indicated time. After incubation, mitochondria were pelleted at 9,000 rpm for 4 min and washed twice with wash buffer (10% glycerol, 10 mM Tris–HCl, pH 6.8, 0.15 mM $MgCl_2$). In the next step, DNA isolation and Southern blot analysis were performed as described above.

## Detection of mtDNA in the cytosolic fraction using ddPCR or dPCR

For isolation of the cytosolic fraction, $2 \times 10^5$ MEF cells were collected at 600*g* for 3 min and resuspended in 400 µl of isolation buffer (150 mM NaCl and 50 mM HEPES). After treating the cells with 25 µg ml$^{-1}$ digitonin in isolation buffer for 12 min on ice, the samples were centrifuged at 800*g* for 5 min at 4 °C. The supernatants were collected and further clarified by centrifugation at 25,300*g* for 10 min at 4 °C to obtain the cytosolic fraction[10]. The samples were normalized for protein concentration (BioRad). DNA was extracted from equal volumes of cytosolic fractions using the DNeasy DNA extraction kit (Qiagen) according to the manufacturer's instructions. The mtDNA copy number was assessed using ddPCR (BioRad) using specific probes directed against different mtDNA regions. A list of the primer and probe sequences used in this study is provided in Supplementary Table 4.

The isolation of cytosolic and mitochondrial fractions from mouse tissues was performed as previously described[46]. All of the samples were normalized to protein concentration using a DC protein assay (BioRad) and dPCR (Qiagen QiaCutie) was performed using the primer probe sets listed in Supplementary Table 4.

## Nanostring analysis

Total RNA was isolated from tissues or cells using the NucleoSpin RNA isolation kit. The quality and quantity were validated using Nanodrop

Analysis. Next, 200–500 ng of total RNA was applied for CodeSet hybridization. After 18 h of hybridization, the samples were loaded onto the Nanosting Cartridge and analysed in the nCounter SPRINT Profiler (nanoString) system according to the manufacturer's instructions. ISGs and non-ISGs were filtered using the INTERFEROME database[47]. The acquired data were analysed with nSolver software (nanoString). $z$-score intensities were calculated using the $\log_2$-transformed intensities and clustering was performed using the Euclidean distance metric and the 'average' method. $q$ values were calculated using unpaired two-tailed Student $t$-tests with multiple-comparison correction using the two-stage step-up method of Benjamini, Krieger and Yekutieli. $q < 0.05$ was considered to be significant.

## SDS–PAGE and immunoblot analysis

Cells were washed with cold PBS and resuspended in ice-cold RIPA buffer (50 mM Tris–HCl, pH 7.4, 150 mM NaCl, 1% Triton X-100, 0.1% SDS, 0.05% sodium deoxycholate, 1 mM EDTA) containing protease inhibitor cocktail (Roche) and phosphatase inhibitor cocktail (PhosSTOP, Roche). The lysates were incubated with constant agitation for 30 min at 4 °C followed by centrifugation at 14,500$g$ for 10 min. Total protein (20–50 µg) was separated using SDS–PAGE, followed by transfer to nitrocellulose membranes and immunoblotting with the antibodies listed in Supplementary Table 5.

## Kidney histology

PAS and immunohistochemistry staining procedures for formalin-fixed paraffin-embedded tissue samples were essentially performed as previously described[48]. CD3 (99940, Cell Signaling, 1:100) and B220 (CD45R, 550286, BD Pharmingen, 1:50) antibodies were used for immunohistochemistry DAB staining. In brief, 2 µm formalin-fixed paraffin-embedded sections of kidney tissue were prepared, stained, digitalized using the Ventana DP 200 slide scanner (Roche Diagnostics) and analysed using QuPath (v.0.4.4)[49]. Finally, whole-kidney sections were analysed (staining artefacts such as tissue detachment were manually excluded). Cells were segmented using the built-in cell segmentation tool of QuPath and marker-positive cells were detected by thresholding. The assessment of glomerular sclerosis was conducted using a five-tier scoring system, with values of 0 representing normal glomeruli; 1 indicating mesangial dilatation or capillary wall thickening; 2 denoting segmental sclerosis in less than 25% of the glomerular tuft area; 3 indicating 25–50% of tuft area; and 4 indicating more than 50% of tuft area sclerosed. The mean score (0 to 4) and the percentage of sclerosed glomeruli (glomeruli with scores ≥ 2) were calculated.

## Immunofluorescence and imaging

Cells were plated at a density of $4 \times 10^4$ cells per coverslip. After incubation for 24 h, the medium was removed and cells were fixed on coverslips using 5% PFA/PBS (pH 7.4) for 15 min at 37 °C. After three quick PBS washes, autofluorescence was quenched using 50 mM NH₄Cl/PBS for 10 min at room temperature with slowly shaking. After two quick rinses with PBS, cells were permeabilized using 0.1% Triton X-100 in PBS for 15 min at room temperature and washed with PBS again. After blocking the coverslips for 20 min at room temperature using 10% FBS/PBS, the coverslips were incubated with primary antibodies (Supplementary Table 5) for 1 h or overnight and washed three times with PBS before incubation with secondary antibodies for 1 h in the dark. After incubation, cells were washed with PBS, dipped into double-distilled $H_2O$ to remove salt residues and mounted using ProLong Gold (Invitrogen). Images were acquired using the Leica SP8-DLS laser-scanning confocal microscope equipped with an ×100 oil HC PL APO CS2 objective (numerical aperture (NA) 1.4), Leica DMI 6000 B wide-field fluorescence microscope equipped with a 100× oil HCX PL APO objective (NA 1.46) or Invitrogen EVOS FL Auto 2 microscope equipped with colour camera (to visualize positive X-Gal staining). Immunofluorescence images shown in Fig. 2d were acquired with a spinning-disk microscope. Stained cells were imaged using a 100× objective lenses (NA 1.4) on a Nikon Eclipse TiE inverted microscope with a four-channel integrated laser engine (ILE-400) using an Andor Dragonfly 500 spinning-disk system, equipped with a Zyla 4.2 PLUS sCMOS camera (Andor), coupled with Fusion software (Andor). Seven stacks of 0.2 µm each were acquired using the 100× objective. All images from a same experiment were acquired with the same parameters including exposure time and laser intensities. Images were compiled using 'max projection', and 'smooth' was applied once for representative images using the FIJI software.

## HydEn-seq analysis

HydEn-seq and DNA libraries for sequencing were prepared as previously described[43]. In brief, WT, $Yme1l^{-/-}$ and $Mgme1^{-/-}$ MEF cells were mapped by HydEn-seq by hydrolysing 1 µg DNA with 0.3 M KOH for 2 h at 55 °C. Then, 1 µg of DNA was digested or left untreated with 10 U of SacI ($Yme1l^{-/-}$ and its WT control) or 10 U of HincII ($Mgme1^{-/-}$ and its WT control), and the digests were purified with HighPrep PCR beads (Mag-Bio) before the KOH treatment. The libraries were then purified, quantified using the Qubit fluorometric instrument (Thermo Fisher Scientific) and 75-bp-end sequenced on an Illumina NextSeq 500 instrument to identify the location of the free 5′ ends. Breakpoints arising from restriction digestion were removed during data processing. Digested and undigested samples, which showed the same trend, were analysed together as independent replicates.

## Library preparation and Illumina paired-end DNA sequencing

Library preparation and Illumina paired-end DNA sequencing were performed by the Cologne Center for Genomics (https://ccg.uni-koeln.de/technologies/next-generation-sequencing). mtDNA quality was verified using the Agilent TapeStation Genomic DNA ScreenTape and the Qubit BR kit. After fragmentation of 500 ng of gDNA to the desired insert size of >500 bp, the libraries were produced using the Illumina DNA PCR-Free Library Prep kit. After sequencing, the following run conditions were used (2 × 300 bp) on the Illumina NextSeq 2000 sequencing system, using the 1x NextSeq 2000, P1 FlowCell from Illumina. Each library was individually indexed with the barcodes provided and sequenced to 100 million reads. Reads were mapped to the mouse MT genome (GRCm39) with bwa (v.0.7.17; mem-T 19).

## Protein digestion

**Subcellular fraction proteomics.** Mitochondria and total (cell pellet) were directly lysed in 4% SDS in 100 mM HEPES buffer (pH 8.5) at 70 °C for 10 min. Cytosolic fractions were subjected to acetone-based protein precipitation. The pellet was washed twice with 80% acetone and dried under the fume hood. The lysates were sonicated (Bandelin, Mini, 60 s, 1 s pulse, 1 s pause). Proteins were reduced (10 mM TCEP) and alkylated (20 mM CAA) in the dark for 45 min at 45 °C. The samples were subjected to an SP3-based digestion. Washed SP3 beads (SP3 beads (Sera-Mag), magnetic carboxylate modified particles (hydrophobic, GE44152105050250), Sera-Mag magnetic carboxylate modified particles (Hydrophilic, GE24152105050250) from Sigma-Aldrich) were mixed equally, and 3 µl of bead slurry was added to each sample. Acetonitrile was added to a final concentration of 50% and washed twice using 70% ethanol ($V = 200$ µl) on an in-house-manufactured magnet. After an additional acetonitrile wash ($V = 200$ µl), 5 µl digestion solution (10 mM HEPES pH 8.5) containing 0.5 µg trypsin (Sigma-Aldrich) and 0.5 µg LysC (Wako) was added to each sample and incubated overnight at 37 °C. Peptides were desalted on a magnet using 2 × 200 µl acetonitrile. Peptides were eluted in 10 µl 5% DMSO in LC–MS water (Sigma-Aldrich) in an ultrasonic bath for 10 min. Formic acid and acetonitrile were added to a final concentration of 2.5% and 2%, respectively. The samples were stored at −20 °C before analysis using LC–MS/MS.

## LC and tandem MS-based proteomics

**Subcellular fraction proteomics.** Tryptic peptides (200 ng) were loaded onto an Evotip and analysed using the Evosep One LC (Evosep) system connected to a timsTOF Pro 2 (Bruker). The Evosep One method was 30 SPD (44 min gradient) using the EV1137 performance column (15 cm × 150 μm, 1.5 μm particle size) at 50 °C using a 20 μm Bruker emitter. The mass spectrometer was operated in DIA-PASEF mode. The DIA-PASEF method consisted of 24 $m/z$ windows leading to a cycle time of 1.38 s. The other mass spectrometer parameters were set as follows: DIA $m/z$ range, 480 to 1,000, the mobility ($1/K_0$) range was set to 0.895 to 1.21 V s cm$^{-2}$, and the accumulation and ramp time was 100 ms.

## Data analysis

Raw files were analysed using Spectronaut (v.18.5.231110.55695)[50] in direct DIA mode using the UniProt (one sequence per gene) *Mus musculus* reference proteome (21,000 sequences, downloaded 2021). Trypsin/P was selected as the cleavage rule using a specific digest type. The minimal peptide length was set to 7 and a total of 2 missed cleavages were allowed. The peptide spectrum match, peptide and protein group false-discovery rate were controlled to 0.01. The mass tolerances were used with the default settings (Dynamic,1). The directDIA+ (deep) workflow was selected. The cross-run normalization was disabled. Peptide and protein group quantity was accessed by the sum of the precursor and peptide quantity, respectively. Mitochondrial proteins were identified using the MitoCarta 3.0 dataset[51]. The intensities were log$_2$-transformed and the median across the subcellular fractions was adjusted to the median of all samples of the respective fraction. The log$_2$-transformed intensities were scaled between 0 and 1 per protein group and visualized in a heat map using Instant Clue[45] by clustering the rows using the Euclidean distance metric and the 'average' method.

## AEX–MS analysis of anionic metabolites

Extracted metabolites were resuspended in 100 μl of Optima LC–MS-grade water (Thermo Fisher Scientific), and transferred to polypropylene autosampler vials (Chromatography Accessories) before anion-exchange chromatography–MS (AEX–MS) analysis.

The samples were analysed using a Dionex ion chromatography system (ICS500+, Thermo Fisher Scientific) as described previously[52]. In brief, 5 μl of polar metabolite extract was injected in push partial mode, using an overfill factor of 2, onto the Dionex IonPac AS11-HC column (2 mm × 250 mm, 4 μm particle size, Thermo Fisher Scientific) equipped with a Dionex IonPac AG11-HC guard column (2 mm × 50 mm, 4 μm, Thermo Fisher Scientific). The column temperature was held at 30 °C, and the auto sampler was set to 6 °C. A potassium hydroxide gradient was generated using a potassium hydroxide cartridge (Eluent Generator, Thermo Fisher Scientific), which was supplied with deionized water. The metabolite separation was carried at a flow rate of 380 μl min$^{-1}$, applying the following gradient conditions: 0–3 min, 10 mM KOH; 3–12 min, 10 – 50 mM KOH; 12–19 min, 50–100 mM KOH; 19–21 min, 100 mM KOH; 21–22 min, 100–10 mM KOH. The column was re-equilibrated at 10 mM for 8 min.

For the MS analysis of metabolic pool sizes and the corresponding deoxy-/ribo- nucleotide ratios, the eluting compounds were detected in negative ion mode [M-H]$^-$ using the multiple reaction monitoring mode on the Xevo TQS triple quadrupole mass spectrometer (Waters). The following settings were set: capillary voltage, 2.1 kV; desolvation temperature, 500 °C; desolvation gas flow, 800 l h$^{-1}$; collision cell gas flow, 0.15 ml min$^{-1}$. The detailed quantitative and qualitative transitions and retention times for the analysed nucleotides are summarized in Supplementary Table 6.

The obtained AEX–MS data analysis was performed by converting the raw data into mzML files using the MSConvert GUI from the ProteoWizzard suite[53]. The obtained mzML files were analysed using the metabolomics software ElMaven[54]. For relative quantification analysis, the area of the quantitative transition of each compound was extracted and integrated using a retention time tolerance of <0.1 min as compared to the independently measured reference compounds. Areas of the cellular pool sizes were normalized to the internal standards ($^{13}C_{10}$ ATP, $^{15}N_5$ ADP or $^{13}C_{10}$$^5N_5$ AMP), followed by a normalization to the protein content of the analysed cell sample.

## Protein expression and purification

POLγA, POLγB and the TWINKLE DNA helicase were cloned and expressed as 6×His-tagged fusion proteins in *Spodoptera frugiperda* (Sf9) cells, according to previously published protocols[55]. The *SSBP1* gene encoding the human mitochondrial single-stranded DNA-binding protein (mtSSB) was cloned into the pET-17b vector in frame with a C-terminal 6×His-tag. The mtSSB protein was expressed in *Escherichia coli* and subsequently purified as previously reported[56].

## In vitro DNA rolling circle replication

A 70-mer oligonucleotide (5'-T$_{42}$ATCTCAGCGATCTGTCTATTTCGTTCAT-3') was annealed to single-stranded pBluescript SK(+) O$_L$[55], followed by one cycle of polymerization with KOD polymerase to produce a 4 kb double-stranded template with a preformed replication fork. The template (0.4 nM) was added to a reaction mixture (25 μl) containing 25 mM HEPES (pH 7.6), 10 mM DTT, 0.1 mg ml$^{-1}$ BSA, 4 mM ATP, all four dNTPs (2.5 μM or 10 μM, as indicated), 2 μCi α-[$^{32}$P]dCTP, 8 nM TWINKLE (calculated as a hexamer), 160 nM mtSSB (calculated as a tetramer) and 4 nM POLγA in complex with 6 nM POLγB (calculated as a dimer). GTP was added at increasing concentrations. In the 0 mM GTP control reaction, 10 mM MgCl$_2$ was included; for the other conditions, the MgCl$_2$ concentration was increased proportionally to match the GTP levels. The reactions were terminated after 60 min at 37 °C by the addition of 8 μl of alkaline loading buffer (18% (w/v) Ficoll, 300 mM NaOH, 60 mM EDTA (pH 8.0), 0.25% (w/v) bromophenol blue and 0.25% (w/v) xylene cyanol). The products were separated at 40 V for 20 h on a 0.8% denaturing agarose gel and visualized by autoradiography.

## In vitro DNA replication on a single-stranded template

A 32-mer oligonucleotide [$^{32}$P]-labelled at the 5′ end (5′-CTATCTCAGCGATCTGTCTATTTCGTTCATCC-3′) was annealed to a single-stranded pBluescript SK(+) O$_L$ plasmid. The reactions (25 μl) contained 0.4 nM of DNA template, 25 mM HEPES (pH 7.6), 10 mM DTT, 10 mM MgCl$_2$, 0.1 mg ml$^{-1}$ BSA, 160 nM mtSSB (calculated as a tetramer) and 4 nM POLγA in complex with 6 nM POLγB (concentration calculated as a dimer). The reactions contained 2.5 μM or 10 μM of dATP, dTTP, dGTP and dCTP. When indicated, ATP, CTP and UTP were added at increasing concentrations.

The reactions were incubated at 37 °C and stopped after 15 min with 5 μl stop buffer (90 mM EDTA, 6% SDS, 30% glycerol and 0.25% bromophenol blue). The products were separated on a 0.8% agarose gel with 0.5 μg ml$^{-1}$ ethidium bromide (EtBr) at 40 V in 1× TBE buffer for 18 h and visualized by autoradiography.

## Reporting summary

Further information on research design is available in the Nature Portfolio Reporting Summary linked to this article.

## Data availability

The MS proteomics data have been deposited at the ProteomeXchange Consortium via the PRIDE repository under dataset identifier PXD053639 (Extended Data Fig. 1d). HydEn-seq fastq data were uploaded to the ENA (PRJEB87869). The Illumina sequencing data are freely available in the SRA database under accession number PRJNA1251856. The metabolic raw data used in this publication and associated metadata including tables with $m/z$ values and retention times of the extracted metabolites have been deposited at Zenodo[57]

(https://doi.org/10.5281/zenodo.15241160). Data supporting the findings of this study are available from the corresponding author on reasonable request. Source data are provided with this paper.

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

**Acknowledgements** We thank the members of the MPI Biology of Ageing core facilities, I. Atanassov and T. Colby (proteomics), J. Boucas (bioinformatics), C. Kukat (FACS and imaging), Y. Hinze (metabolomics) and D. Diehl for expert technical assistance; and the staff at the Cologne Centre for Genomics for performing paired-end sequencing. This work was supported by the Deutsche Forschungsgemeinschaft (DFG, German Research Foundation) under Germany's Excellence Strategy—CECAD, EXC2030 (grant no. 390661388) and as part of the SFB1403 (grant no. 414786233; project A06). M.F. was supported by the Knut and Alice Wallenberg foundation. J.P. was supported by the Medical Research Council (MRC) (MC_UU_00028/5). C.S. was supported by the DFG SFB1453 (431984000), SCHE 2092/4-1 (386793560) and the Heisenberg program to C.S. (501370692) and further supported by the Wilhelm Sander-Stiftung (2023.010.1).

**Author contributions** A.B., D.M and T.L. conceptualized research goals and experiments, interpreted the results and wrote the manuscript with contributions from all of the authors. A.B. and D.M. contributed equally to this work and performed and analysed the majority of experiments. E.C. performed and analysed dPCR, ddPCR and IF experiments with the help of V.P.; M.B. performed and analysed experiments with primary MEFs. T.M. performed metabolomics experiments. M.S. analysed rNTP incorporation into mtDNA after drug-induced nucleotide imbalance. I.M., E.L. and A.R.C. planned, performed and analysed HydEn-seq experiments. M.R. and C.S. performed the histological kidney analysis. S.N. and A.K. performed experiments in senescent cells. H.N. performed and analysed proteomics data and assisted data analysis and interpretation. P.G. performed metabolomics, assisted analysis and interpretation. P.K. assisted with mouse, molecular biology and cell culture work. L.J. and M.F. designed and performed in vitro replication assays. J.P., N.-G.L. and M.F. contributed to data interpretation.

**Funding** Open access funding provided by Max Planck Society.

**Competing interests** M.F. and N.-G.L. are co-founders and shareholders of Pretzel Therapeutics. The other authors declare no competing interests.

**Additional information**
**Correspondence and requests for materials** should be addressed to Thomas Langer.

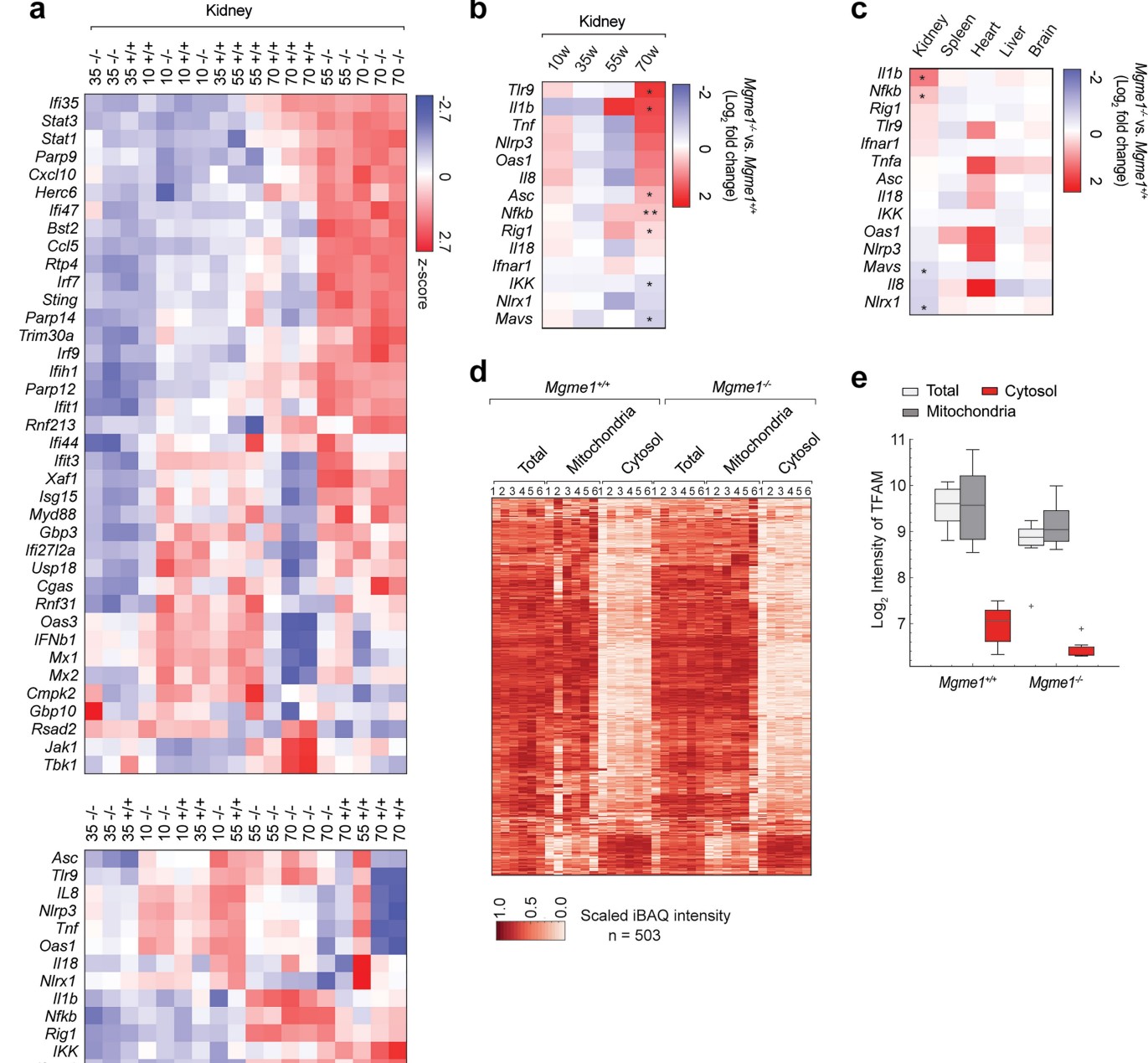

**Extended Data Fig. 1 | Increased ISG signalling in kidneys from *Mgme1*[-/-] mice. a**, Agglomerative heat map showing the distribution of mRNA ISGs (upper panel) and non-ISGs (lower panel) intensities in *Mgme1*[+/+] and *Mgme1*[-/-] mice kidneys at different ages. Z-score intensities were calculated and clustering was performed using the Euclidean distance metric and the 'average' method. Each column represents a different mouse. **b, c**, Heat map (log$_2$ fold change, decreasing order) of non-ISG inflammation related mRNA profiles in *Mgme1*[+/+] and *Mgme1*[-/-] kidneys at different ages (b) and in various tissues of 55-week-old mice (c) obtained by NanoString technologies. **d**, Heat map showing the distribution of mitochondrial proteins (MitoCarta 3.0, n = 503, quantified in all samples) in subcellular fractions of kidney lysates using the iBAQ intensity scaling the lowest and highest value between zero and one. The rows were clustered using the Euclidean distance metric and the 'average' method. Each column represents a different mouse (n = 6 mice per genotype). **e**, TFAM protein levels (log2 iBAQ intensity) in subcellular fractions of kidneys of 55-week-old *Mgme1*[+/+] and *Mgme1*[-/-] mice, identified by quantitative proteomics. The boxplot is defined as follows: each box spans from the 25% to the 75% quantile and the line center indicates the median. The whiskers indicate the minimum and maximum excluding outliers. Outliers are determined by using 1.5 times the inter-quantile range (a cross indicates outliers). n = 6 animals. *q < 0.05, **q < 0.01.

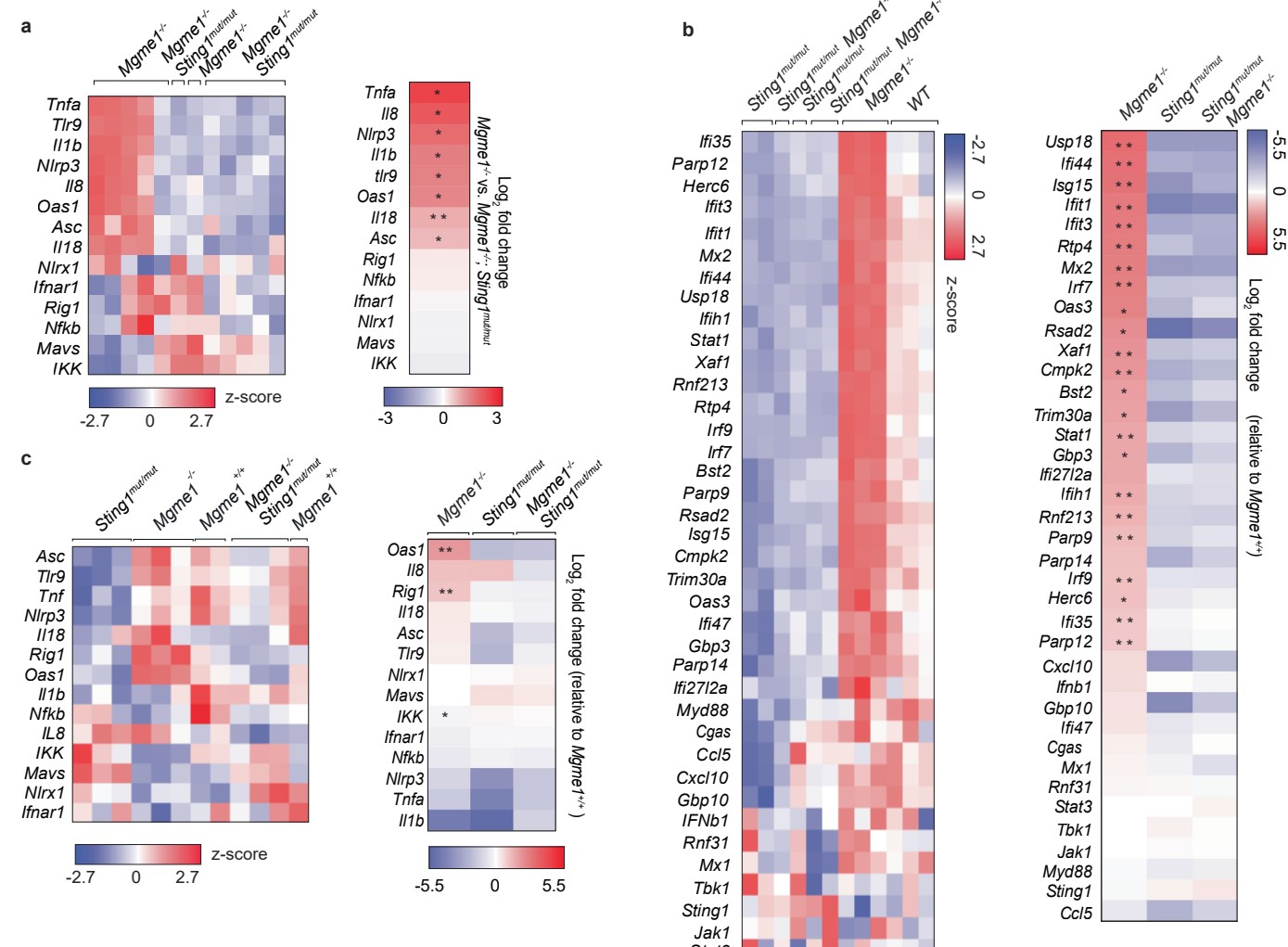

**Extended Data Fig. 2 | cGAS-STING-dependent signalling upon MGME1 loss. a**, Agglomerative heat map showing the distribution of non-ISG inflammation related mRNA intensities in kidneys of 55-week-old $Mgme1^{-/-}$ and $Mgme1^{-/-}$; $Sting1^{mut/mut}$ mice obtained by NanoString profiling. Each column represents a different mouse (left panel). Heat map ($\log_2$ fold change, decreasing order) of mRNA non-ISGs profiles obtained by NanoString technologies (right panel). Asterisks mark ISGs whose expression was significantly altered. n = 6 mice per genotype. **b**, Agglomerative heat map showing the distribution of mRNA ISGs intensities in $Mgme1^{-/-}$, $Sting1^{mut/mut}$ and $Mgme1^{-/-}$; $Sting1^{mut/mut}$ primary MEFs obtained by NanoString profiling. Each column represents a biological replicate (left panel). Heat map ($\log_2$ fold change, decreasing order) of mRNA ISGs profiles of $Mgme1^{-/-}$, $Sting1^{mut/mut}$ and $Mgme1^{-/-}$; $Sting1^{mut/mut}$ primary MEFs obtained by NanoString profiling (right panel). Asterisks mark ISGs whose expression was significantly altered. n = 3 clones per genotype. q values were calculated using unpaired two-tailed Student $t$-test with multiple comparisons correction using the two-stage step-up method of Benjamini, Krieger and Yekutieli. **c**, Agglomerative heat map showing the distribution of non-ISG inflammation related mRNA intensities in $Mgme1^{+/+}$, $Mgme1^{-/-}$, $Sting1^{mut/mut}$ and $Mgme1^{-/-}$; $Sting1^{mut/mut}$ primary MEFs obtained by NanoString profiling. Each column represents a biological replicate (left panel). Row-clustered heat map ($\log_2$ fold change) of mRNA non-ISGs profiles of $Mgme1^{-/-}$, $Sting1^{mut/mut}$ and $Mgme1^{-/-}$; $Sting1^{mut/mut}$ primary MEFs obtained by NanoString technologies (right panel). Asterisks mark ISGs whose expression was significantly altered. n = 3 clones per genotype. *q < 0.05, ** q < 0.01.

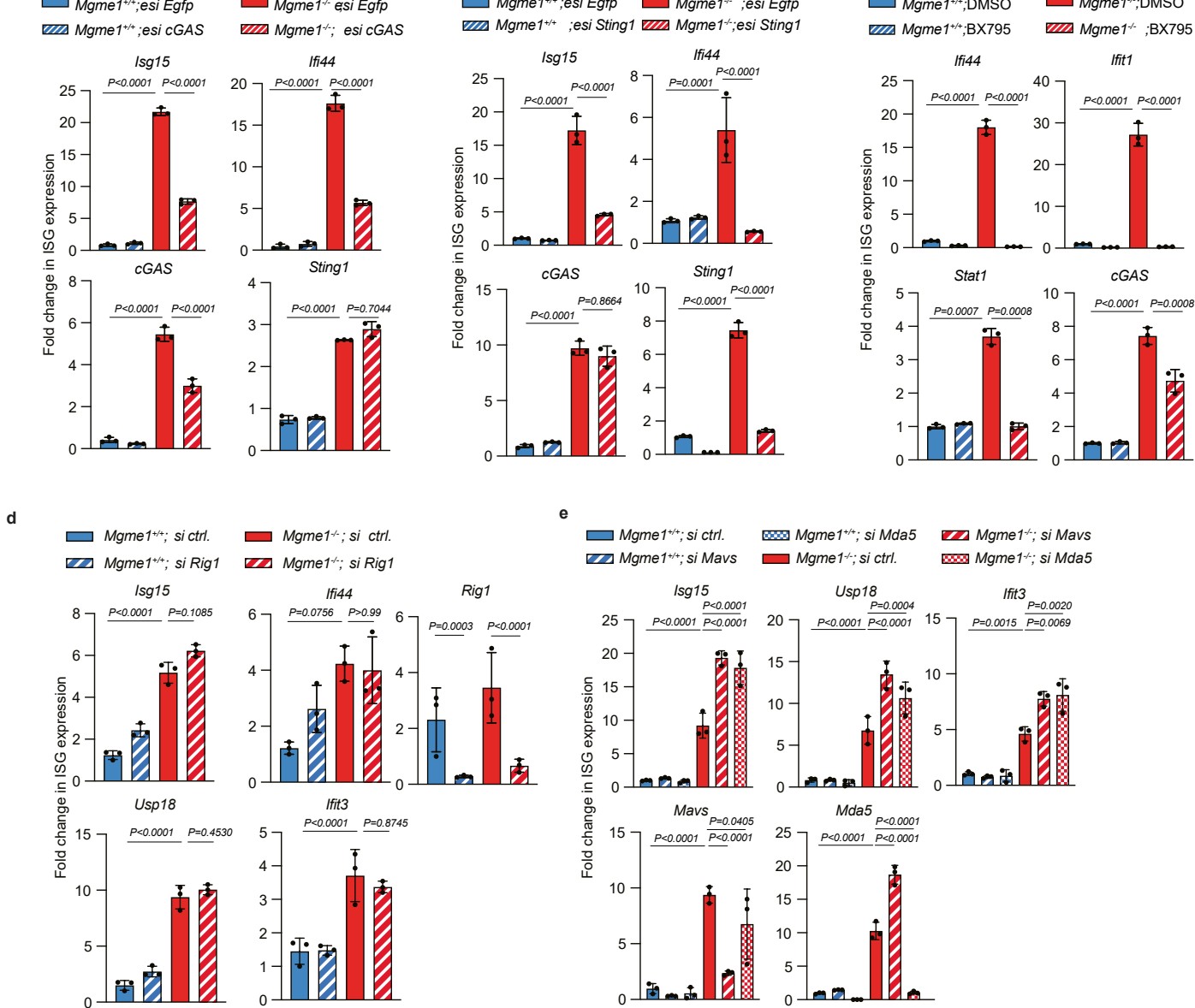

**Extended Data Fig. 3 | Loss of MGME1 elicits cGAS-STING-dependent innate immune signalling. a, b**, ISG expression in *Mgme1*⁺/⁺ and *Mgme1*⁻/⁻ immortalized MEFs treated with the indicated esiRNA for 72 h. n = 3 biologically independent experiments. **c**, ISG expression in *Mgme1*⁺/⁺ and *Mgme1*⁻/⁻ immortalized MEFs treated with the TBK1 inhibitor BX795 (0.5 µM) for 72 h. n = 3 biologically

independent experiments. **d, e**, ISG expression in *Mgme1*⁺/⁺ and *Mgme1*⁻/⁻ immortalized MEFs treated with the indicated siRNA for 72 h and measured by RT-qPCR. n = 3 biologically independent experiments. *P* values were calculated using two-way analysis of variance (ANOVA) with Tukey's multiple comparisons test (a, b, c, d, e). Data are presented as mean ± SD.

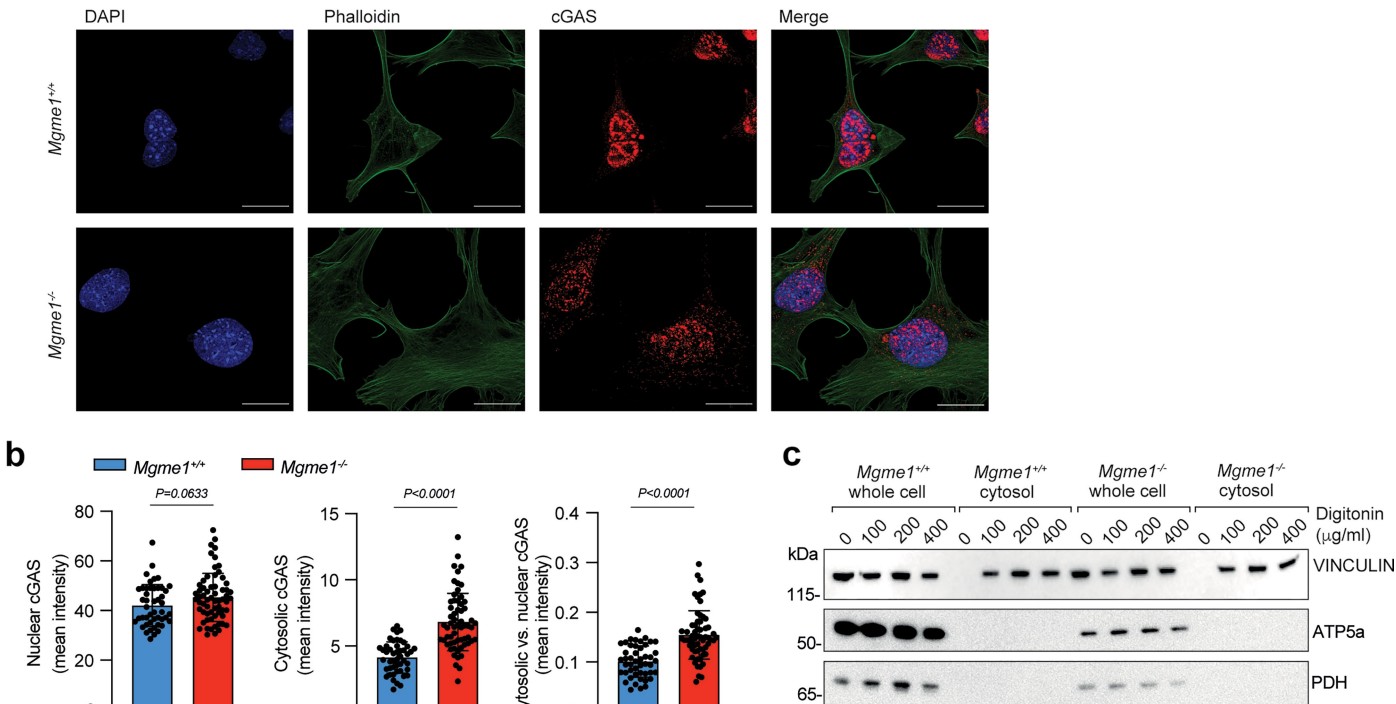

**Extended Data Fig. 4 | Increased cytosolic cGAS localization in *Mgme1*⁻/⁻ cells. a**, cGAS localization in *Mgme1*⁺/⁺ and *Mgme1*⁻/⁻ immortalized MEFs, monitored by confocal immunofluorescence imaging. Cells were stained with DAPI (blue), Phalloidin (green) and anti-cGAS antibody (red). Scale bars, 5 μm. **b**, Quantification of cGAS localization in the nucleus and the cytosol of *Mgme1*⁺/⁺ and *Mgme1*⁻/⁻ immortalized MEFs obtained by confocal immunofluorescence analysis. cGAS nuclear mean intensity (left panel), cGAS cytosolic mean intensity (middle panel) and ratio of cytosolic to nuclear mean intensities (right panel) are shown. n = 51 and n = 64 independent *Mgme1*⁺/⁺ and *Mgme1*⁻/⁻ cells were analysed, respectively. **c**, Immunoblot analysis of total and cytosolic fractions of *Mgme1*⁺/⁺ and *Mgme1*⁻/⁻ MEFs used in Fig. 2c. A representative blot of three independent fractionation experiments is shown. *P* values were calculated using unpaired two-tailed Student *t*-test (b). Data are presented as mean ± SD.

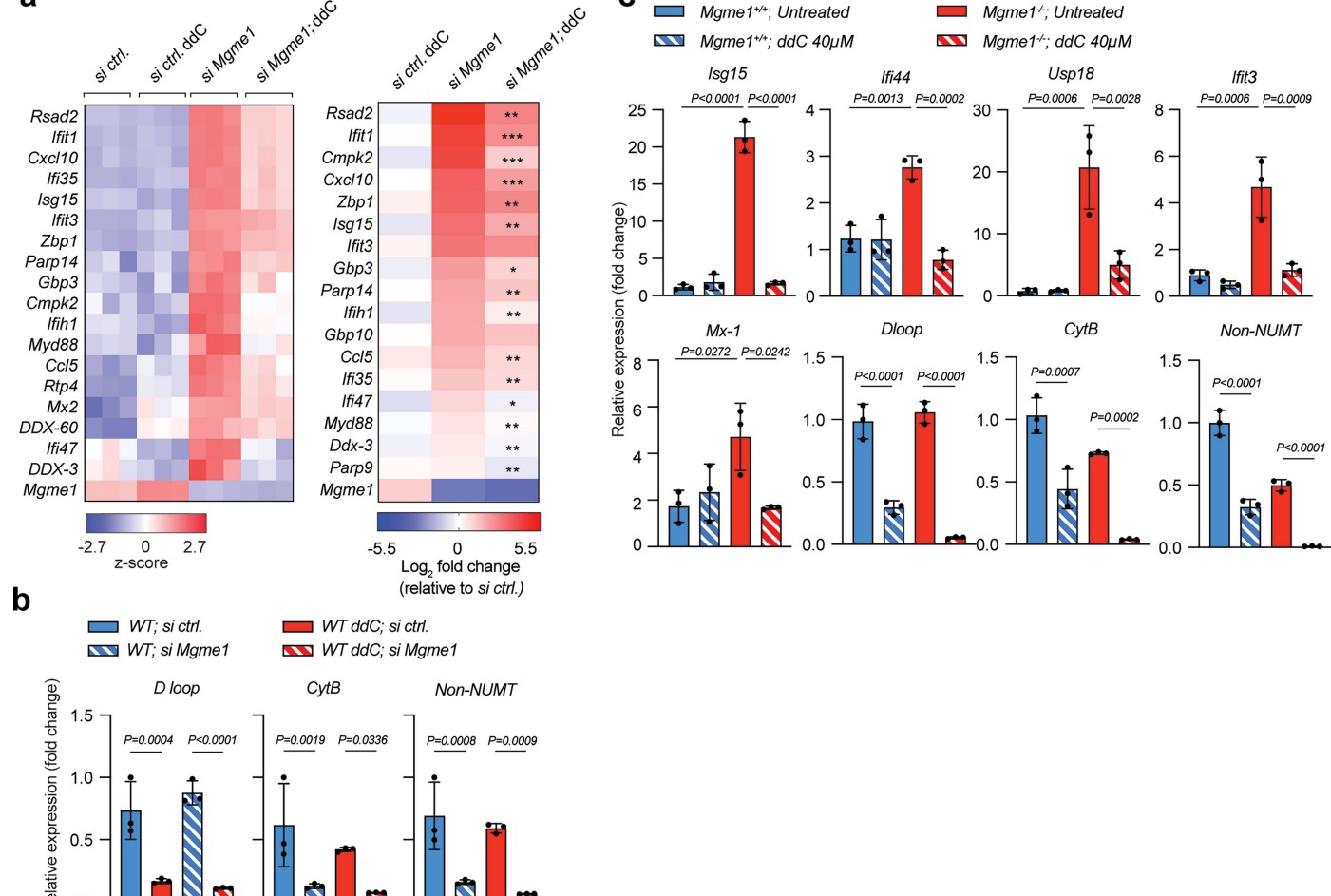

**Extended Data Fig. 5 | mtDNA-dependent ISG response in *Mgme1^-/-* cells.**
**a**, Agglomerative heat map showing the distribution of mRNA ISGs intensities in wild-type immortalized MEFs either left untreated or treated with 40 μM ddC and indicated siRNA for 72 h (left panel). Z-score intensities were calculated and clustering was performed using the Euclidean distance metric and the 'average' method (left panel). Heat map (log$_2$ fold change, decreasing order) of ISGs mRNA profiles in wild-type immortalized MEFs treated with the indicated siRNA for 72 h (right panel). Asterisks mark ISGs whose expression was significantly altered between siMgme1 and siMgme1; ddC (n = 3 biologically independent experiments). *q < 0.05; **q < 0.01, ***q < 0.001. **b**, RT-qPCR analysis of mtDNA

levels in wild-type immortalized MEFs either left untreated or treated with 40 μM ddC and indicated siRNA for 72 h. mtDNA levels were monitored with probes for *D loop*, cytochrome b (*CytB*) and non-NUMT. n = 3 biologically independent experiments. **c**, ISG expression and mtDNA levels in *Mgme1^+/+* and *Mgme1^-/-* immortalized MEFs treated with 40 μM ddC for 6 days. n = 3 biologically independent experiments. *P* values were calculated using unpaired two-tailed Student *t*-test (a), two-way ANOVA with Tukey's multiple comparison test (b) or one-way ANOVA with Tukey's multiple comparison test (c). data are presented as mean ± SD. *q < 0.05, **q < 0.01, ***q < 0.001 (**a**).

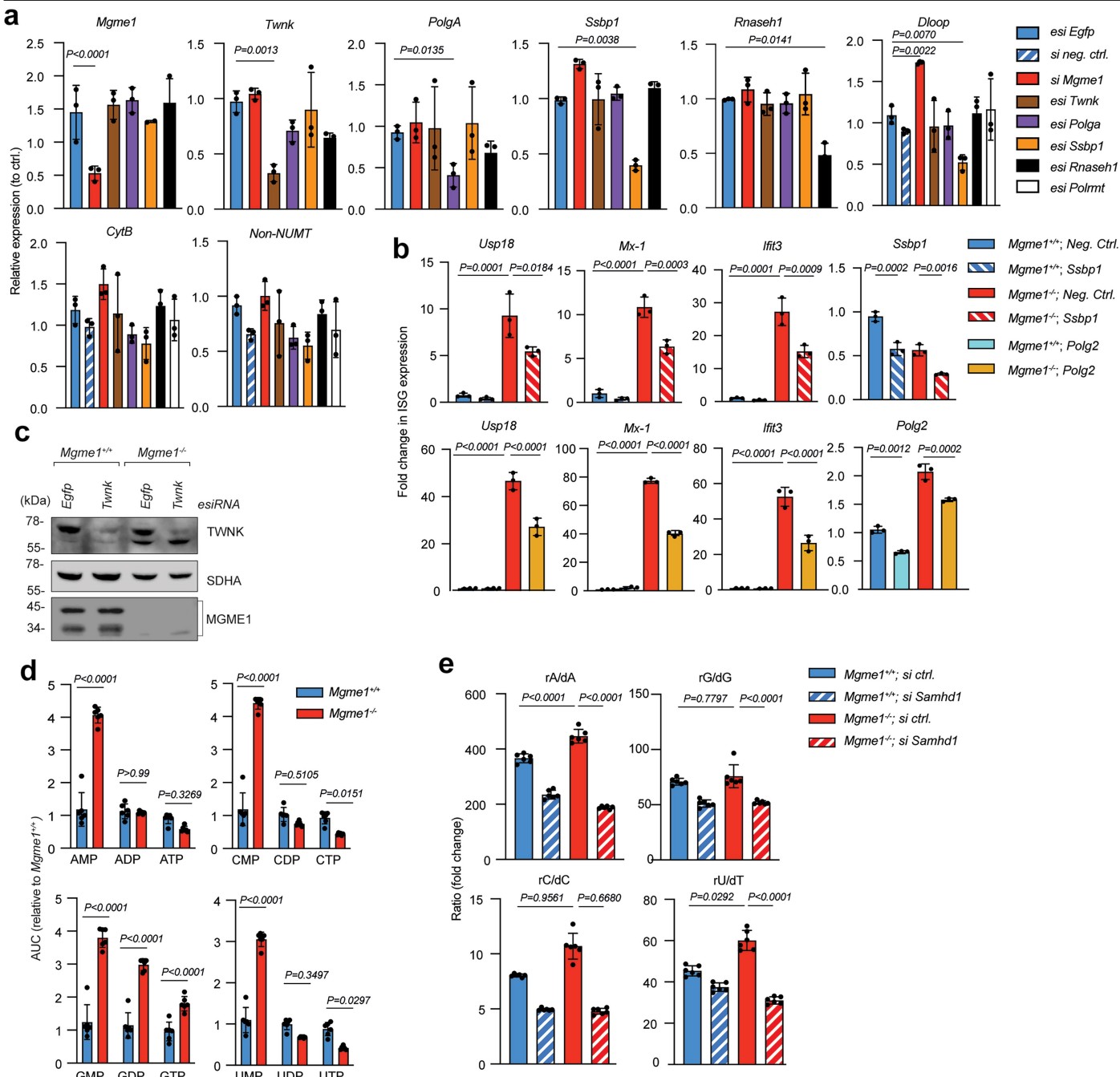

**Extended Data Fig. 6 | Ongoing mtDNA replication is required for ISG expression. a**, RT-qPCR analysis of wild-type MEFs depleted of replication enzymes with the indicated siRNAs for 72 h. mtDNA levels were monitored with probes for *D loop*, cytochrome b (*CytB*) and non-NUMT (non-nuclear mitochondrial DNA). n = 3 biologically independent experiments. **b**, ISG expression in *Mgme1*⁺/⁺ and *Mgme1*⁻/⁻ immortalized MEFs treated with the indicated siRNA or esiRNA for 72 h. n = 3 independent cultures. **c**, A representative immunoblot analysis with the indicated antibodies in *Mgme1*⁺/⁺ and *Mgme1*⁻/⁻ immortalized MEFs treated with the indicated siRNA for 72 h. **d**, Ribonucleotide

levels in cell lysates of *Mgme1*⁺/⁺ and *Mgme1*⁻/⁻ immortalized MEFs, determined by LC-MS (n = 6 independent cultures). Samples were normalized to protein levels. **e**, Ratio of ribonucleotides triphosphates (rNTP) and deoxyribonucleotides triphosphates (dNTP) from cell extracts of immortalized *Mgme1*⁺/⁺ and *Mgme1*⁻/⁻ cells treated with control and siRNAs for *Samhd1*, determined by LC-MS (n = 6 independent cultures). *P* values were calculated using two-way ANOVA with Tukey's (a, e) or Šidák's multiple comparison test (d) or one-way ANOVA with Tukey's multiple comparison test (b). data are presented as mean ± SD.

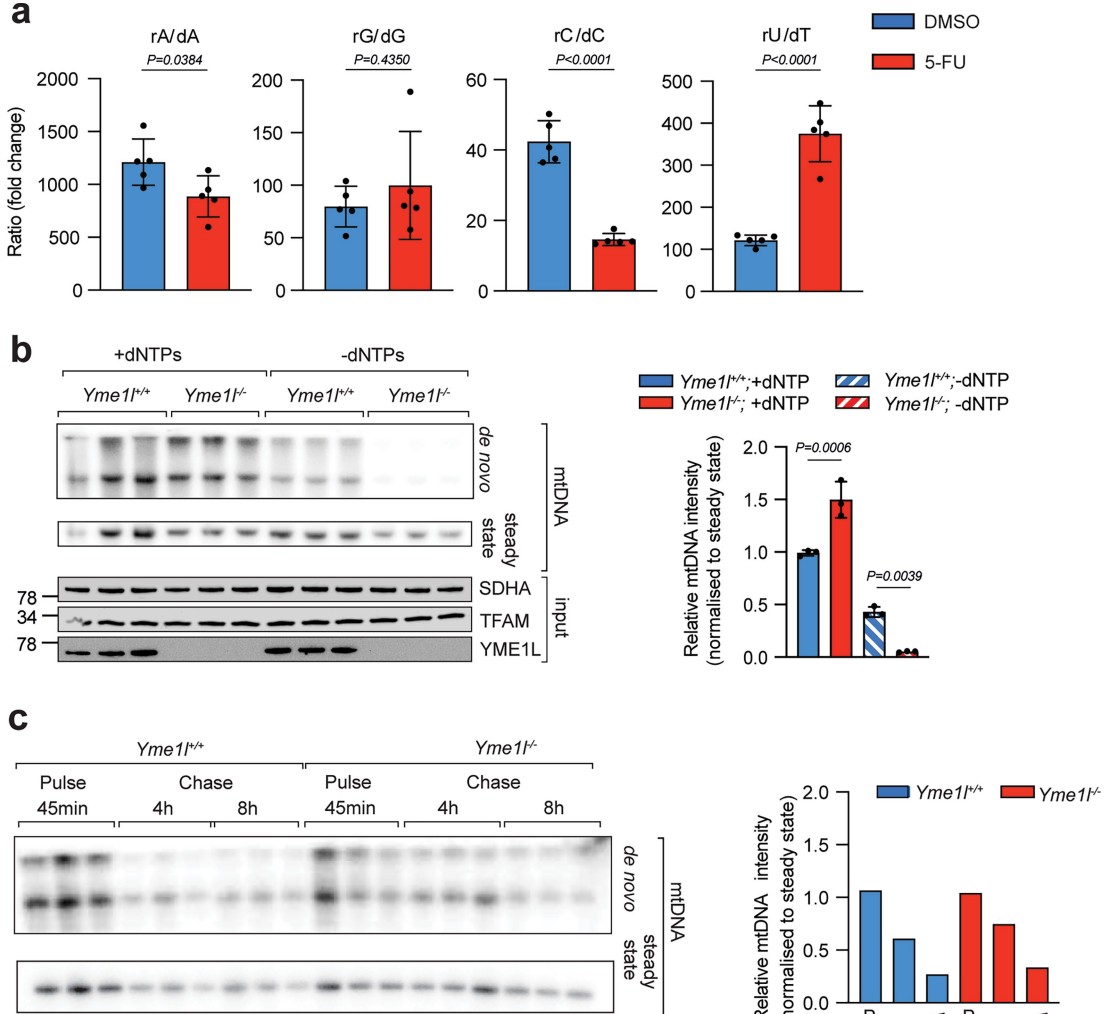

**Extended Data Fig. 7 | The rate of mtDNA synthesis depends on the rNTP/
dNTP ratio. a**, rNTP/dNTP ratios from cell extracts of immortalized wild-type
MEFs determined by LC-MS. Cells were either left untreated or treated with
5-FU at 5 μM for 16 h (n = 5 independent cultures). **b**, De novo mtDNA synthesis
in *Yme1l*[+/+] and *Yme1l*[−/−] mitochondria in the presence or absence of exogenous
dNTPs. Isolated mitochondria were pulse labelled for 1 h. Steady state levels of
mtDNA and the level of newly synthesized mtDNA are shown in the left-hand
side upper panels. The input fraction was analysed by immunoblotting
(left-hand side lower panel). Quantification of de novo synthesized mtDNA in

*Yme1l*[+/+] and *Yme1l*[−/−] mitochondria is shown on the right panel. n = 3 biologically
independent experiments. **c**, De novo mtDNA synthesis in mitochondria
isolated from *Yme1l*[+/+] and *Yme1l*[−/−] MEFs. Isolated mitochondria were pulsed
labelled for 45 min and further incubated for 4 and 8 h. De novo and steady state
mtDNA levels are shown (left panel). Newly synthesized mtDNA was quantified
relative to mtDNA steady state levels in *Yme1l*[+/+] and *Yme1l*[−/−] mitochondria
(right panel). n = 2 biologically independent experiments. *P* values were
calculated using unpaired two-tailed Student *t*-test (a) or using one-way ANOVA
with Tukey's multiple comparison test (b). Data are presented as mean ± SD.

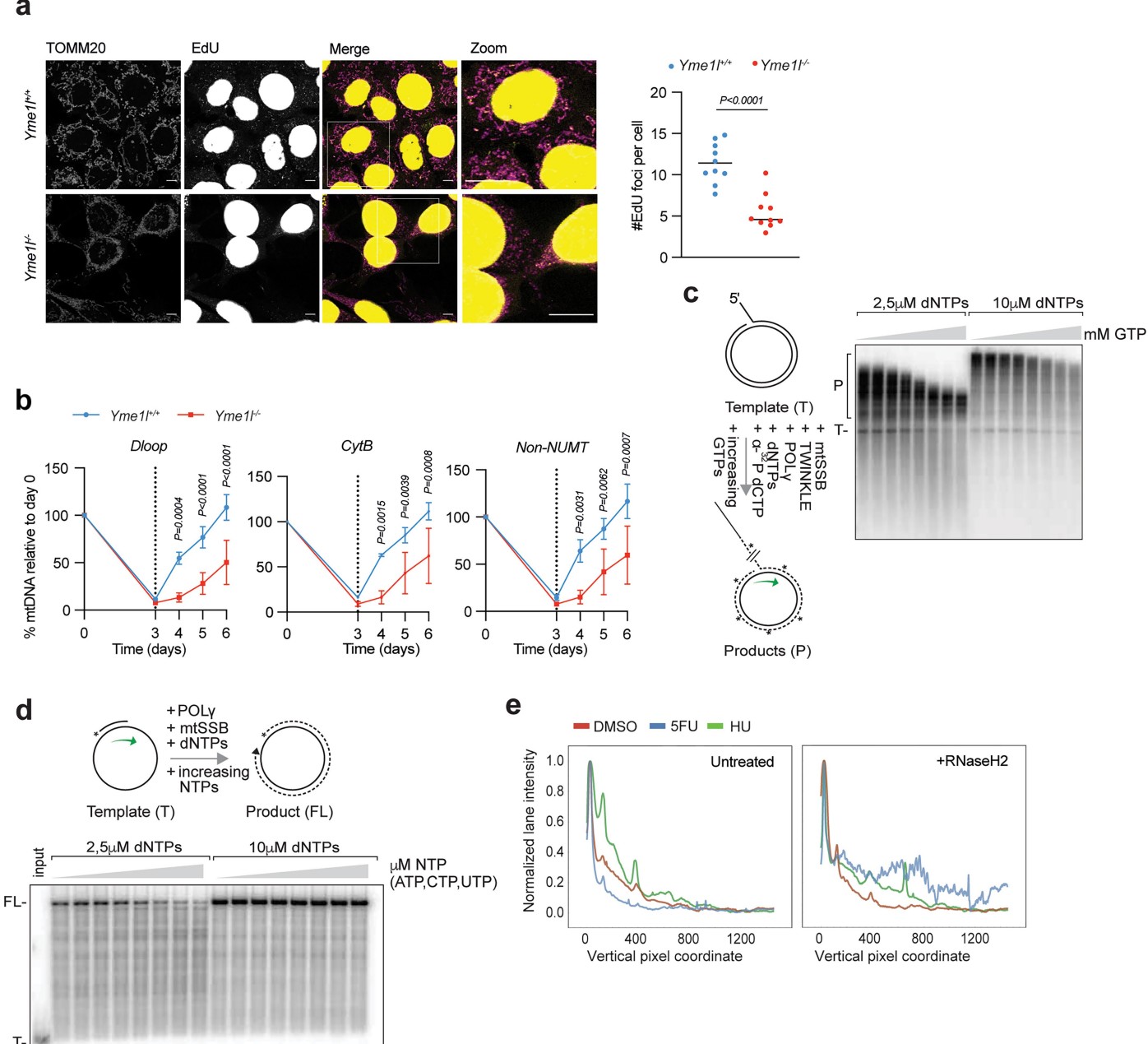

**Extended Data Fig. 8 | rNTP/dNTP ratio influence rate of mtDNA synthesis.**
**a**, EdU labelling of *Yme1l*[+/+] and *Yme1l*[−/−] MEFs for 4 h to visualize newly synthesized
DNA (yellow), following by TOMM20 antibody staining (magenta). Confocal
images are shown including an enlargement image of the inset. Nuclear genomic
replication is shown together with smaller foci of newly synthesized mtDNA.
Scale bars, 5 μm. Each quantified dot (right panel) represents the average number
of newly synthesized mtDNA foci per cell in ten different images. **b**, mtDNA
depletion/repopulation assay in *Yme1l*[+/+] and *Yme1l*[−/−] MEFs. After mtDNA
depletion with 2´,3´-dideoxycytidine (ddC; 40 μM) and removal of ddC, mtDNA
replenishment was monitored. n = 3 biologically independent experiments.
**c**, Effects of increasing concentration of GTP on in vitro DNA replication, which
was performed for 60 min at 37 °C. DNA products with incorporated radioactive
nucleotides were separated on a 0.8% alkaline agarose gel. Template (T);

Products (P). **d**, DNA replication reactions containing recombinant POLγ,
mtSSB, and a radiolabeled, primed single-stranded DNA template were
incubated with increasing concentrations of CTP, UTP, or ATP (0, 10, 50, 100,
200, 300, 400, and 500 μM for each NTP) for 15 min at 37 °C. DNA synthesis
products were separated on an agarose gel. Full-length (FL) product and the
radioactively labelled primer/template (T) are indicated. **e**, Quantification of
Southern blot analysis of alkaline agarose gel resolving mtDNA isolated from
wild-type MEFs treated with DMSO (control), 5 μM 5-FU and 100 nM HU. Signal
intensities of individual pixels were determined to generate plots for untreated
(left panel) and RNaseH2-treated samples (right panel). The intensity data were
scaled between 0 and 1. *P* values were calculated using unpaired two-tailed
Student *t*-test (a) or using two-way ANOVA with Šidák's multiple comparison
test (b). Data are presented as mean ± SD.

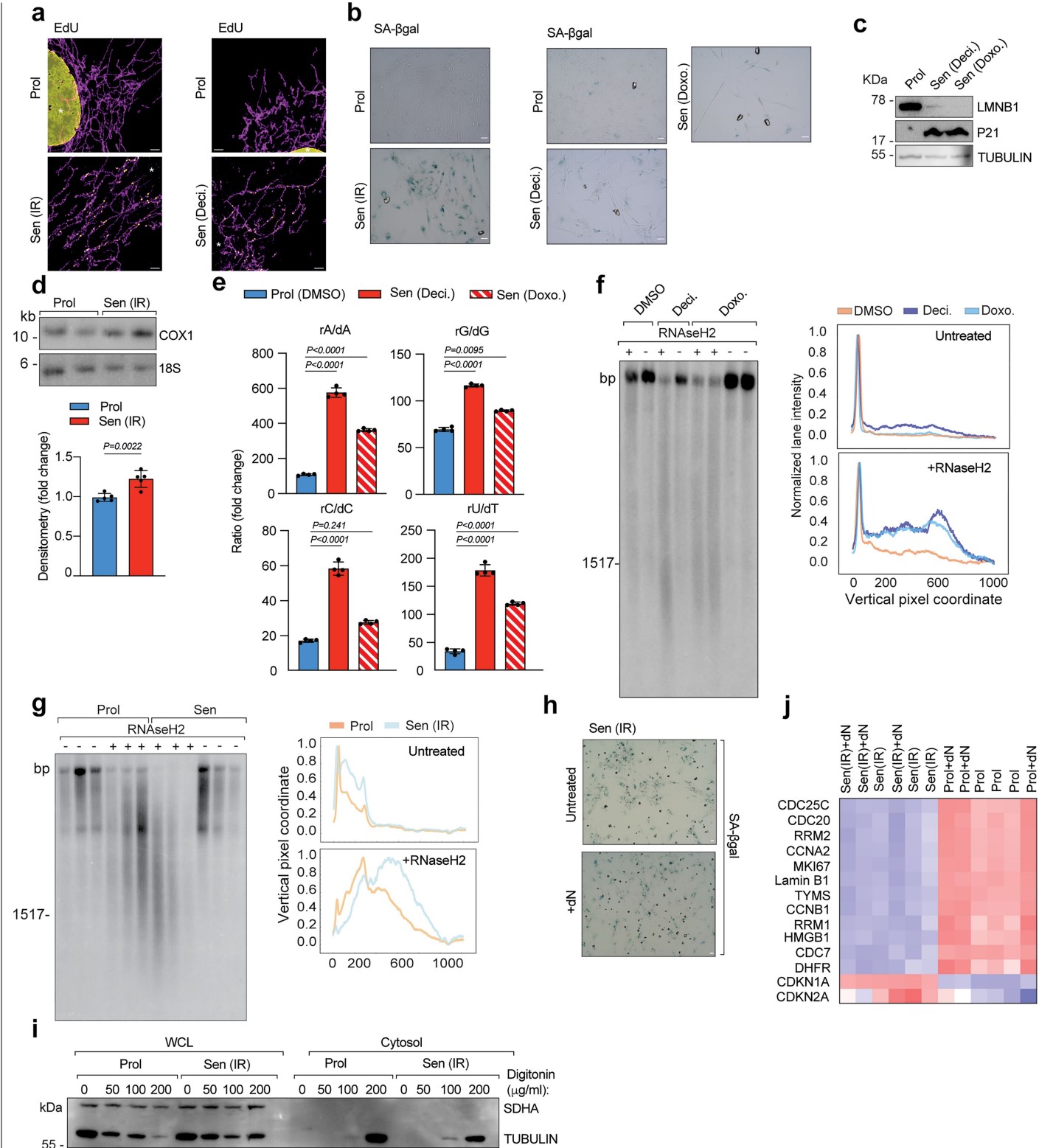

**Extended Data Fig. 9** | See next page for caption.

**Extended Data Fig. 9 | Senescence is associated with increased cellular rNTP/dNTP ratios and increased ribonucleotide incorporation into mtDNA. a**, EdU labelling in proliferating, irradiation-induced senescent cells (Sen IR) or TIS therapy-induced senescent (Deci.-decitabine;) for 24 h (IMR90). Confocal images show newly synthesized DNA in yellow and mitochondrial TOMM20 (after antibody staining) in magenta. Asterix labels nuclei. Scale bars, 2 µm. **b**, Representative images of senescence-associated ß-galactosidase (SA-β-GAL) staining from proliferating (Prol) and radiation-induced senescent (Sen IR) or (TIS) therapy-induced senescent (Deci; decitabine; Doxo; doxorubicine) human fibroblasts (IMR90). Scale bars, 50 µm. n = 3 biologically independent experiments. **c**, Senescent state of TIS cells was analysed by immunoblot with indicated antibodies. **d**, Southern blot analysis of total DNA from proliferating and irradiation-induced senescent IMR90 cells using a *Cox1* probe and quantification (lower panel). DNA levels for nuclear 18S rRNA are used as a loading control. n = 3 biologically independent experiments. **e**, Ratio of rNTP to dNTP in cell extracts from proliferating and senescent IMR90 fibroblast treated with decitabine or doxorubicine was measured by LC-MS. n = 3 biologically independent experiments. **f**, Alkaline agarose gel electrophoresis was performed on mtDNA, which was isolated from proliferating and senescent (Deci. and Doxo.) IMR90 cells and treated with RNaseH2 where indicated. Southern blots were analysed as in Fig. 4f. Representative plot of two biologically independent experiments is shown. **g**, same like f except that irradiated primary renal proximal tubule epithelial cells (RPTEC) were analysed. A representative plot of n = 2 biologically independent experiments is shown. **h**, SA-β-GAL staining of IR-senescent IMR90 cells in presence and absence of deoxyribonucleosides. Scale bars, 50 µm. **i**, Immunoblot analysis of cytosolic fractions of proliferating and irradiation-induced senescent cells used in Fig. 5c. **j**, Agglomerative heat map showing the distribution of mRNA between proliferating and irradiation-induced senescent IMR90 cells, in presence or absence of deoxyribonucleosides. n = 3 biologically independent experiments. *P* values were calculated using paired two-tailed Student *t*-test (d) or using two-way ANOVA with Tukey's multiple comparison test (e). Data are presented as mean ± SD.

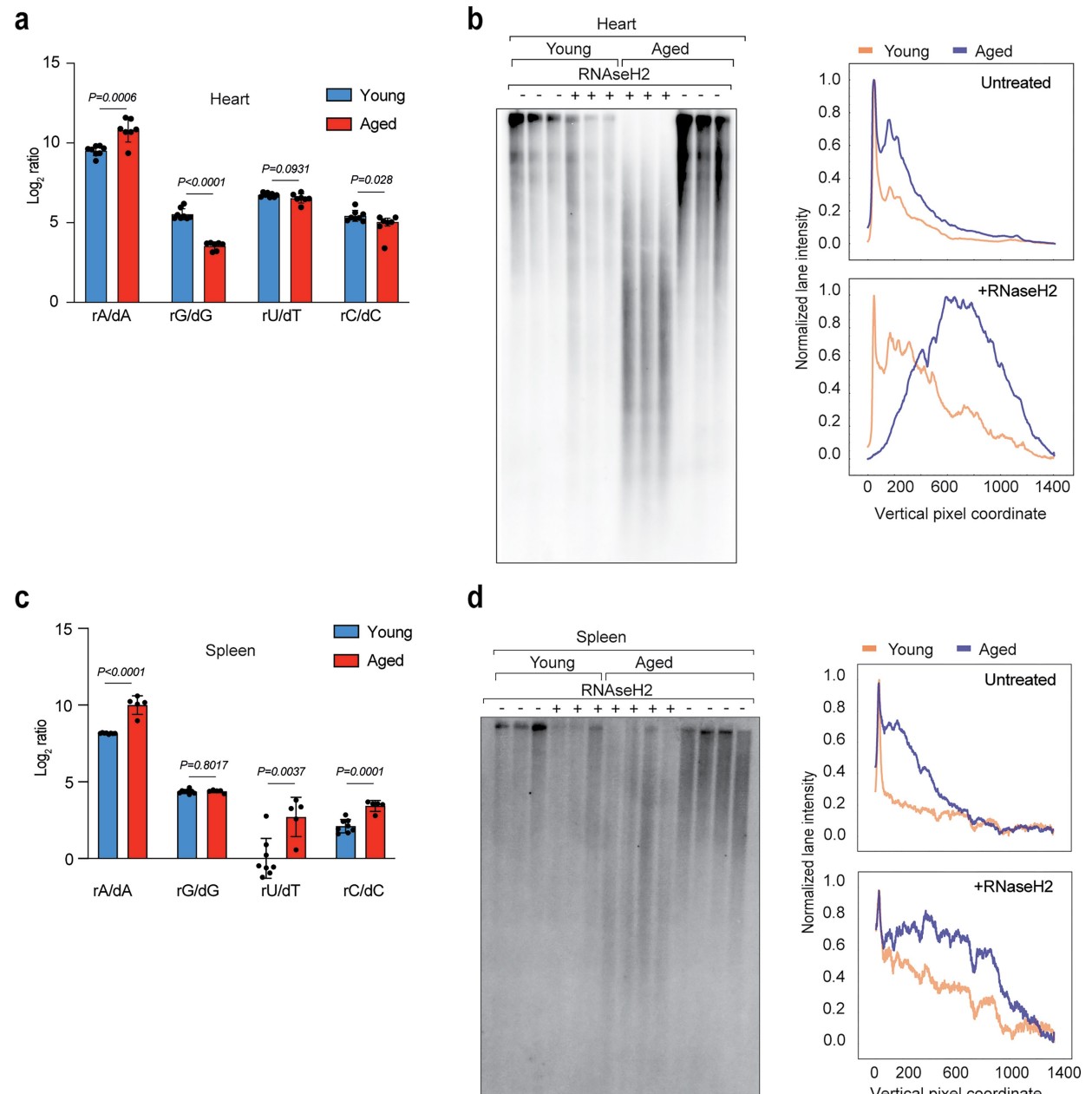

**Extended Data Fig. 10 | Increased ribonucleotide content of mtDNA in aged heart and spleen tissues. a**, Ratio of ribonucleotides triphosphates (rNTP) and deoxyribonucleotides triphosphates (dNTP) from heart lysates of young (1 week of age, n = 8 animals) and old (80–87 weeks of age, n = 6 animals) mice was determined by LC-MS **b**, Untreated and RNaseH2-treated DNA isolated from heart tissue of young (1 week of age) and old (80–97 weeks of age) mice was analysed by alkaline agarose gels and Southern blot (left). Quantification of Southern blot from was performed as described in Fig. 4f (right panel). **c**, same as **a** except that spleen tissue was analysed. Young (1 week of age, n = 8 animals) and old (80–87 weeks of age, n = 5 animals). **d**, same as **b** except that spleen tissue was analysed. P values were calculated using paired two-tailed Student *t*-test (a,c). Data are presented as mean ± SD.

# Reporting Summary

## Statistics

For all statistical analyses, confirm that the following items are present in the figure legend, table legend, main text, or Methods section.

| n/a | Confirmed | |
|---|---|---|
| ☐ | ☒ | The exact sample size (*n*) for each experimental group/condition, given as a discrete number and unit of measurement |
| ☐ | ☒ | A statement on whether measurements were taken from distinct samples or whether the same sample was measured repeatedly |
| ☐ | ☒ | The statistical test(s) used AND whether they are one- or two-sided *Only common tests should be described solely by name; describe more complex techniques in the Methods section.* |
| ☒ | ☐ | A description of all covariates tested |
| ☐ | ☒ | A description of any assumptions or corrections, such as tests of normality and adjustment for multiple comparisons |
| ☐ | ☒ | A full description of the statistical parameters including central tendency (e.g. means) or other basic estimates (e.g. regression coefficient) AND variation (e.g. standard deviation) or associated estimates of uncertainty (e.g. confidence intervals) |
| ☐ | ☒ | For null hypothesis testing, the test statistic (e.g. *F*, *t*, *r*) with confidence intervals, effect sizes, degrees of freedom and *P* value noted *Give P values as exact values whenever suitable.* |
| ☒ | ☐ | For Bayesian analysis, information on the choice of priors and Markov chain Monte Carlo settings |
| ☒ | ☐ | For hierarchical and complex designs, identification of the appropriate level for tests and full reporting of outcomes |
| ☒ | ☐ | Estimates of effect sizes (e.g. Cohen's *d*, Pearson's *r*), indicating how they were calculated |

*Our web collection on statistics for biologists contains articles on many of the points above.*

## Software and code

Policy information about availability of computer code

| Data collection | Radioactive signals were developed using Typhoon FLA 9500. qPCR were performed using 384 Quant studio 7 flex real time pCR macine (Thermo fisher). Nanostringprofiling was performed using nCOUNTER SPRINT profiler. Fluorescence images were acquired using confocal TSC SP8 DLS  and TSC SP8X microscopes. dd PCR was performed using digital droplet PCR (Biorad) and dPCR using Qiagen QiaCutie . |
|---|---|
| Data analysis | Fiji software 4;InstantClue 2; GraphPad Prism 10;Multi Gauge V3.0, nSolver analysis software, Adobe Photoshop 23.5.0, Adobe Illustrator 26.5 |

For manuscripts utilizing custom algorithms or software that are central to the research but not yet described in published literature, software must be made available to editors and reviewers. We strongly encourage code deposition in a community repository (e.g. GitHub). See the Nature Portfolio guidelines for submitting code & software for further information.

## Data

Policy information about availability of data

All manuscripts must include a data availability statement. This statement should provide the following information, where applicable:
- Accession codes, unique identifiers, or web links for publicly available datasets
- A description of any restrictions on data availability
- For clinical datasets or third party data, please ensure that the statement adheres to our policy

The mass spectrometry proteomics data have been deposited to the ProteomeXchange Consortium via the PRIDE repository with the dataset identifier PXD053639 (Extended Data Fig. 1d ). HydEn seq fastq data is uploaded to ENA (accession number PRJEB87869). The Illumina sequencing data is data freely available in the SRA

## Research involving human participants, their data, or biological material

Policy information about studies with human participants or human data. See also policy information about sex, gender (identity/presentation), and sexual orientation and race, ethnicity and racism.

| | |
|---|---|
| Reporting on sex and gender | N/A |
| Reporting on race, ethnicity, or other socially relevant groupings | N/A |
| Population characteristics | N/A |
| Recruitment | N/A |
| Ethics oversight | N/A |

Note that full information on the approval of the study protocol must also be provided in the manuscript.

# Field-specific reporting

Please select the one below that is the best fit for your research. If you are not sure, read the appropriate sections before making your selection.

☒ Life sciences ☐ Behavioural & social sciences ☐ Ecological, evolutionary & environmental sciences

For a reference copy of the document with all sections, see nature.com/documents/nr-reporting-summary-flat.pdf

# Life sciences study design

All studies must disclose on these points even when the disclosure is negative.

| | |
|---|---|
| Sample size | No sample size was calculated. Based on previous experience and published data for similar experiments sample size was determined. PMID: 3390377, PMID:31695197 |
| Data exclusions | Due to a genotyping error, we had to exclude one knock-out (sample #2) from the Mgme1-/- kidney and, conversely, one sample from each of the other sets for Illumina sequencing was excluded.. |
| Replication | Where applicable the experiments were repeated at least three times and representative experiments were shown. For some mouse experiments involving more than three individual animals, the experiment was not replicated. The number of mice used for the experiments is indicated in the figure legends. Where representative data are shown, the experimental findings were successfully replicated in independent experiments (numbers indicated in respective figure legends). |
| Randomization | Mice were assigned to experimental groups based on genotypes and age. Cells were allocated by genotype and treatment. |
| Blinding | Analysis were not blinded because experiments were performed and analysed by the same researchers. |

# Reporting for specific materials, systems and methods

We require information from authors about some types of materials, experimental systems and methods used in many studies. Here, indicate whether each material, system or method listed is relevant to your study. If you are not sure if a list item applies to your research, read the appropriate section before selecting a response.

## Materials & experimental systems

| n/a | Involved in the study |
|---|---|
| ☐ | ☒ Antibodies |
| ☐ | ☒ Eukaryotic cell lines |
| ☒ | ☐ Palaeontology and archaeology |
| ☐ | ☒ Animals and other organisms |
| ☒ | ☐ Clinical data |
| ☒ | ☐ Dual use research of concern |
| ☒ | ☐ Plants |

## Methods

| n/a | Involved in the study |
|---|---|
| ☒ | ☐ ChIP-seq |
| ☒ | ☐ Flow cytometry |
| ☒ | ☐ MRI-based neuroimaging |

## Antibodies

| | |
|---|---|
| Antibodies used | The information regarding the antibodies are provided in Supplementary table 5. |
| Validation | Antibodies were used according to the manufacturer's recommendation and/or previous published work. PMID: 33903774 PMID: 31695197 |

## Eukaryotic cell lines

Policy information about cell lines and Sex and Gender in Research

| | |
|---|---|
| Cell line source(s) | IMR90 ,RPTEC and HeLa cell lines were acquired from ATCC. MEFs were described in PMID: 3390377, PMID: 24616225 and PMID: 29572490 |
| Authentication | None of the cell lines was authenticated. |
| Mycoplasma contamination | All cell lines were regulary tested and were negative for mycoplasma contamination |
| Commonly misidentified lines (See ICLAC register) | No commonly misidentified cell lines were used. |

## Animals and other research organisms

Policy information about studies involving animals; ARRIVE guidelines recommended for reporting animal research, and Sex and Gender in Research

| | |
|---|---|
| Laboratory animals | Mus musculus C57BL/6N mouse strain  and mice of mixed C57BL/6N x C57BL/6J background were used. C57BL/6N mice of 2w, 55w,80-86w and  mixed C57BL/6N x C57BL/6J of 10, 35,55,70w of age were used for experiemnts as   reported in the figure legends/ and or method section. |
| Wild animals | The study did not use wild animals |
| Reporting on sex |  Animals of both sexes  were used in the study |
| Field-collected samples | The study did not used samples collected from the field. |
| Ethics oversight | The study was approved by the Landesamt für Natur, Umwelt und Verbraucherschutz Nordrhein–Westfalen (reference numbers 81.02.04.2020.A082, 81-02.04.2022.A453) and  performed in accordance with the recommendations and guidelines of the Federation of European Laboratory Animal Science Associations (FELASA). |

Note that full information on the approval of the study protocol must also be provided in the manuscript.

## Plants

| | |
|---|---|
| Seed stocks | *Report on the source of all seed stocks or other plant material used. If applicable, state the seed stock centre and catalogue number. If plant specimens were collected from the field, describe the collection location, date and sampling procedures.* |
| Novel plant genotypes | *Describe the methods by which all novel plant genotypes were produced. This includes those generated by transgenic approaches, gene editing, chemical/radiation-based mutagenesis and hybridization. For transgenic lines, describe the transformation method, the number of independent lines analyzed and the generation upon which experiments were performed. For gene-edited lines, describe the editor used, the endogenous sequence targeted for editing, the targeting guide RNA sequence (if applicable) and how the editor was applied.* |
| Authentication | *Describe any authentication procedures for each seed stock used or novel genotype generated. Describe any experiments used to assess the effect of a mutation and, where applicable, how potential secondary effects (e.g. second site T-DNA insertions, mosiacism, off-target gene editing) were examined.* |

