## [Peer Review file · Nature]

Ribonucleotide incorporation into mitochondrial DNA drives inflammation

Corresponding Author: Professor Thomas Langer

Version 0:

Reviewer comments:

Referee #1

(Remarks to the Author)

Summary

Bahat et al. provide a compelling, clear and well-written manuscript investigating how increased ribonucleotide incorporation into mtDNA leads to increased cytosolic mtDNA, cGAS-STING activation, and inflammatory response. While previous work has demonstrated that mtDNA release into the cytoplasm through various pathways leads to this inflammatory response, this work is novel in that it rigorously demonstrates one mechanism by which this is triggered – imbalanced nucleotide pools and subsequent ribonucleotide incorporation during mtDNA replication. In particular, this is demonstrated using both mouse models and cell culture through multiple approaches, and further supported by the amelioration of these phenotypes by increasing deoxyribonucleotides. Their data support a model by which an imbalance in nucleotide pools leads to increased ribonucleotide incorporation. There are a few results that, based on my understanding, are not fully explained by their model and aren't adequately addressed in the text. Although, I believe many of these can be addressed by discussion and clarity in the text. Overall, this is a strong manuscript that will be of interest to many fields.

Major Comments

1. Figure 1d shows an increase in cytosolic mtDNA, specifically with regions proximal to the heavy strand origin of replication, consistent with these being non-productive replication intermediates that have been released. The identity of these fragments (e.g. the D-loop and Cytb) is consistent with this being a replication product. In which case, do you see the cytosolic presence of fragments derived from the light strand origin of replication (e.g. Cox1)?

In addition, do you know if the cytosolic fragments are single-stranded or double-stranded? My understanding was that cGAS-STING is primarily activated by dsDNA. However, the identity of these fragments from nucleotide imbalance as ssDNA or dsDNA would suggest potentially distinct mechanisms for their generation and release. Previously published results about mtDNA release into the cytoplasm would suggest these are dsDNA molecules.

2. Figure 3b demonstrates that depletion of Twnk suppresses the ISG response observed in *Mgme1*^{-/-} immortalized MEFs. The authors argue that this could be due to ongoing mtDNA synthesis being necessary to trigger mtDNA release. However, it is not clear to me why this wouldn't be observed with depletion of other replisome components. Is this something you have tested and is this suppression unique to Twnk?

3. In Figure 4b, the authors demonstrate that the rA/dA, rG/dG, and rU/dT ratios increase in *Mgme1*^{-/-} MEFs (although all 4 increase in Sen (IR)). Can the authors speculate why the rC/dC ratio decreases? Can the rC/dC ratio be increased by other means?

In addition, if all of these ratios increase except rC/dC, why is there only an increased incorporation of rG in the heavy strand? Why aren't other rNTPs significantly increased in the heavy strand and why are none significantly increased in the light strand? Related to this, Xu et al., 2023 (NAR) previously showed using ribose-seq that rNMPs are incorporated into the light strand and is strongly biased toward rC (in human mtDNA). Are the authors able to explain the discrepancies in these results?

Connected to this question about incorporation, do you think this is why only fragments corresponding to the D-loop and Cytb are significantly increased in the cytosol (because you only observe a significant increase in rG incorporation in the heavy strand)?

4. The authors demonstrate that *Yme11*^{-/-} doesn't affect mtDNA stability (Extended Data Fig 6c). However, I'm confused as to why that would be the case. If *Yme11*^{-/-} results in increased rG incorporation, wouldn't that lead to mtDNA fragments and subsequent release? In which case, shouldn't you see a decrease in mtDNA stability relative to the control? Although decreased, mtDNA synthesis does appear to be happening in *Yme11*^{-/-} cells, so I would expect ribonucleotide incorporation in these (and throughout, related to point 3).

5. The data suggest that an imbalance nucleotide pool results in increased ribonucleotide incorporation which would then result in DSBs or other integrity issues. If that's the case, why are only fragments from the D-loop and Cytb significantly enriched

6. The authors demonstrate that senescent cells have increased ribonucleotide incorporation and ISG response. These data are sound, but wouldn't this indicate that senescent/terminally-differentiated cells are constantly releasing mtDNA and triggering cGAS-STING inflammatory pathways? Are there mechanisms in place that reduce/prevent this?

7. Is an increased nucleotide imbalance something that is observed to happen during aging (in WT conditions), leading to increases in this phenomenon (not necessarily just in senescent cells)? Is it possible to measure nucleotide pools from any samples previously collected from mice of various ages? I should note that I'm not asking for new samples to be collected as I don't believe it's reasonable for me to ask as collecting these from new mice up to 70 weeks sounds overly costly and time intensive and could be left to future studies.

Minor Comments

1. Throughout the manuscript, clarity about statistical tests can be improved. Figure legends sometimes indicate the test used, but it's not always apparent if that applies to all panels in the figure or just some. This can be fixed with changes in the figure legends.

2. Throughout, axes from a number of figures could be more clear. The information is always present in the figure legend, but it's preferable to be able to understand based solely on the panel alone. (e.g. 3a change "Fold change" to "Fold change in ISG expression, Extended Data Fig 1d change "log2 intensity" to "log2 intensity of TFAM levels" or something to that effect).

3. Figure 1c: Is it possible to visually indicate on Fig 1c that each column is a different mouse rather than needing to refer to the legend

4. Figure 1d: What is the cluster of proteins that are enriched in the cytosolic fraction but not mitochondrial? I'm assuming these are expected to be cytosolic but it would be helpful to indicate as much.

5. In the "Loss of MGME1 induces mtDNA-dependent cGAS-STING-TBK1 signaling" subsection, I believe "To investigate whether the ISG response in the absence of MGM1 depends on mtDNA" should be "... depends on mtDNA replication".

Referee #2

(Remarks to the Author)

In this manuscript, Bahat et al. describe an interesting and novel mechanism regulating mtDNA cytosolic release and innate immune signaling. More specifically, they use a mouse model lacking *Mgme1* (*Mgme1*^{-/-}), which is a mitochondrial exonuclease that regulates mtDNA maintenance. These mice display renal inflammation and die from renal failure at around 1 year of age. The authors show that innate immune signaling is increased in the kidneys of these mice in an age-dependent manner, which coincides with increased cytosolic leakage of mtDNA. Following this observation, the authors performed an in-depth analysis in *Mgme1*^{-/-} mouse embryonic fibroblast (MEFs) to investigate the mechanisms contributing to mtDNA release and subsequent activation of an innate immune response. They found that loss of MGME1 leads to an imbalance of nucleotides in the cell, such that deoxyribonucleotide triphosphates (dNTPs) were decreased and ribonucleotide monophosphates (rNMPs) increased, and this imbalance contributed directly to increased expression of interferon-stimulated genes (ISGs) in *Mgme1*-deficient cells. In addition, they found that there is an increased ratio of rNTP to dNTP in these cells, which contributes to increased incorporation of rNTP into mtDNA. The authors also demonstrated that senescent cells, which have increased mtDNA leakage and inflammation, also show higher incorporation of rNTP into their mtDNA. Interestingly, supplementation with exogenous deoxyribonucleotides could suppress the SASP of senescent cells. They conclude that an imbalance in nucleotides in cells leads to increased rNTP incorporation into mtDNA, which drives mtDNA leakage into the cytosol and a cGAS-STING-dependent activation of innate immune signaling.

These findings are novel and could be of high importance in the field, as elucidating mechanisms contributing to inflammation during aging and disease could reveal novel therapeutic targets. However, although the authors showed that this mechanism is present in *Mgme1*-deficient mice as they age, in my opinion, this study would be strengthened if the authors could show that this also happens during natural aging or in other age-related diseases in vivo. Can they demonstrate that increased rNTP/dNTP ratio and increased rNTP incorporation into mtDNA also happens in different tissues from naturally aged animals or in mouse models of age-related diseases?

In general, this study appears to be very well conducted, the rationale for each experiment is well explained and the

manuscript is easy to follow. The statistical analysis performed is also appropriate for the data presented. Below, I outline more specific comments and suggestions:

Figure 2:

- C) In figure 1D, they did not detect an increase in cytosolic mtDNA fragments containing regions further away from origin of heavy-strand replication (OH). However, in *Mgme1*^{-/-} MEFs, ND1 and COX3 are enriched in the cytosol, which is the opposite of what is seen in the kidneys in 1D. Can the authors explain the disparity between the two?
- D) Please, indicate in the image what each color represents (e.g. TOM20 in green, DNA in red). Also, the images seem to be very low resolution, so we cannot clearly distinguish the mitochondrial network. Do they have a better image to show? Lastly, there is no nuclear staining, so how was the quantification in (e) done? In other words, how were they able to tell what is one cell from these images?

Figure 3:

- B) The knockdown efficiency of Twnk is very low; however, there are still significant decreases in ISG expression. Are there any other components of the replication machinery that can be affected by manipulating levels of Twnk that could explain the effects on ISGs? Are protein levels of Twnk downregulated by this si? Show Western blot for Twnk.

Figure 4:

- D) This image should be shown in combination with a mitochondrial marker, such as TOM20, so that we can definitively say that the EdU-positive foci is of mitochondrial origin.

Figure 5:

- A) Given the higher levels of guanine ribonucleotide in the heavy strand, do the authors also see an increase in mtDNA breaks or mtDNA mutations in these cells?
- C) The difference between *Mgme1*^{+/+} and *Mgme1*^{-/-} with RNaseH2 treatment does not seem so striking as in *Yme1*^{-/-} cells shown in B. How can this be explained? Do *Yme1*^{-/-} cells have a higher content of rNTP in mtDNA? Can this comparison be made between the two different models? Also, the authors should show the mtDNA ribonucleotide content (as shown in A) for *Mgme1*^{-/-} cells too.

Figure 6:

- C and D) The differences seem very mild. Can the authors provide another representative blot that would show the differences more clearly?
- The authors show that dN supplementation decreases expression of some SASP factors. Does it also reduce mtDNA cytosolic leakage? Please, show these data.

Extended data Figure 3:

- D) Can the authors provide a quantification of cytosolic cGAS signal intensity (or loss of nuclear cGAS signal intensity) for these images? Otherwise, they can also measure cGAS activity in these conditions by an alternative way, such as using an ELISA assay for 2'3'-cGAMP. Moreover, it would also be interesting to show this (either cGAS staining or ELISA) in kidneys from *MGME1*^{-/-} mice.
- F) Did the authors analyze additional ISGs? It would be interesting to provide a more comprehensive list of ISGs affected by mtDNA depletion. Also, can the same experiment be done in *Mgme1*^{-/-} MEFs – i.e. can the authors treat these MEFs with ddC to deplete mtDNA and assess ISG expression? Is there a particular reason why they chose to si *Mgme1* rather than using *Mgme1*^{-/-} MEFs?
- F) Please, provide PCR data to show that mtDNA depletion was successfully achieved following ddC treatment in these cells.

Extended data Figure 5:

- A and B) Does knocking down these genes have an impact on cytosolic leakage of mtDNA? Please, show these data.

Extended data Figure 6:

- A) The authors show that 5-FU treatment, which inhibits cytosolic thymidylate synthase, increases the ratio of rU/dT. In the text they say “mtDNA-dependent innate immune signaling”... is associated with this increase. However, they do not show any data regarding the effects of 5-FU treatment on ISG expression and mtDNA cytosolic leakage. This is an important piece of data that would support their conclusions and should be included.

Extended data Figure 7:

- D) It is difficult to see mitochondrial EdU foci from these images. Please, enlarge or show magnification of an area next to the images. Also, can the authors provide some type of quantification of EdU co-localization with TOM20 in both conditions?

Referee #3

(Remarks to the Author)

Bahat et al, Nature

Ribonucleotide incorporation into mitochondrial DNA induces cGAS-STING dependent inflammation

In this manuscript, Bahat et al. report that increased ribonucleotide misincorporation into mitochondrial DNA triggers mtDNA release, cGAS-STING activation, and inflammatory SASP gene expression. Through a series of genetic and biochemical experiments, the authors elucidate a role for elevated incorporation of rNTPs into mtDNA during states of stress (*Yme1*^{-/-}, *Mgme1*^{-/-}, ionizing radiation-induced senescence, chemotherapeutic-induced senescence) that result in depletion of cellular dNTP availability. This leads to increased misincorporation of rNTPs into mtDNA and promotes instability and linearization observed in the *Mgme1*^{-/-} model. Furthermore, the authors show that loss of STING is sufficient to reduce interferon-stimulated gene signatures in aged *Mgme1*^{-/-} mouse kidneys. Although there are interesting and important aspects of the study, the paper lacks mechanistic depth in several key areas. As presented, it doesn't dramatically advance knowledge in the mtDNA-cGAS-STING field beyond what other key papers have shown (including prior work from the authors

themselves). Therefore, it is unclear that the paper represents enough of an advance to warrant publication in Nature.

Major points:

1. This manuscript leverages two mouse models that exhibit aberrant mtDNA phenotypes that contribute to pathology. *Mgme1*^{-/-} (PMID: 29572490) and *Yme1*^{-/-} (PMID: 33903774) both exhibit increased mtDNA instability and elevated linearized mtDNA. The *Mgme1*^{-/-} was previously shown to exhibit elevated plasma cytokines including interferon gamma, as well as late-developing inflammatory kidney diseases characterized by lymphocyte infiltration. Previous work from the Langer group has also shown a nucleotide imbalance in the *Yme1*^{-/-} and reported a role for mtDNA release and cGAS-STING activation in null MEFs. Thus, the first 3 figures of the paper do not provide much of a mechanistic advance and perhaps miss opportunities to document the relevance of this pathway for disease. The authors report that the DKO mouse lacking STING and MGME1 does not exhibit late-onset kidney inflammation and interferon-stimulated gene expression. However, the paper does not provide data examining how ablation of the inflammatory signature impacts disease phenotypes. Do DKO mice have improved kidney function or improved retina morphology? Does STING ablation reduce T cell infiltrates into *Mgme1*^{-/-} KO mouse kidneys? Do DKO mice have reduced circulating cytokines in plasma relative to *Mgme1*^{-/-} alone? Does STING ablation extend lifespan of *Mgme1*^{-/-} mice? In the absence of some functional phenotypic rescue, it is unclear whether the activation of cGAS in this model contributes to disease. Recently, cGAS ablation was shown to reduce interferon responses in an RNASEH2 mutant Aicardi-Goutières syndrome (AGS) mouse model, but not improve cerebellar defects observed (PMID: 34655526), suggesting that the IFN response is not directly responsible for neurodegeneration. Thus, it is critical that the authors reveal more phenotypic data from the DKO mouse to better solidify links between mtDNA-mediated inflammation and disease.

2. Why does MGME1 deficiency only result in mtDNA release and cGAS-STING activation in the kidney? This is a very interesting finding, but the paper fails to examine the mechanisms behind this. Are other body organs more resistant to rNTP incorporation into mtDNA? Are there RNASEH-like repair mechanisms in certain organs that are absent in the kidney? At the very least, the authors need to examine cytosolic mtDNA and ribonucleotide incorporation into mtDNA from other tissues as was done in Figs. 1,5 to show specificity for the kidney. Moreover, it is unclear where the cGAS-STING pathway is activated in the *Mgme1*^{-/-} KO kidneys. Is there a particular cell type that is susceptible to ribonucleotide incorporation and cGAS-STING activation?

3. RNA-DNA hybrids are known ligands for cGAS. Does mtDNA containing ribonucleotides serve as a more potent ligand for cGAS, or is it simply that dNTP imbalance and ribonucleotide misincorporation leads to increased mtDNA release and cGAS binding? More information here would increase understanding and broaden key findings of the paper.

4. The senescence data in Fig. 6 are interesting, but still preliminary. The identification that ribonucleotide incorporation into mtDNA underlies SASP is quite interesting, but it is unclear how translatable these findings are to the rest of the study. For example, it is unclear why kidney cell lines weren't utilized here to tie into the mouse inflammatory pathology. Do the authors see increased numbers of senescent cells in aged MGME1 KO kidneys, and if so, are these cells the sources of cGAS-STING activation? Does ablation of STING in MGME1 KO kidneys impact senescence phenotypes, if any?

Other questions/concerns:

1. Extended data Fig 3 suggests that knocking down STING or cGAS in immortalized MGME1 KO MEFs reduces ISG expression. While this is clear, knock down does not entirely ablate ISG expression as is shown in the DKO kidneys. Is this due to a limited efficiency of knock down in the *Mgme1*^{-/-} cells or are there other sensors that work with cGAS or perhaps even directly through STING to drive IFN in this model? RNA sensors don't seem to be active, but is there any role for other cytosolic DNA sensors (IFI200/IFI16 family members, for example)?

2. In Figure 3 the authors argue that depletion of *Twinkle*, *Samhd1*, or *Slc25a33* suppresses the ISG response in *Mgme1*^{-/-} MEFs. The authors should examine and quantify cytosolic mtDNA in all of these double mutant conditions.

3. The confocal microscopy is not sufficient as displayed. While the quantitation of some images is generally convincing, the quality of the representative images and lack of global quantification reduce enthusiasm for these findings. Specifically in Figure 2d, the image provided needs to be enlarged with labels for the different stains and their colors in the figure. This image could also be improved by including a zoom. In extended data Fig. 3d, the images provided do not convincingly support the conclusion that cGAS has a cytosolic localization in the *Mgme1*^{-/-} MEFs. In the images provided the *Mgme1*^{-/-} MEFs seem to still have a large amount of cGAS in the nucleus and the cytosolic portion is difficult to discern. This could be improved by including a whole cell marker and quantitating cGAS intensity in the nucleus vs cytoplasm based on these markers. In figure 4d, a zoomed image and inclusion of mitochondrial and whole cell markers would improve confidence that the spots labeled are freshly synthesized mtDNA. In extended data Fig. 7, please provide a zoom image and quantitation to prove less new mtDNA colocalizing with TOMM20.

4. In extended data Fig. 3f the authors use ddC in MEFs to conclude that loss of mtDNA inhibited immune signaling however they did not prove that they depleted mtDNA and the results shown do not reach significance. Can this be expanded on? Additionally, this experiment was done using MGME1 knock down in WT MEFs and should be repeated in the *Mgme1*^{-/-} MEFs.

5. Data from extended data Fig. 5 is used to conclude an increase in rNMPs in *Mgme1*^{-/-}, however these findings do not appear to reach statistical significance.

6. More thorough labeling of figures would improve readability of the manuscript. For example, clearly delineating “cytosolic kidney extracts” in Figure 1d and “whole cell kidney extracts in extended data Fig. 1e.
7. “Each column represents a different mouse” is repeated twice in the legend of extended data Fig. 1 a.
8. In extended data Fig. 2 b, there is a discrepancy between 18s rDNA shown in the figure and 18s rRNA mentioned in the figure legend.
9. The text describing Fig 1d is out of place and should be moved later in the manuscript to improve readability.

Version 1:

Reviewer comments:

Referee #1

(Remarks to the Author)

In their revised manuscript, the authors have addressed my concerns. The revised version has improved, and the addition of new data further strengthens their findings and model. I was particularly excited to see the increase in rNTP/dNTP ratio in aged mice as well as the increase sensitivity to alkaline/RNaseH2 treatment. I think this study will be of great interest to the readership. As a very minor point, I might suggest labeling the tissue in Figures 6a, 6c, ED10a, and ED10c (even though it is indicated on 6b/6d/ED10b/ED10d.

Referee #2

(Remarks to the Author)

The authors have addressed my concerns, and I recommend that the manuscript is acceptable for publication.

Referee #3

(Remarks to the Author)

In their revised manuscript, Bahat, Milenkovic et al. expand their findings to show that loss of MGME1 promotes increased mtDNA replication initiation. This depletes mitochondrial deoxynucleotide pools and increases ribonucleotide incorporation into mtDNA, causing mtDNA release and activation of the cGAS-STING pathway. Interestingly, the sequences that accumulate in the cytosol of cells and kidneys of MGME1^{-/-} mice are proximal to O_H. The inclusion of next generation sequencing and additional tissue analysis has strengthened the study. However, key open questions remain that warrant additional clarification before this study is suitable for publication in Nature.

1. I thank the authors for providing more data on the kidney immune and pathological phenotypes in DKO mice. They clearly show a role for STING in both kidney pathology and infiltration of T and B cells at late timepoints. The overall mechanism presented is that mtDNA release, presumably in Mgme1^{-/-} kidney cells, drives cGAS-STING signaling, leading to a cell-intrinsic activation of immunity/inflammation that causes kidney failure. Experiments from MEFs show that STING knockdown/TBK1 inhibition is much more potent at blocking interferon genes than cGAS knockdown. STING can integrate many signals from nucleic acid sensors and mediates many responses beyond interferon signaling, including cell death, which can occur in a cGAS-independent pathway in vivo. STING activation, in both cGAS-dependent and -independent mechanisms can also alter T cell activity in vivo, which may compound the kidney injury of Mgme1^{-/-} mice. Given that no data are provided to untangle where the interferon/immune response initiates in Mgme1^{-/-} kidneys, and whether the interferon signature initiates from resident kidney cells or infiltrating immune cells (i.e. via signal cell approaches), evidence for in vivo cGAS-dependency are limited. In the absence of more conclusive data, the authors should remove cGAS from the title and focus the story on STING.

2. The mechanisms explaining why inflammatory signatures are only seen in the kidney of aged Mgme1^{-/-} mice remain mostly unaddressed. Why does it take over a year for the immune response to be detectable in the kidney, when increased mtDNA replication and ribonucleotide incorporation is likely happening from early life onward? Do young Mgme1^{-/-} mice have increased mtDNA in kidney extracts, and if so, why is this not sufficient to activate the STING pathway? Could it be that Mgme1 dependent mitochondrial deficits compromise the tissue over time, leading to immune infiltrates that express interferon genes at high levels and further damage the kidney? Moreover, the increase in ribonucleotide incorporation and cytosolic mtDNA in liver is interesting, especially because no immune response is seen in these tissues. The resident immune profile of the kidney is quite different from the liver, and perhaps this is responsible for differences in interferon and inflammatory gene activation. However, in their rebuttal letter, the authors discuss potential thresholding effects and propose Trex1 might be higher in non-kidney tissues to control sensing of mtDNA. But if this were the case, higher Trex1 would result in reduced mtDNA in Mgme1^{-/-} liver, and the authors report significantly increased cytosolic mtDNA there. I feel that the absence of molecular and cellular mechanisms here decrease the novelty of the story, because much of what the authors report in the paper has already been documented in other published studies (i.e. mtDNA replication defects in Mgme1^{-/-}, mitochondrial nucleotide pools impacting ribonucleotide incorporation in mtDNA, inflammation in Mgme1^{-/-} kidneys). The tissue specificity of this model is interesting and a better understanding could help explain how cytosolic mtDNA sensing

specifically drives immune activation in the kidney, which the authors note is a frequent target in autoimmune and auto-inflammatory disorders.

3. Regarding the senescence aspects of the study, the authors do not provide any data documenting that cGAS-STING pathway is required for senescence or the immune/SASP gene signature seen. No genetic experiments were done to show that ablation of STING inhibits senescence phenotypes or SASP. The data do support that increased nucleotide pool imbalance and ribonucleotide incorporation into mtDNA does occur in senescent cells in vitro, and that this is correlated with increased mtDNA release. dN supplementation brings down expression of some SASP genes (IL1B, etc), but curiously the authors did not examine interferon-induced genes as in Figs 1-2, or show that the dN-dependent decrease in SASP genes is dependent on reduced cGAS, STING signaling. In figure R3, the expression of senescence marker genes is not dependent on STING, which argues that the senescence and mtDNA-STING phenotypes may be separate. The authors argue that they have identified rNTP incorporation into mtDNA as a central mechanism leading to release and STING activation in both their genetic models, in vitro senescence, and in vivo aging, yet the data do not convincingly show this to be the case. They also did not examine abundance of senescent cells in the kidney of *Mgme1*^{-/-}, or test whether the SASP is what recruits T cells into the kidney. No evidence was provided elevated ribonucleotide incorporation into the aged heart and liver mtDNA is linked to increased STING activation. Overall, the last two figures of the paper are critical in extending the findings beyond the *Mgme1* model, but the data are largely correlative. It is this unclear whether increased rNTP incorporation into mtDNA is truly the unifying mechanism leading to mtDNA release, STING activation, and senescence.

4. The authors provide evidence in the response that other sensors may be involved in sensing ribonucleotide incorporation and mtDNA release in the *Mgme1*^{-/-} model. These data are compelling and suggest a more complex immune complex immune mechanisms than the linear cGAS-STING pathway nominated in in the manuscript. This partial rescue of kidney pathology in the DKO implicates other immunological mechanisms beyond STING, so I believe the authors must include these data for all readers and discuss them in the paper.

Point-by-point response to the Referees' comments

Reply to Referee #1:

Summary

Bahat et al. provide a compelling, clear and well-written manuscript investigating how increased ribonucleotide incorporation into mtDNA leads to increased cytosolic mtDNA, cGAS-STING activation, and inflammatory response. While previous work has demonstrated that mtDNA release into the cytoplasm through various pathways leads to this inflammatory response, this work is novel in that it rigorously demonstrates one mechanism by which this is triggered – imbalanced nucleotide pools and subsequent ribonucleotide incorporation during mtDNA replication. In particular, this is demonstrated using both mouse models and cell culture through multiple approaches, and further supported by the amelioration of these phenotypes by increasing deoxyribonucleotides. Their data support a model by which an imbalance in nucleotide pools leads to increased ribonucleotide incorporation. There are a few results that, based on my understanding, are not fully explained by their model and aren't adequately addressed in the text. Although, I believe many of these can be addressed by discussion and clarity in the text. Overall, this is a strong manuscript that will be of interest to many fields.

Major Comments

1. Figure 1d shows an increase in cytosolic mtDNA, specifically with regions proximal to the heavy strand origin of replication, consistent with these being non-productive replication intermediates that have been released. The identity of these fragments (e.g. the D-loop and Cytb) is consistent with this being a replication product. In which case, do you see the cytosolic presence of fragments derived from the light strand origin of replication (e.g. Cox1)?

Mgme1^{-/-} mice show a tissue-specific replication stalling phenotype. Previous sequencing of brain and heart mtDNA (Matic et al., 2018; PMID: 29572490) revealed frequent replication initiation at O_H and premature termination, resulting in accumulation of replication intermediates proximal to O_H and lower sequence coverage further away from O_H. This is consistent with our observation that sequences proximal to O_H are enriched in cytosolic fraction, whereas we do not observe an enrichment of DNA fragments proximal to O_L.

To extend these findings and directly address the reviewer's question, we performed next generation sequencing of mtDNA from kidney of *Mgme1*^{-/-} mice (new Fig. 3c). As observed before in brain and heart tissues of these mice, we observed a decline in the sequence coverage with increasing distance from O_H, which is consistent with the increased detection of fragments proximal to O_H in cytosolic mtDNA (new Fig. 1c). Notably, the number of reads in proximity to O_H in *Mgme1*^{-/-} kidneys significantly exceeded those in wild type animals. This suggests significantly increased unproductive replication events in *Mgme1*^{-/-} kidneys that lead to nucleotide exhaustion and explain the observed depletion of nucleotides in the absence of MGME1. Since this was not observed in brain and muscular tissue (Matic et al., 2018; PMID: 29572490), this may also explain the vulnerability of kidney tissue in this mouse model.

We also performed next generation sequencing of *Yme1*^{-/-} MEFs. In contrast to *Mgme1*^{-/-} kidneys, we observed a similar number of reads in the presence and absence of YME1L. This is consistent with our previous results that loss of YME1L causes a nucleotide imbalance by stabilization of the mitochondrial nucleotide carrier SLC25A33 and by affecting nucleotide synthesis (Sprenger et al., Nat. Metab., 2021; PMID: 33903774). Thus, the absence of YME1L impairs nucleotide synthesis whereas the loss of MGME1 likely increases nucleotide

consumption (by increased, unproductive mtDNA replication events), both resulting in an increased rNTP/dNTP ratio. Since the sequencing results for *Yme1*^{-/-} cells are negative and not relevant for any conclusion, we did not include them in the manuscript but only show them to the reviewers in Figure R1b.

In addition, do you know if the cytosolic fragments are single-stranded or double-stranded? My understanding was that cGAS-STING is primarily activated by dsDNA. However, the identity of these fragments from nucleotide imbalance as ssDNA or dsDNA would suggest potentially distinct mechanisms for their generation and release. Previously published results about mtDNA release into the cytoplasm would suggest these are dsDNA molecules.

We agree with the reviewer that cytoplasmic mtDNA fragments most likely are dsDNA. We speculate that base pairing of single-stranded D-loops or of other replication intermediates leads to the formation of dsDNA (at least 18 bp) that is recognized by cGAS (Kim et al., Exp. Mol. Med., 2023; PMID: 36964253).

2. Figure 3b demonstrates that depletion of Twnk suppresses the ISG response observed in Mgme1-/- immortalized MEFs. The authors argue that this could be due to ongoing mtDNA synthesis being necessary to trigger mtDNA release. However, it is not clear to me why this wouldn't be observed with depletion of other replisome components. Is this something you have tested and is this suppression unique to Twnk?

We also have depleted SSBP1 and POLG2 from wild type and *Mgme1*^{-/-} cells. Similar to TWINKLE loss, the depletion of SSBP1 decreases ISG expression in *Mgme1*^{-/-} cells but does not induce an ISG response in WT cells. These results further confirm that mtDNA replication is required to induce ISG expression and are now shown in the new Extended Data Fig. 6b.

3. In Figure 4b, the authors demonstrate that the rA/dA, rG/dG, and rU/dT ratios increase in Mgme1-/- MEFs (although all 4 increase in Sen (IR)). Can the authors speculate why the rC/dC ratio decreases? Can the rC/dC ratio be increased by other means?

Our present and previous experiments (Sprenger et al., Nat. Metab., 2021; PMID: 33903774) using different genetic models and pharmacological treatments suggest a critical role of nucleotide imbalance rather than depletion of specific nucleotides for mtDNA release. Consistently, interference with nucleotide hydrolases, such as SAMHD1, or the pyrimidine carrier SLC25A33, both of which have broad nucleotide specificity, modulate the inflammatory response. Given the complexity of the nucleotide metabolism (*de novo* synthesis, salvage pathways) and the plethora of nucleotide hydrolases, specific changes in the nucleotide levels are likely to be cell specific. In addition, nucleotide hydrolases are also part of the ISG response and can affect nucleotide levels. For example, the expression of RSAD2 (viperin), a CTP hydrolyzing antiviral enzyme, is induced upon mtDNA release and may influence the rC/dC ratio at steady state.

In addition, if all of these ratios increase except rC/dC, why is there only an increased incorporation of rG in the heavy strand? Why aren't other rNTPs significantly increased in the heavy strand and why are none significantly increased in the light strand? Related to this, Xu et al., 2023 (NAR) previously showed using ribose-seq that rNMPs are incorporated into the light strand and is strongly biased

toward rC (in human mtDNA). Are the authors able to explain the discrepancies in these results?

The incorporation of ribonucleotides into mtDNA depends on the amount of nucleotides available, but also on the substrate specificity of the enzymes involved. The mitochondrial DNA polymerase γ was found to be less efficient in discriminating between rCTP and rGTP when compared to rATP and rUTP, which likely contributes to the observed binding of ribonucleotide incorporation into the GC-rich heavy strand (Kasiviswanathan and Copeland, J. Biol. Chem., 2011; PMID: 21778232). Notably, a similar ribonucleotide pattern was observed in the mtDNA of patient fibroblasts carrying mutation in DGUOK or MPV17 (Berglund et al., PLoS Genet., 2017; PMID: 28207748; Moss et al., NAR, 2017; PMID:29106596).

Connected to this question about incorporation, do you think this is why only fragments corresponding to the D-loop and Cytb are significantly increased in the cytosol (because you only observe a significant increase in rG incorporation in the heavy strand)?

Previous work has shown that ribonucleotides can be incorporated into mtDNA in a sequence- and region-unspecific manner (Berglund et al., PLoS Genet., 2017; PMID: 28207748; Forslund et al., PLoS Genet., 2018; PMID:29601571). As discussed in response to point 1, increased ribonucleotide incorporation combined with the replication stalling and the increased number of unproductive replication initiation events explains the enrichment of D-loop and CytB regions in the cytosol in the absence of MGME1.

4. The authors demonstrate that Yme1^{-/-} doesn't affect mtDNA stability (Extended Data Fig 6c). However, I'm confused as to why that would be the case. If Yme1^{-/-} results in increased rG incorporation, wouldn't that lead to mtDNA fragments and subsequent release? In which case, shouldn't you see a decrease in mtDNA stability relative to the control? Although decreased, mtDNA synthesis does appear to be happening in Yme1^{-/-} cells, so I would expect ribonucleotide incorporation in these (and throughout, related to point 3).

Ribonucleotide incorporation into mtDNA slows down replication and, in the case of *Mgme1^{-/-}* cells, has been shown to culminate in replication stalling (see also reply to point 1). Replication intermediates are susceptible to degradation resulting in the release of mtDNA fragments from the mitochondria. In the *Yme1^{-/-}* cells, we have reported slower replication but could not detect replication stalling and mtDNA depletion (see pair end sequencing pattern in wt vs *Yme1^{-/-}* (Figure R1c)). It should be noted that mtDNA release and inflammation has been observed in cells both with decreased or unaltered/increased mtDNA levels. In addition, mtDNA nucleoids or released from mitochondria in heterozygous *Tfam^{+/-}* cells (rather than mtDNA fragments as observed in *Yme1^{-/-}* and *Mgme1^{-/-}* cells), which have decreased mtDNA levels (50%).

5. The data suggest that an imbalance nucleotide pool results in increased ribonucleotide incorporation which would then result in DSBs or other integrity issues. If that's the case, why are only fragments from the D-loop and Cytb significantly enriched

Please see our reply to point 1 and our new next generation sequencing data. Loss of MGME1 increases replication initiation and replication stalling, leading to the accumulation of mtDNA fragments in proximity to O_H.

6. The authors demonstrate that senescent cells have increased ribonucleotide incorporation and ISG response. These data are sound, but wouldn't this indicate that senescent/terminally-differentiated cells are constantly releasing mtDNA and triggering cGAS-STING inflammatory pathways? Are there mechanisms in place that reduce/prevent this?

Cell cycle arrested senescent cells are characterized by arrested nuclear DNA replication but ongoing mtDNA replication. Ribonucleotide reductase is inhibited in senescent cells causing an increased rNTP/dNTP ratio, ribonucleotide incorporation into mtDNA, the release of mtDNA fragments and inflammation. The extent to which this applies in general to postmitotic cells, remains to be investigated and may depend on cell-specific regulation of nucleotide synthesis pathways and defence mechanisms (for example, see also a very recent publication from the Chabes/Wanrooj labs (Awoyomi et al., 2025; PMID: 40244665)). We have also previously shown that the DNase TREX1 limits the cytosolic accumulation of mtDNA fragments and cGAS-STING signalling in *Yme1^{-/-}* cells (Sprenger et al., Nat. Metab., 2021; PMID: 33903774). Of note, expression of TREX1 increases with age and mutations in TREX1 cause senescence in *Drosophila* (Chauvin et al., Nat. Comm. 2024; PMID: 38824133) and inflammatory diseases (Aicardi-Goutières syndrome, Systemic Lupus erythematosus).

7. Is an increased nucleotide imbalance something that is observed to happen during aging (in WT conditions), leading to increases in this phenomenon (not necessarily just in senescent cells)? Is it possible to measure nucleotide pools from any samples previously collected from mice of various ages? I should note that I'm not asking for new samples to be collected as I don't believe it's reasonable for me to ask as collecting these from new mice up to 70 weeks sounds overly costly and time intensive and could be left to future studies.

To investigate the relevance of rNTP into mtDNA during aging, we have isolated mtDNA from different tissues (kidney, liver, heart and spleen) of old (87 weeks) and young (1 week) mice and monitored ribonucleotide incorporation into mtDNA after Southern blot on alkaline gels. We were excited to observe a significantly increased rNTP incorporation into mtDNA in the aged tissues, as now shown in the new Fig. 6 and new Extended Data Fig. 10. We also observed an increased rNTP/NTP ratio in these aged tissues (Fig. 6 and Extended Data Fig. 10). These data are consistent with previous results in muscular tissues (Wanrooj et al., PNAS, 2020; PMID:32513727) and demonstrate that nucleotide imbalance and increased rNTP incorporation into mtDNA occurs in different tissues during natural aging. Therefore, we conclude that perturbations in nucleotide metabolism represent a challenge to cellular defence mechanisms that increases with age and may contribute to or even drive age-associated inflammatory responses.

Minor Comments

1. Throughout the manuscript, clarity about statistical tests can be improved. Figure legends sometimes indicate the test used, but it's not always apparent if that applies to all panels in the figure or just some. This can be fixed with changes in the figure legends.

We have corrected the figure legends to comply with the reviewer's request.

2. Throughout, axes from a number of figures could be more clear. The information is always present in the figure legend, but it's preferable to be able to understand based solely on the panel alone. (e.g. 3a change "Fold change" to "Fold change in ISG expression, Extended Data Fig 1d change "log2

intensity” to “log2 intensity of TFAM levels” or something to that effect).

We have improved the labelling of the figures.

3. Figure 1c: Is it possible to visually indicate on Fig 1c that each column is a different mouse rather than needing to refer to the legend

We have numbered the individual columns in former Fig. 1c (now Extended Data Fig. 1d) to clarify that they refer to individual mice.

4. Figure 1d: What is the cluster of proteins that are enriched in the cytosolic fraction but not mitochondrial? I’m assuming these are expected to be cytosolic but it would be helpful to indicate as much.

While the vast majority of mitochondrial proteins (according to MitoCarta3.0) are present in the mitochondrial fraction as expected, a number of mitochondrial proteins appear to be dually localized and also present in the cytosol. This includes some known proteins (for example STARD7) but also a number of proteins that have not been described to be also cytosolic. This is an interesting observation on its own deserving further investigation, which is however beyond the scope of the present study. We include below a table showing mitochondrial proteins that routinely are also found in the cytosolic fraction (see also our previous publication Deshwal et al., Nat Cell Biol., 2023; PMID: 36658222).

Rmdn1	Qdpr	Kars1	Plpbb	Park7	Pdhab
Acot7	Prdx6	Prdx2	Trnt1	Aldh9a1	Echdc3
Glod4	Dcxr	Nudt5	Msra	Akr1b10	Ak2
Rida	Aldh1l1	Paics	Lap3	Elac2	Echdc1
Gstz1	Ldhab	Osbpl1a	Flad1	Lypla1	Cmpk2
Gars1	Grhpr	Casp8	Nit1	Comt	Gpx1
Gsr	Acaca	Tomm34	Suox	Nit2	Oxnad1
Cryz	Fasn	Dnm1l	Hagh	Prdx5	Rpia
Txnrd1	Acly	Fth1	Akr7a2	Aldh7a1	Nln

Table R1. List of mitochondrial proteins (according to MitoCarta3.0) enriched in the cytosolic fraction in the experiment from Extended Data Fig. 1d.

5. In the “Loss of MGME1 induces mtDNA-dependent cGAS-STING-TBK1 signaling” subsection, I believe “To investigate whether the ISG response in the absence of MGM1 depends on mtDNA” should be “... depends on mtDNA replication”.

We have corrected this mistake in the text.

Reply to Referee #2:

In this manuscript, Bahat et al. describe an interesting and novel mechanism regulating mtDNA cytosolic release and innate immune signaling. More specifically, they use a mouse model lacking Mgme1 (Mgme1^{-/-}), which is a mitochondrial exonuclease that regulates mtDNA maintenance. These mice display renal inflammation and die from renal failure at around 1 year of age. The authors show that innate immune signaling is increased in the kidneys of these mice in an age-dependent manner, which coincides with increased cytosolic leakage of mtDNA. Following this observation, the authors performed an in-depth analysis in Mgme1^{-/-} mouse embryonic fibroblast (MEFs) to investigate the mechanisms contributing to mtDNA release and subsequent activation of an innate immune response. They found that loss of MGME1 leads to an imbalance of nucleotides in the cell, such that deoxyribonucleotide triphosphates (dNTPs) were decreased and ribonucleotide monophosphates (rNMPs) increased, and this imbalance contributed directly to increased expression of interferon-stimulated genes (ISGs) in Mgme1-deficient cells. In addition, they found that there is an increased ratio of rNTP to dNTP in these cells, which contributes to increased incorporation of rNTP into mtDNA. The authors also demonstrated that senescent cells, which have increased mtDNA leakage and inflammation, also show higher incorporation of rNTP into their mtDNA. Interestingly, supplementation with exogenous deoxyribonucleotides could suppress the SASP of senescent cells. They conclude that an imbalance in nucleotides in cells leads to increased rNTP incorporation into mtDNA, which drives mtDNA leakage into the cytosol and a cGAS-STING-dependent activation of innate immune signaling.

These findings are novel and could be of high importance in the field, as elucidating mechanisms contributing to inflammation during aging and disease could reveal novel therapeutic targets. However, although the authors showed that this mechanism is present in Mgme1-deficient mice as they age, in my opinion, this study would be strengthened if the authors could show that this also happens during natural aging or in other age-related diseases in vivo. Can they demonstrate that increased rNTP/dNTP ratio and increased rNTP incorporation into mtDNA also happens in different tissues from naturally aged animals or in mouse models of age-related diseases?

As described in more detail in response to point 7 of Reviewer 1, we have analyzed nucleotide pools and rNTP incorporation into mtDNA in several tissues of young and aged mice (new Fig. 6 and new Extended Data Fig. 10). Our results unambiguously demonstrate that rNTP incorporation occurs in naturally aged animals, significantly strengthening the significance of our findings.

In general, this study appears to be very well conducted, the rationale for each experiment is well explained and the manuscript is easy to follow. The statistical analysis performed is also appropriate for the data presented. Below, I outline more specific comments and suggestions:

Figure 2:

- C) In figure 1D, they did not detect an increase in cytosolic mtDNA fragments containing regions further away from origin of heavy-strand replication (OH). However, in Mgme1^{-/-} MEFs, ND1 and COX3 are enriched in the cytosol, which is the opposite of what is seen in the kidneys in 1D. Can the authors explain the disparity between the two?

We always observe mtDNA fragments containing the D-loop and CytB regions being strongest enriched in the cytosolic fraction, while the fragments more distant to O_H accumulate at lower levels or are not enriched at a detectable level. As discussed in detail in response to point 1 of Reviewer 1, this is explained by increased replication initiation and replication stalling in Mgme1^{-/-} cells. However, we are aware of the apparent differences

with respect to COX1 and ND1 containing mtDNA fragments in the experiments shown in Fig. 1c (kidney; formerly 1d) and Fig. 2b (MEFs; formerly 2c). A possible explanation could be differences between fast proliferating, immortalized MEFs and post-mitotic kidney, which are difficult to compare. Even more, kidney is a complex organ that contains 26 different cell types (Schumacher et al., Nat. Reg. Med., 2021; PMID: 34381054). Scattered OXPHOS deficient cells were found in kidneys of one-year-old *Mgme1*^{-/-} mice showing that not all cells are equally affected by the loss of MGME1 (Milenkovic et al., PLoS Gen., 2022; PMID: 35533204).

- D) Please, indicate in the image what each color represents (e.g. TOM20 in green, DNA in red). Also, the images seem to be very low resolution, so we cannot clearly distinguish the mitochondrial network. Do they have a better image to show? Lastly, there is no nuclear staining, so how was the quantification in (e) done?

We apologize for the insufficient quality of Fig. 2c (formerly Fig. 2d), which we have significantly improved in the revised manuscript. We have also included nuclear staining and the description of quantification.

Figure 3:

- B) The knockdown efficiency of *Twnk* is very low; however, there are still significant decreases in ISG expression. Are there any other components of the replication machinery that can be affected by manipulating levels of *Twnk* that could explain the effects on ISGs? Are protein levels of *Twnk* downregulated by this si? Show Western blot for *Twnk*.

We have included a Western blot for TWINKLE to monitor the knock-down efficiency (new Extended Data Fig. 6c). Previous transcriptomic and proteomic analyses of mouse hearts of lacking *Twnk* did not reveal changes in the levels of other mtDNA replication components (Kühl et al., eLife, 2017; PMID: 29132502). In addition, we have now depleted other essential component of the mtDNA replication machinery, SSBP1 and POLG2. Similar to TWINKLE depletion, SSBP1 or POLG2 depletion impairs ISG expression in *Mgme1*^{-/-} cells, but does not affect the inflammatory response in WT cells (new Extended Data Fig. 6b). These results support our conclusion that mtDNA replication is required to induce an ISG response in *Mgme1*^{-/-} cells.

Figure 4:

- D) This image should be shown in combination with a mitochondrial marker, such as TOM20, so that we can definitively say that the EdU-positive foci is of mitochondrial origin.

We have improved the images in Extended Figure 8a (previously Fig. 4d) and included TOMM20 as a mitochondrial marker. The figure shows unambiguously that the EdU-positive foci are of mitochondrial origin.

Figure 5:

- A) Given the higher levels of guanine ribonucleotide in the heavy strand, do the authors also see an increase in mtDNA breaks or mtDNA mutations in these cells?

We have analyzed the mutation load and mtDNA integrity of mtDNA in *Yme1^{-/-}* cells using Southern blot analysis, PCR-based cloning and sequencing of the WANCY region of mtDNA to score for the mutations and next generation sequencing, but could not detect increased mtDNA mutations or any aberrant mtDNA species (Fig. R1a, b). It should be noted that mtDNA breaks are generally difficult to detect, because they induce rapid mtDNA degradation. This is illustrated in Reviewer Fig. 1, which shows rapid degradation of mtDNA after inducing mtDNA breaks by expressing the restriction enzyme *EagII* in mitochondria (Fig. R1c).

Figure R1. mtDNA integrity in *Yme1^{-/-}* mitochondria.

a, Quantitative assessment of mtDNA point mutations in *Yme1^{-/-}* MEFs. Post-PCR cloning and sequencing was used to quantify mtDNA mutation load. **b**, Sequence coverage of the mouse mtDNA samples from MEFs of *Yme1^{-/-}* and controls (*Yme1^{+/+}*). Mitochondrial genome position (x-axis) versus sequence coverage divided by the total coverage for each sample. For each genotype two samples were analysed. The approximate locations of the origins of light-strand (O_L) and heavy-strand (O_H) replication are indicated by dotted lines with arrows. **c**, Southern blot analysis of mtDNA from MEFs expressing tetracycline (Tet)-inducible, mitochondrially targeted restriction enzyme *EagII*. MEFs were additionally transfected with control (*EGFP*) and siRNA against *Twnk*. Cox1 radiolabelled probe was used to visualize mtDNA. DNA for 18S rRNA serves as a loading control.

- C) The difference between *Mgme1*^{+/+} and *Mgme1*^{-/-} with RNaseH2 treatment does not seem so striking as in *Yme1*^{-/-} cells shown in B. How can this be explained? Do *Yme1*^{-/-} cells have a higher content of rNTP in mtDNA? Can this comparison be made between the two different models? Also, the authors should show the mtDNA ribonucleotide content (as shown in A) for *Mgme1*^{-/-} cells too.

A quantitative comparison of the two models based on alkaline gels is difficult, because fast proliferating MEFs in Fig. 4d (formerly Fig. 5b) and post-mitotic kidney in Fig. 4f (formerly Fig. 5c) have been analyzed. As discussed in response to the comments on Fig. 2c, it is likely that the loss of MGME1 has different effects on different kidney cell types. Therefore, the analysis of the kidney shows the average rNTP incorporation into mtDNA of different cells.

As requested from the reviewer, we have performed Hyden-seq of *Mgme1*^{-/-} cells (new Figure 4e). In agreement with our findings in *Yme1*^{-/-} cells (Fig. 4e; formerly Fig. 5a), we observed an increased incorporation of rG and rC into heavy strand of the mtDNA in *Mgme1*^{-/-} cells, further substantiating our conclusions. We show now the results of Hyden-seq and an alkaline gel analysis for both models in Figure 4c-f.

Figure 6:

- C and D) The differences seem very mild. Can the authors provide another representative blot that would show the differences more clearly?

To comply with the request of the reviewer, we provide now another representative blot of the alkaline gel analysis of senescent cells with quantification in Figure R2. In our opinion, both figures (Figure R2 and Figure 5d in the manuscript) clearly show an increased rNTP incorporation into mtDNA of senescent cells, which is partially suppressed upon nucleoside complementation.

Figure R2. Increased ribonucleotide incorporation into mtDNA from senescent cells. Untreated and RNaseH2-treated mtDNA isolated from proliferating and irradiation-induced senescent IMR90 cells, which were supplemented with all four deoxyribonucleosides where indicated, was resolved on alkaline agarose gels and further analyzed by Southern blot and quantified.

- The authors show that dN supplementation decreases expression of some SASP factors. Does it also reduce mtDNA cytosolic leakage? Please, show these data.

We have performed cell fractionation experiments to monitor the accumulation of mtDNA in the cytosol of senescent cells and of senescent cells supplemented by nucleosides. As shown in the new Fig. 5c and new Extended Data Fig. 4c, mtDNA detected with probes for D-loop accumulated in the cytosol of senescent cells. Nucleoside complementation significantly decreases cytosolic mtDNA, consistent with the observed suppression of ISG expression.

Extended data Figure 3:

- D) Can the authors provide a quantification of cytosolic cGAS signal intensity (or loss of nuclear cGAS signal intensity) for these images? Otherwise, they can also measure cGAS activity in these conditions by an alternative way, such as using an ELISA assay for 2'3'-cGAMP. Moreover, it would also be interesting to show this (either cGAS staining or ELISA) in kidneys from MGME1^{-/-} mice.

We have carefully quantified the cytosolic cGAS signal and observed a significant increase in cytosolic cGAS relative to nuclear cGAS (new Extended Data Figure 4a, b; formerly Extended Data Fig. 3d). We haven't analyzed the subcellular localization of cGAS in kidney, a complex tissues containing at least 26 different cell types that are likely affected differently.

- F) Did the authors analyze additional ISGs? It would be interesting to provide a more comprehensive list of ISGs affected by mtDNA depletion. Also, can the same experiment be done in Mgme1^{-/-} MEFs – i.e. can the authors treat these MEFs with ddC to deplete mtDNA and assess ISG expression? Is there a particular reason why they chose to si Mgme1 rather than using Mgme1^{-/-} MEFs?

We have increased the number of ISGs whose expression was analyzed upon MGME1 depletion by NanoString analysis to 18 ISGs and show now that mtDNA depletion significantly reduces their expression (new Extended Data Fig. 5). In addition, we would like to point out that we have previously reported the identification of >180 ISGs that are upregulated in response to a cellular nucleotide imbalance. The expression of 90 of these genes was sensitive to ddC treatment, i.e. dependent on mtDNA (Sprenger et al., Nat. Metab., 2021; PMID: 33903774). For control, we routinely use both *Mgme1^{-/-}* cells and cells depleted of MGME1 and obtained similar results. As requested by the reviewer, we have also depleted mtDNA from *Mgme1^{-/-}* cells by ddC treatment and show that this ameliorates ISG expression (new Extended Data Fig. 5a).

- F) Please, provide PCR data to show that mtDNA depletion was successfully achieved following ddC treatment in these cells.

We now demonstrate that ddC treatment successfully depleted mtDNA both in *Mgme1^{-/-}* cells and after depletion of MGME1 in the new Extended Data Fig. 5b, c.

Extended data Figure 5:

- A and B) Does knocking down these genes have an impact on cytosolic leakage of mtDNA? Please, show these data.

We show in this experiment (now Extended Data Fig. 6a) that loss of MGME1, but not the knockdown of other components involved in mtDNA replication, induces ISG expression. MGME1 is not essential for mtDNA replication and our new next generation sequencing data even show an increased number of mtDNA replication events in *Mgme1^{-/-}* kidney (see response to point 1 of reviewer 1), which explains the specific effect of MGME1 loss on mtDNA dependent inflammation. Moreover, we provide additional evidence that ISG expression depends on mtDNA replication (new Extended Data Fig. 6b). Therefore, we did not perform cell fractionation experiments, which require careful controls and may need to be optimized for different conditions.

Extended data Figure 6:

- A) The authors show that 5-FU treatment, which inhibits cytosolic thymidylate synthase, increases the ratio of rU/dT. In the text they say “mtDNA-dependent innate immune signaling”... is associated with this increase. However, they do not show any data regarding the effects of 5-FU treatment on ISG expression and mtDNA cytosolic leakage. This is an important piece of data that would support their conclusions and should be included.

We apologize for the lack of clarity on this point. We have already published that 5-FU treatment induces mtDNA release and cGAS-STING signaling (Sprenger et al., Nat. Metab., 2021; PMID: 33903774). We show in the present manuscript that these treatments affect the rNTP/NTP ratio, leading to increased incorporation of rNTPs into mtDNA. We have now complemented these data analyzing the mtDNA of HU- and 5-FU-treated cells by alkaline gel electrophoresis showing increased rNTP incorporation after pharmacological interventions affecting the nucleotide metabolism (new Extended Data Fig. 8e).

Extended data Figure 7:

- D) It is difficult to see mitochondrial EdU foci from these images. Please, enlarge or show magnification of an area next to the images. Also, can the authors provide some type of quantification of EdU co-localization with TOM20 in both conditions?

We have improved and enlarged the images shown in Extended Data Fig. 8a (formerly Extended Data Fig. 7d) and included TOM20 stainings. Although quantification is difficult, the analysis shows unambiguously that mtDNA replication is ongoing (at least to some extent) in senescent cells.

Reply to Referee #3:

In this manuscript, Bahat et al. report that increased ribonucleotide misincorporation into mitochondrial DNA triggers mtDNA release, cGAS-STING activation, and inflammatory SASP gene expression. Through a series of genetic and biochemical experiments, the authors elucidate a role for elevated incorporation of rNTPs into mtDNA during states of stress (Yme1^{-/-}, Mgm1^{-/-}, ionizing radiation-induced senescence, chemotherapeutic-induced senescence) that result in depletion of cellular dNTP availability. This leads to increased misincorporation of rNTPs into mtDNA and promotes instability and linearization observed in the Mgm1^{-/-} model. Furthermore, the authors show that loss of STING is sufficient to reduce interferon-stimulated gene signatures in aged Mgm1^{-/-} mouse kidneys. Although there are interesting and important aspects of the study, the paper lacks mechanistic depth in several key areas. As presented, it doesn't dramatically advance knowledge in the mtDNA-cGAS-STING field beyond what other key papers have shown (including prior work from the authors themselves). Therefore, it is unclear that the paper represents enough of an advance to warrant publication in Nature.

We thank the reviewer for their careful assessment of our work, but respectfully disagree with their general statement about the advance provided to the field. The focus of the work is on the mechanisms that can induce mtDNA release rather than on the downstream cGAS-STING signaling cascade. Although a plethora of conditions have been described that lead to mtDNA release and although metabolic perturbations are emerging as important triggers of

this inflammatory response, the processes that occur at the mtDNA level are not known. Here, we show that rNTP incorporation into mtDNA causes mtDNA replication stress, which induces mtDNA release. We show that this mechanism occurs in several models in vivo and in vitro, is increasingly important with age and explains mtDNA release and SASP expression in senescent cells. We therefore believe that our findings significantly advance our current understanding of mtDNA-dependent inflammatory responses. While our study focuses on the mechanisms at the level of mtDNA that induce mtDNA release at the level of mtDNA, we agree with the reviewer that the characterization of mtDNA-dependent cGAS-STING signaling - although novel for MGME1-deficient cells - does not provide additional information on the downstream signaling cascade and therefore condensed this part of the manuscript (also to be able to accommodate our new findings).

Major points:

1. This manuscript leverages two mouse models that exhibit aberrant mtDNA phenotypes that contribute to pathology. Mgme1^{-/-} (PMID: 29572490) and Yme1l^{-/-} (PMID: 33903774) both exhibit increased mtDNA instability and elevated linearized mtDNA. The Mgme1^{-/-} was previously shown to exhibit elevated plasma cytokines including interferon gamma, as well as late-developing inflammatory kidney diseases characterized by lymphocyte infiltration. Previous work from the Langer group has also shown a nucleotide imbalance in the Yme1l^{-/-} and reported a role for mtDNA release and cGAS-STING activation in null MEFs. Thus, the first 3 figures of the paper do not provide much of a mechanistic advance and perhaps miss opportunities to document the relevance of this pathway for disease. The authors report that the DKO mouse lacking STING and MGME1 does not exhibit late-onset kidney inflammation and interferon-stimulated gene expression. However, the paper does not provide data examining how ablation of the inflammatory signature impacts disease phenotypes. Do DKO mice have improved kidney function or improved retina morphology? Does STING ablation reduce T cell infiltrates into Mgme1^{-/-} KO mouse kidneys? Do DKO mice have reduced circulating cytokines in plasma relative to Mgme1^{-/-} alone? Does STING ablation extend lifespan of Mgme1^{-/-} mice? In the absence of some functional phenotypic rescue, it is unclear whether the activation of cGAS in this model contributes to disease. Recently, cGAS ablation was shown to reduce interferon responses in an RNASEH2 mutant Aicardi-Goutières syndrome (AGS) mouse model, but not improve cerebellar defects observed (PMID: 34655526), suggesting that the IFN response is not directly responsible for neurodegeneration. Thus, it is critical that the authors reveal more phenotypic data from the DKO mouse to better solidify links between mtDNA-mediated inflammation and disease.

As pointed out by the reviewer, inflammatory kidney disease has previously been described in *Mgme1^{-/-}* mice, which originally drew our attention to this model. In Figures 1-3, we now show for the first time that the loss of *Mgme1* triggers mtDNA release and cGAS-STING signaling in vivo and in vitro. We show that mtDNA fragments, predominantly from regions in proximity to O_H, are released from MGME1-deficient mitochondria in vivo, establish a nucleotide imbalance in MGME1 deficiency and demonstrate the dependence of inflammatory signaling on ongoing mtDNA replication.

We have now extended our analysis and performed next generation sequencing of mtDNA in *Mgme1^{-/-}* kidneys (new Fig. 3c). We show that the loss of MGME1 increases replication initiation, which together with the known replication stalling in the absence of MGME1, suggests that increased nucleotide consumption leads to nucleotide imbalance and increased rNTP incorporation into mtDNA. Therefore, our experiments define the molecular basis of the observed inflammatory phenotype in *Mgme1^{-/-}* mice. Notably, although we observed an increased rNTP/dNTP ratio both in *Mgme1^{-/-}* and *Yme1l^{-/-}* models, the

underlying mechanisms leading to this nucleotide imbalance are different: while loss of MGME1 increases nucleotide consumption due to increased, unproductive mtDNA replication initiation (new Fig. 3c), loss of YME1L affects nucleotide synthesis and transport into mitochondria (Sprenger et al., Nat. Metab., 2021; PMID: 33903774).

Although our study focuses on processes at the level of the mtDNA that induce its release, we agree with the reviewer that our results in *Mgme1*^{-/-} kidney opens up the possibility to investigate the contribution of mtDNA dependent cGAS-STING signaling to renal disease. Therefore, we have now performed a detailed histological analysis of 55-week-old *Mgme1*^{-/-} and *Mgme1*^{-/-};*TMEM173*^{mut/mut} mice. Strikingly, deletion of *TMEM173* ameliorated the renal phenotypes of *Mgme1*^{-/-} mice, including glomerular sclerosis accompanied by the formation of proteinaceous casts, and peri-glomerular infiltrates of CD3-positive T cells. Our results suggest that the disruption of the glomerular filtration barrier is the causative factor for the observed renal phenotype and establish a critical role for mtDNA-dependent cGAS-STING signaling in the disease. The effects of STING ablation on *Mgme1*^{-/-} physiology and lifespan are going to be investigated further but are beyond the scope of the present study.

2. Why does MGME1 deficiency only result in mtDNA release and cGAS-STING activation in the kidney? This is a very interesting finding, but the paper fails to examine the mechanisms behind this. Are other body organs more resistant to rNTP incorporation into mtDNA? Are there RNASEH-like repair mechanisms in certain organs that are absent in the kidney? At the very least, the authors need to examine cytosolic mtDNA and ribonucleotide incorporation into mtDNA from other tissues as was done in Figs. 1,5 to show specificity for the kidney. Moreover, it is unclear where the cGAS-STING pathway is activated in the Mgme1^{-/-} KO kidneys. Is there a particular cell type that is susceptible to ribonucleotide incorporation and cGAS-STING activation?

We have analyzed rNTP incorporation into mtDNA isolated from different tissues (kidney, liver, heart, and spleen) of old (87 weeks) and young (1 week) wild-type mice and show that mtDNA in all tissues contains rNTPs (new Fig. 1c Extended Data Fig. 1d). Cell fractionation experiments in kidney and liver of wild-type vs *MGME1*^{-/-} combined with quantitative dPCR show increased cytosolic mtDNA in both tissues, however, to a much lower extent in the liver (new Fig. 1c). Nevertheless, we do observe increased rNMP incorporation into *Mgme1*^{-/-} liver (Figure R3).

Figure R3. Increased ribonucleotide incorporation into mtDNA from liver of 87-week-old *MGME1*^{-/-} mice.

Untreated and RNaseH2-treated mtDNA was resolved on alkaline agarose gels and further analyzed by Southern blot and quantified.

The restriction of the inflammatory response to *MGME1*-deficient kidney most likely relies on threshold effects. It is possible that the two tissues differ in the rate of mtDNA replication or in the capacity of defence mechanisms, such as expression of the DNase TREX1, which limits mtDNA dependent cGAS-STING signaling (Sprenger et al., Nat. Metab., 2021; PMID:

33903774). In addition, different types of replication stalling have been observed in liver and kidney (Matic et al., Nat. Com., 2018; PMID:29572490), which may contribute to the differences in the sensitivity of both tissues. Finally, our next generation sequencing data of *Mgme1*^{-/-} kidney show an increased coverage of regions in proximity to O_H indicating massive initiation of mtDNA replication, which occurs in heart and brain to the much lower extent (Matic et al., Nat. Com., 2018; PMID:29572490). Since the molecular basis of the observed tissue-specificity is likely the result of various effects and warrants in-depth studies that go beyond the scope of this manuscript, we only show our data on mtDNA release in *Mgme1*^{-/-} liver in the new Fig. 1c, but did not include the data on rNTP incorporation in the manuscript.

The kidney is often the target of immune-mediated damage in autoimmune diseases, but it is largely unknown which of the 26 different cell types are predominantly affected. We would also like to point out that high cell- and tissue-specificity is the hallmark of mitochondrial diseases, including mtDNA-linked diseases, although they are caused by mutations in ubiquitously expressed proteins. The kidney, as a mitochondria-rich tissue is often affected (in addition to brain and muscle tissue). For example, mice that lack the mitochondrial transporter MPV17 limiting uracil accumulation in mtDNA (Alonzo et al., J. Cell. Biol. 2018; PMID: 30385507) and leading to increased rNTP incorporation into mtDNA (Berglund et al., PLoS Genet., 2017; PMID: 28207748), develop focal segmental glomerulosclerosis (FSGS) with massive proteinuria (Viscomi et al., Hum. Mol. Genet., 2009; PMID:18818194).

3. RNA-DNA hybrids are known ligands for cGAS. Does mtDNA containing ribonucleotides serve as a more potent ligand for cGAS, or is it simply that dNTP imbalance and ribonucleotide misincorporation leads to increased mtDNA release and cGAS binding? More information here would increase understanding and broaden key findings of the paper.

rNTP incorporation to mtDNA is scattered and sequence and region independent (Berglund et al., PLoS Genet., 2017; PMID: 28207748; Forslund et al., PLoS Genet., 2018; PMID:29601571) and the enzyme RNASEH1 that is capable of removing longer stretches of rNMPs incorporated into mtDNA is present in mitochondria. Moreover downregulation of RNASEH1 (Figure 3a) that would lead to potential accumulation of long stretches of ribonucleotides into mtDNA does not elicit ISG signaling. In addition, given the low frequency of rNTP incorporation into mtDNA, DNA-RNA hybrids if released may reflect only a very small fraction of mtDNA fragments. We therefore strongly favor that slower mtDNA replication after rNTP incorporation (and replication stalling in the absence of MGME1) leads to increased mtDNA release and cGAS signaling. Similarly, an Aicardi-Goutieres syndrome-related mutation in RNaseH2, a component of the nuclear base excision repair machinery, causes increased rNTP incorporation into nuclear DNA and increased cGAS-STING signaling (Aditi et al., Neuron, 2021; PMID: 34655526).

4. The senescence data in Fig. 6 are interesting, but still preliminary. The identification that ribonucleotide incorporation into mtDNA underlies SASP is quite interesting, but it is unclear how translatable these findings are to the rest of the study. For example, it is unclear why kidney cell lines weren't utilized here to tie into the mouse inflammatory pathology. Do the authors see increased numbers of senescent cells in aged MGME1 KO kidneys, and if so, are these cells the sources of cGAS-STING activation? Does ablation of STING in MGME1 KO kidneys impact senescence phenotypes, if

any?

We have extended our analysis of mtDNA release in senescent cells and demonstrate now directly that nucleoside complementation suppresses both mtDNA release and ISG expression (new Fig. 5c; new Extended Data Fig. 9i). Moreover, we show now rNTP incorporation and SASP in senescent kidney epithelial cells (new Extended Data Fig. 9g). The analysis of the expression of senescence marker genes are upregulated in kidney lysates of 55- and 70-week-old *Mgme1*^{-/-} mice (Figure R3). However, the interpretation of these data is difficult considering the complex cellular composition of kidneys, which would require a detailed scRNA seq analysis to identify specific cell types affected.

Figure R3. Increased expression of senescence markers in kidney lysates of *Mgme1*^{-/-} mice. a,b, RT-qPCR expression analysis in kidneys from 55-week-old (a) and 70-week-old (b) animals of denoted genotypes.

Other questions/concerns:

1. Extended data Fig 3 suggests that knocking down *STING* or *cGAS* in immortalized *MGME1* KO MEFs reduces ISG expression. While this is clear, knock down does not entirely ablate ISG expression as is shown in the DKO kidneys. Is this due to a limited efficiency of knock down in the *Mgme1*^{-/-} cells or are there other sensors that work with *cGAS* or perhaps even directly through *STING* to drive IFN in this model? RNA sensors don't seem to be active, but is there any role for other cytosolic DNA sensors (*IFI200/IFI16* family members, for example)?

As suggested by the reviewer, the partial suppression of ISG expression in the knockdown experiments shown in Fig. 3 are likely explained by inefficient depletion. However, we do not exclude the involvement of other sensors in response to mtDNA release induced by rNTP incorporation into mtDNA. Indeed, in preliminary experiments we observe effects of *AIM2* or *MYD88* depletion (Figure R4). Since the role of various sensors likely depend on the cell line analyzed and the expression of some sensors may also be interdependent, this deserves an in-depth analysis of the downstream signaling cascade, which goes beyond the scope of the present study. We therefore prefer to not include these data in the manuscript.

Figure R4. Downregulation of AIM2 and MYD88 downregulation reduces ISG expression in *Mgme1*^{-/-} MEFs. RT-qPCR analysis of ISG expression in *Mgme1*^{+/+} and *Mgme1*^{-/-} MEFs (relative to *Mgme1*^{+/+}) treated with the indicated siRNA for 72 h.

2. In Figure 3 the authors argue that depletion of *Twinkle*, *Samhd1*, or *Slc25a33* suppresses the ISG response in *Mgme1*^{-/-} MEFs. The authors should examine and quantify cytosolic mtDNA in all of these double mutant conditions.

We have previously analyzed the effect of SAMHD1 and SLC25A33 on nucleotide metabolism, mtDNA release and ISG expression in *Yme1*^{-/-} MEFs and therefore feel that these technically challenging experiments would not provide further mechanistic insight into the main conclusion of the present manuscript that nucleotide imbalance causes increased rNTP incorporation into mtDNA.

3. The confocal microscopy is not sufficient as displayed. While the quantitation of some images is generally convincing, the quality of the representative images and lack of global quantification reduce enthusiasm for these findings. Specifically in Figure 2d, the image provided needs to be enlarged with labels for the different stains and their colors in the figure. This image could also be improved by including a zoom. In extended data Fig. 3d, the images provided do not convincingly support the conclusion that cGAS has a cytosolic localization in the *Mgme1*^{-/-} MEFs. In the images provided the *Mgme1*^{-/-} MEFs seem to still have a large amount of cGAS in the nucleus and the cytosolic portion is difficult to discern. This could be improved by including a whole cell marker and quantitating cGAS intensity in the nucleus vs cytoplasm based on these markers. In figure 4d, a zoomed image and inclusion of mitochondrial and whole cell markers would improve confidence that the spots labeled are freshly synthesized mtDNA. In extended data Fig. 7, please provide a zoom image and quantitation to prove less new mtDNA colocalizing with TOMM20.

We have improved the quality of all confocal images shown in the manuscript (Fig. 2c; Extended Data Figs., 4a; 8a; 9a). Fig. 2c (formerly Fig. 2d) includes now labels for the different stains and a zoom. We have also carefully quantified cytosolic cGAS in *Mgme1*^{-/-} MEFs in Extended Data Fig. 4a, b (formerly Extended Data Fig. 3d). We also included a mitochondrial marker and a zoom image in the Extended Data Fig. 8a (formerly Fig. 4d). Similarly, we provide a zoom image and a mitochondrial marker in Extended Data Fig. 9a.

4. In extended data Fig. 3f the authors use ddC in MEFs to conclude that loss of mtDNA inhibited immune signaling however they did not prove that they depleted mtDNA and the results shown do not reach significance. Can this be expanded on? Additionally, this experiment was done using MGME1 knock down in WT MEFs and should be repeated in the *Mgme1*^{-/-} MEFs.

We apologize for not including the statistical analysis in the experiment shown in the original version of the manuscript, which confirmed that the observed effects were statistically

significant (formerly Extended Data Fig. 3f). We have now analyzed the effect of mtDNA depletion by ddC in *Mgme1*^{-/-} cells and after siRNA-mediated knockdown of *Mgme1* using an larger panel of ISGs (new Extended Data Fig. 5). We also included controls to demonstrate efficient depletion of mtDNA (new Extended Data Fig. 5b, c). The results substantiate our conclusion that the observed ISG expression depends on mtDNA.

5. Data from extended data Fig. 5 is used to conclude an increase in rNMPs in Mgme1^{-/-}, however these findings do not appear to reach statistical significance.

The increase in rNMPs in *Mgme1*^{-/-} cells is statistically significant, as now indicated in Extended Data Fig. 6d (formerly Extended Data Fig. 5c). We apologize for this mistake in the original version of the manuscript.

6. More thorough labeling of figures would improve readability of the manuscript. For example, clearly delineating “cytosolic kidney extracts” in Figure 1d and “whole cell kidney extracts in extended data Fig. 1e.

We have improved the labeling of the figures and figure captions to improve the readability of the manuscript.

7. “Each column represents a different mouse” is repeated twice in the legend of extended data Fig. 1 a.

We have corrected the figure caption.

8. In extended data Fig. 2 b, there is a discrepancy between 18s rDNA shown in the figure and 18s rRNA mentioned in the figure legend.

We have corrected this typo.

9. The text describing Fig 1d is out of place and should be moved later in the manuscript to improve readability.

The experiments in Figure 1 (and Figure 2) show the release of mtDNA fragments and ISG expression in *Mgme1*-deficient tissues (now shown for kidney and liver in Fig. 1c), whereas the experiments in Fig. 2 establish cGAS-STING signaling in the absence of MGME1. Figs. 3 and 4 then focus on the mechanistic analysis. Therefore, we prefer to describe the data shown in Fig. 1c (formerly Fig. 1d) in this context, but adjusted the text to improve readability.

Reply to Referees' comments on the revised manuscript

Referee #1 (Remarks to the Author):

In their revised manuscript, the authors have addressed my concerns. The revised version has improved, and the addition of new data further strengthens their findings and model. I was particularly excited to see the increase in rNTP/dNTP ratio in aged mice as well as the increase sensitivity to alkaline/RNaseH2 treatment. I think this study will be of great interest to the readership. As a very minor point, I might suggest labeling the tissue in Figures 6a, 6c, ED10a, and ED10c (even though it is indicated on 6b/6d/ED10b/ED10d).

We can add labeling as suggested.

Referee #2 (Remarks to the Author):

The authors have addressed my concerns, and I recommend that the manuscript is acceptable for publication.

Referee #3 (Remarks to the Author):

In their revised manuscript, Bahat, Milenkovic et al. expand their findings to show that loss of MGME1 promotes increased mtDNA replication initiation. This depletes mitochondrial deoxynucleotide pools and increases ribonucleotide incorporation into mtDNA, causing mtDNA release and activation of the cGAS-STING pathway. Interestingly, the sequences that accumulate in the cytosol of cells and kidneys of MGME1^{-/-} mice are proximal to O_H. The inclusion of next generation sequencing and additional tissue analysis has strengthened the study. However, key open questions remain that warrant additional clarification before this study is suitable for publication in Nature.

1. I thank the authors for providing more data on the kidney immune and pathological phenotypes in DKO mice. They clearly show a role for STING in both kidney pathology and infiltration of T and B cells at late timepoints. The overall mechanism presented is that mtDNA release, presumably in Mgm1^{-/-} kidney cells, drives cGAS-STING signaling, leading to a cell-intrinsic activation of immunity/inflammation that causes kidney failure. Experiments from MEFs show that STING knockdown/TBK1 inhibition is much more potent at blocking interferon genes than cGAS knockdown. STING can integrate many signals from nucleic acid sensors and mediates many responses beyond interferon signaling, including cell death, which can occur in a cGAS-independent pathway in vivo. STING activation, in both cGAS-dependent and -independent mechanisms can also alter T cell activity in vivo, which may compound the kidney injury of Mgm1^{-/-} mice. Given that no data are provided to untangle where the interferon/immune response initiates in Mgm1^{-/-} kidneys, and whether the interferon signature initiates from resident kidney cells or infiltrating immune cells (i.e. via signal cell approaches), evidence for in vivo cGAS-dependency are limited. In the absence of more conclusive data, the authors should remove cGAS from the title and focus the story on STING.

We agree that STING has recently been shown to be multifunctional, and we have only demonstrated cGAS dependency in primary Mgm1^{-/-} fibroblasts. Therefore, we can change the wording throughout the manuscript to refer only to STING dependency when discussing our *in vivo* experiments, thus avoiding confusion.

2. The mechanisms explaining why inflammatory signatures are only seen in the kidney of aged Mgm1^{-/-} mice remain mostly unaddressed. Why does it take over a year for the immune response to be detectable in the kidney, when increased mtDNA replication and ribonucleotide incorporation is likely happening from early life onward? Do young Mgm1^{-/-} mice have increased mtDNA in kidney extracts, and if so, why is this not sufficient to activate the STING pathway? Could it be that Mgm1 dependent mitochondrial deficits compromise the tissue over time, leading to immune infiltrates that express interferon genes at high levels and further damage the kidney? Moreover, the increase in ribonucleotide incorporation and

cytosolic mtDNA in liver is interesting, especially because no immune response is seen in these tissues. The resident immune profile of the kidney is quite different from the liver, and perhaps this is responsible for differences in interferon and inflammatory gene activation. However, in their rebuttal letter, the authors discuss potential thresholding effects and propose Trex1 might be higher in non-kidney tissues to control sensing of mtDNA. But if this were the case, higher Trex1 would result in reduced mtDNA in Mgme1^{-/-} liver, and the authors report significantly increased cytosolic mtDNA there. I feel that the absence of molecular and cellular mechanisms here decrease the novelty of the story, because much of what the authors report in the paper has already been documented in other published studies (i.e. mtDNA replication defects in Mgme1^{-/-}, mitochondrial nucleotide pools impacting ribonucleotide incorporation in mtDNA, inflammation in Mgme1^{-/-} kidneys). The tissue specificity of this model is interesting and a better understanding could help explain how cytosolic mtDNA sensing specifically drives immune activation in the kidney, which the authors note is a frequent target in autoimmune and auto-inflammatory disorders.

Cell- and tissue-specificity is a hallmark of all mitochondrial diseases and is hardly understood for any the various diseases. As outlined in our rebuttal letter, a kidney phenotype has been observed in some mitochondrial diseases that intriguingly are all linked to the cellular nucleotide metabolism (MPV17, DUOGK). Replication defects in *Mgme1^{-/-}* mice are tissue-specific, and our NGS data suggest that the rate at which mtDNA replication initiates differs between tissues. This difference may account for the tissue-specificity of the disease. Similarly, differences in immune profiles or nucleotide metabolism (e.g., nucleotide concentrations, *de novo* synthesis rates, nucleotide salvage rates, and SAMHD1 levels) can contribute to tissue specificity.

Evaluating these possibilities and identifying, which of the approximately 30 cell types in the kidney are most affected, are intriguing questions that require extensive experimentation, representing a project in and of itself. Most importantly, this does not contribute to our main discovery and conclusion in the manuscript regarding the role of ribonucleotide incorporation into mtDNA as a trigger for mtDNA release and inflammation.

Some comments to the questions raised by the reviewer:

a) Why does it take over a year for the immune response to be detectable in the kidney, when increased mtDNA replication and ribonucleotide incorporation is likely happening from early life onward? Do young Mgme1^{-/-} mice have increased mtDNA in kidney extracts, and if so, why is this not sufficient to activate the STING pathway?

Given the numerous factors that influence nucleotide metabolism and mtDNA replication, deficiencies and tissue damage likely accumulate over time, leading to pathophysiological consequences only when they surpass a threshold. Notably, infiltration of immune cells over time may also contribute to the pathophysiology of the disease. To address the reviewer's concern, we can perform tissue fractionation experiments and analyze cytosolic mtDNA in young animals. We can also monitor the steady-state levels of TREX1 and SAMHD1 in the kidneys and livers. However, in our opinion, this would only reveal additional tissue-specific differences and would not allow us to draw significant conclusions concerning the reviewer's general concern about tissue specificity.

b) Moreover, the increase in ribonucleotide incorporation and cytosolic mtDNA in liver is interesting, especially because no immune response is seen in these tissues.

Our quantitative data show that a drastically lower amount of mtDNA is found in the cytosol of liver when compared to kidney, which explains tissue-specific inflammation. As discussed, absolute nucleotide concentrations (and therefore the

susceptibility for disturbances), the activity of different nucleotide synthesis pathways and defence mechanisms (such as TREX1 or also SAMHD1) may vary between tissues. We would also like to point out that in contrast to the assumption of the reviewer, TREX1 activity does not reduce mtDNA levels under normal conditions because it is only acting on mtDNA after its release to the cytosol (as we have shown in our previous publication (Sprenger et al., 2021)).

We agree with the reviewer that the molecular function of MGME1 has been studied by several groups before. However, the molecular characterization of MGME1 function and its pathophysiological consequences was not the intention of this study and therefore was not in the focus of our experiments. Instead, we used *Mgme1*^{-/-} deficient mice as a model to test the physiological relevance of ribonucleotide incorporation into mtDNA *in vivo* and to demonstrate its role in the aged tissue.

3. Regarding the senescence aspects of the study, the authors do not provide any data documenting that cGAS-STING pathway is required for senescence or the immune/SASP gene signature seen. No genetic experiments were done to show that ablation of STING inhibits senescence phenotypes or SASP. The data do support that increased nucleotide pool imbalance and ribonucleotide incorporation into mtDNA does occur in senescent cells in vitro, and that this is correlated with increased mtDNA release. dN supplementation brings down expression of some SASP genes (IL1B, etc), but curiously the authors did not examine interferon-induced genes as in Figs 1-2, or show that the dN-dependent decrease in SASP genes is dependent on reduced cGAS, STING signaling. In figure R3, the expression of senescence marker genes is not dependent on STING, which argues that the senescence and mtDNA-STING phenotypes may be separate. The authors argue that they have identified rNTP incorporation into mtDNA as a central mechanism leading to release and STING activation in both their genetic models, in vitro senescence, and in vivo aging, yet the data do not convincingly show this to be the case. They also did not examine abundance of senescent cells in the kidney of Mgme1^{-/-}, or test whether the SASP is what recruits T cells into the kidney. No evidence was provided elevated ribonucleotide incorporation into the aged heart and liver mtDNA is linked to increased STING activation. Overall, the last two figures of the paper are critical in extending the findings beyond the Mgme1 model, but the data are largely correlative. It is this unclear whether increased rNTP incorporation into mtDNA is truly the unifying mechanism leading to mtDNA release, STING activation, and senescence.

We have not added any data on the involvement of the cGAS-STING pathway in SASP to the manuscript as this has been published repeatedly before (Yang et al., PNAS, 2017; Guo et al., CDD, 2021; Gulen et al., Nature, 2023; Victorelli et al., Nature, 2023; Liu et al., JCI, 2023) and was one of the reasons why we have chosen senescent cells as a model to study the role of ribonucleotide incorporation into mtDNA for inflammatory responses. However, we can add (confirmatory) data to the manuscript to comply with the request of the reviewer, which – as we note - was not raised in their comments on the original manuscript. Similarly, we can show the effect of nucleoside complementation on additional SASP genes/ISGs (for instance by Nanostring data as in Fig. 1 and 2). However, we would like to emphasize that our nucleoside complementation studies, rather than being correlative, unambiguously demonstrate the critical role of nucleotide disturbances for rNTP incorporation into mtDNA, mtDNA release and SASP in senescent cells.

In figure R3, the expression of senescence marker genes is not dependent on STING, which argues that the senescence and mtDNA-STING phenotypes may be separate.

The establishment of the senescent state and SASP are indeed separable phenotypes as shown in numerous studies before. Therefore, STING ablation (in figure R3) is expected not to interfere with the expression of senescence markers.

Similarly, nucleoside complementation did not interfere with their expression but suppressed SASP.

They also did not examine abundance of senescent cells in the kidney of $Mgme1^{-/-}$, or test whether the SASP is what recruits T cells into the kidney.

Although we demonstrate that the loss of STING reduces immune infiltration into $Mgme1^{-/-}$ kidney *in vivo* (Fig. 2) and although SASP expression is known to be STING dependent (Yang et al., PNAS, 2017; Guo et al., CDD, 2021; Gulen et al., Nature, 2023; Victorelli et al., Nature, 2023; Liu et al., JCI, 2023), our experiments did not directly identify senescent cells in the kidney. We did not intend to propose that the kidney-specific phenotype of $Mgme1^{-/-}$ mice is caused by increased senescence, but rather show different physiological and pathophysiological conditions that are associated with increased rNTP incorporation into mtDNA and inflammation. As outlined in response to point 3, the detailed characterization of the kidney-specific phenotype of $Mgme1^{-/-}$ mice is in our opinion beyond the scope of the present manuscript.

No evidence was provided elevated ribonucleotide incorporation into the aged heart and liver mtDNA is linked to increased STING activation.

In response to reviewer's comments on the original manuscript, we have analysed rNTP incorporation into mtDNA in various aged tissues, but only demonstrate STING dependency in the kidney (Fig. 2), since STING deficient tissues of old-age mice were not available. It appears likely that the contribution of an increased rNTP incorporation into mtDNA to inflammaging is tissue- and context dependent and may also involve nuclear DNA, justifying interesting follow-up studies, which however we strongly feel are beyond of the scope of this manuscript.

4. The authors provide evidence in the response that other sensors may be involved in sensing ribonucleotide incorporation and mtDNA release in the $Mgme1^{-/-}$ model. These data are compelling and suggest a more complex immune complex immune mechanisms than the linear cGAS-STING pathway nominated in in the manuscript. This partial rescue of kidney pathology in the DKO implicates other immunological mechanisms beyond STING, so I believe the authors must include these data for all readers and discuss them in the paper.

We agree with the reviewer that the immune response to $Mgme1^{-/-}$ deficiency is likely more complex and may involve other DNA sensors in a tissue- and cell-type specific manner. We can therefore include and discuss these data as suggested by the reviewer.